# Mechanisms of PfDNMT2 inhibition and PfATP6-mediated resistance to the antimalarial candidate SC83288 in *Plasmodium falciparum*

Cecilia P. Sanchez[1], Maëlle Duffey[1,2], Romina V. Celada[1], Michal Kucharski[3,4], Marie Hoarau[5], Thanaya Saeyang[5], Sumalee Kamchonwongpaisan[5], Farzin Sohraby[6], Svenja de Buhr[1], Ariane Nunes Alves[6], Amuza Byaruhanga Lucky[7], Jun Miao[7], Clement Regnault[8], Jerzy M. Dziekan[3,9], Zbynek Bozdech[3], Michael P. Barrett[10] & Michael Lanzer[1] ✉

The emergence of multi-drug resistant *Plasmodium falciparum* underscores the urgent need for new antimalarial therapies. SC83288, a chemically distinct antimalarial compound, is highly effective against *P. falciparum* both in vivo and in vitro, including strains resistant to artemisinin and partner drugs. Here, we show that SC83288 disrupts blood-stage development by blocking DNA replication and arresting karyokinesis. We identify the parasite's DNA and tRNA[Asp] methyltransferase PfDNMT2 as a primary molecular target, linking drug action to impaired epigenetic regulation, altered S-adenosylmethionine fluxes, and compensatory transcriptional responses. Resistance to SC83288 arises through mutations in the parasite's SERCA-type $Ca^{2+}$ ATPase PfATP6, which enable transport of the compound into the endoplasmic reticulum, away from its nuclear targets. This resistance mechanism carries a substantial fitness cost, limiting its potential for spread. Together, target validation, a unique resistance profile, and high fitness cost strengthen SC83288's potential as a promising clinical development candidate for malaria treatment.

Malaria remains a major infectious disease, with an estimated 263 million cases and ~597,000 deaths in 2023 alone, primarily in sub-Saharan Africa and mainly affecting children under the age of five[1]. Malaria is caused by protozoan parasites of the genus *Plasmodium*, with *Plasmodium falciparum* being responsible for the most virulent form. The pathology of falciparum malaria is associated with the intraerythrocytic life cycle of the parasite and initially manifests itself in the form of chills, fevers, nausea, headaches and vomiting. In severe cases,

[1]Center for Infectious Diseases, Parasitology, University Hospital Heidelberg, Medical Faculty, Heidelberg University, Heidelberg, Germany. [2]The Global Antibiotic Research & Development Partnership, Geneva, Switzerland. [3]School of Biological Sciences, Nanyang Technological University, Singapore, Singapore. [4]Amsterdam UMC, University of Amsterdam, Department of Global Health, Amsterdam Institute for Global Health and Development, Amsterdam, The Netherlands. [5]National Center for Genetic Engineering and Biotechnology (BIOTEC), National Science and Technology Development Agency, Thailand Science Park, Khlong Luang, Pathum Thani, Thailand. [6]Institute of Chemistry, Technische Universität Berlin, Berlin, Germany. [7]Division of Infectious Diseases and International Medicine, Department of Internal Medicine Morsani College of Medicine, University of South Florida, Tampa, FL, USA. [8]Glasgow Shared Research Facility, College of Medical, Veterinary & Life Sciences, University of Glasgow, Glasgow, UK. [9]Walter and Eliza Hall Institute, Parkville, VIC, Australia. [10]School of Infection & Immunity, College of Medical, Veterinary and Life Sciences, University of Glasgow, Glasgow, UK. ✉e-mail: Michael.Lanzer@med.uni-heidelberg.de

falciparum malaria can progress to an acute respiratory syndrome, severe anemia, and impaired consciousness, potentially resulting in multi-organ failure, coma and eventually death if left untreated.

The treatment of malaria primarily relies on three major drug classes: artemisinins, aryl aminoalcohol compounds, including quinoline and quinoline-like drugs, and antifolates[2]. Yet, the efficacy of these medications is under threat, with *P. falciparum* demonstrating a remarkable capability to adapt to drug pressure. Resistance impacts monotherapeutic use but also increasingly compromises the effectiveness of drug combinations[3,4]. A particularly worrying trend is the emergence and spread of *P. falciparum* lines that are partially resistant to artemisinin-based combinations, the first line treatment against uncomplicated falciparum malaria and the cornerstone of current malaria control strategies[5]. Hence, there is a need for novel drugs that not only promise heightened efficacy but also prioritize safety and availability given that malaria is a disease that predominantly affects vulnerable populations in resource-limited settings.

We previously conducted a medicinal chemistry program inspired by amicarbalide, a benzamidine derivative formerly used as a veterinary antiprotozoal drug[6,7]. This effort resulted in SC83288 (Fig. 1a), a compound with enhanced antiplasmodial and pharmacological properties[8]. SC83288 meets multiple criteria outlined in the target profile for next-generation antimalaria drugs. The compound features novel chemistry and demonstrates potent activity against the disease-causing asexual blood stages of *P. falciparum*, achieving low nanomolar $IC_{50}$, $IC_{90}$, $IC_{99}$ values (3 nM, 8 nM, and 20 nM, respectively), a 99.9% parasite clearance time ($PCT_{99.9\%}$) of 48 h, and a parasite reduction rate (logPRR) of 3.0 in vitro and 5.3 in vivo[8,9]. SC93288 is effective against single and multi-drug resistant *P. falciparum* strains, including those resistant to artemisinin, antifolates, sulfa-drugs and quinoline and quinoline-like compounds. Resistance to SC83288 emerges slowly under in vitro culture conditions[8,9]. The compound also targets early-stage gametocytes, with an $IC_{50}$ value of $76 \pm 6$ nM; and it cures a *P. falciparum* infection in the humanized NOD/SCID mouse model following a single dose regimen of 5 mg kg$^{-1}$ administered for three consecutive days[8]. Additionally, comprehensive preclinical regulatory evaluation, including acute and repeat-dose toxicity studies in two species, safety pharmacology and genetic toxicology, revealed no significant safety liabilities, supporting the compound's suitability for first-in-human studies.

Here, we have investigated the mode of action and the mechanism of resistance of SC83288, using a combination of genetic, biochemical, omics, and in silico approaches. We identify PfDNMT2 as a primary target of the drug, resulting in loss of epigenetic control and ultimately inhibition of DNA replication during karyokinesis. SC83288 resistance is mediated by PfATP6, the parasite's orthologue of the mammalian sarcoplasmic and endoplasmic reticulum Ca$^{2+}$ ATPases, which sequesters the drug in the endoplasmic reticulum (ER) away from its nuclear targets at the expense of a heavy fitness cost.

## Results

### Mutations in PfATP6 confer SC83288 resistance

*P. falciparum* lines previously selected on gradually increasing concentrations of SC83288 or its closely related derivative SC81458 over the course of one year, harbored point mutations in genes encoding PfATP6 (Pf3D7_0106300), PfMDR2 (Pf3D7_1447900) and a putative ATP-dependent RNA helicase DBP9 (Pf3D7_1241800), as revealed by ultra-deep sequencing[8]. In addition, four out of five SC83288-resistant clonal lines investigated had the PfATP6 locus amplified[8], suggesting that PfATP6, the sarcoplasmic reticulum Ca$^{2+}$ pump of the parasite, is under strong selective pressure in the presence of SC83288 and its analogues. Notably, strong cross-resistance was observed between SC83288- and SC81458-resistant lines, further supporting the conclusion that both compounds share a common resistance determinant and likely interact with PfATP6 or associated pathways.

To assess PfATP6's role in SC83288 responsiveness, we introduced the phenylalanine to tyrosine substitution at position 972 on transmembrane domain 5, found in resistant lines (Fig. 1b and Supplementary Fig. 1), into the genomic locus of the wild type *P. falciparum* line 3D7, using CRISPR/Cas9 genome editing technology[10]. Several clonal lines were obtained by limiting dilution, and the anticipated nucleotide substitutions were confirmed through DNA sequencing. Intriguingly, the mutants (as represented by clone E4) were highly SC83288-resistant ($5 \pm 0.3$ μM versus $8 \pm 2$ nM; $p < 0.001$, two-sided Student´s $t$ test) (Fig. 1c). Notably, the F972Y mutation in PfATP6 incurred a substantial fitness cost, as is evident from growth competition assays where the 3D7 line almost entirely outcompeted the mutant within 20 replicative cycles (Fig. 1d).

### PfATP6 transports SC83288

The observed decline in fitness suggests that the mutation impacts PfATP6's function as a regulator of the intracellular Ca$^{2+}$ concentration. To test this hypothesis, we determined the cytosolic resting free Ca$^{2+}$ concentration, using the ratiometric Ca$^{2+}$ sensor Fura-Red in a live cell confocal set-up, combined with single cell in situ calibration of the fluorescence ratios. We found that the PfATP6 mutant line E4 exhibited a significantly lower cytosolic resting free Ca$^{2+}$ concentration compared with 3D7 ($260 \pm 40$ nM versus $420 \pm 70$ nM; $p < 0.001$, two-sided Mann–Whitney Rank Sum Test) (Fig. 1e, f). The value obtained for 3D7 aligns with previous determinations[11].

Treatment of the cells with 400 nM of SC83288 ($0.4 \times$ the $IC_{50}$-value of E4) for 60 min significantly elevated the cytosolic free Ca$^{2+}$ concentration in E4 to $630 \pm 40$ nM ($p < 0.001$, Kruskal–Wallis one-way ANOVA on ranks) (Fig. 1f). In contrast, SC83288 had no effect on the resting Ca$^{2+}$ concentration in 3D7, which remained at ~420 nM (Fig. 1f). Continuous, real-time live cell measurements employing the Ca$^{2+}$ sensitive fluorochrome Fluo-4 corroborated these results. Once more, exposure to SC83288 had no impact on the cytosolic free Ca$^{2+}$ concentration in 3D7, but led to a steady increase in the PfATP6 mutant line (Fig. 1g). This contrasted with the rapid, hyperbolic Ca$^{2+}$ response induced by the addition of cyclopiazonic acid (CPA, 10 μM), an established reversible inhibitor of PfATP6[12] (Fig. 1g). The Na$^+$/K$^+$ ATPase inhibitor ouabain (10 μM) had no discernable effect on the cytoplasmic Ca$^{2+}$ concentration and served as a negative control (Fig. 1g).

The SC83288-induced rise in cytosolic free Ca$^{2+}$ would be consistent with a competition reaction in which both Ca$^{2+}$ and SC83288 compete for transport by the mutated PfATP6. To test this hypothesis, we expressed both the wild-type and the mutated PfATP6 (PfATP6$^{F972Y}$) in the *Saccharomyces cerevisiae* K667 strain (Fig. 2a). The K667 strain lacks the endogenous vacuolar Ca$^{2+}$ exchanger Vcx1, the calcineurin regulatory subunit B cnb1, and the vacuolar calcium pump Pmc1, rendering it incapable of growth under high Ca$^{2+}$ conditions[13,14]. Both the wild-type and the mutant PfATP6 could complement this deficiency, enabling the yeast cells to grow under non-permissive conditions in the presence of high Ca$^{2+}$ (Fig. 2b).

Subsequently, we isolated vesicles from the yeast strains (latency of 98%) and incubated them for 15 min at 30°C in buffer containing tritiated SC83288 (100 nM), with or without ATP (1 mM), CaCl$_2$ (10 μM), CPA (30 μM), and NP40 (0.1%). PfATP6$^{F972Y}$ containing vesicles took up significant amounts of radiolabel in an ATP-dependent and CPA-sensitive manner, compared with NP40-treated vesicles or experiments conducted without ATP ($p = 0.015$ and $p < 0.001$, respectively; Kruskal–Wallis one-way ANOVA on ranks) (Fig. 2c). Moreover, uptake of SC83288 was inhibited by a 100-fold molar excess of CaCl$_2$ over SC83288 (Fig. 2c), indicative of Ca$^{2+}$ outcompeting the compound for PfATP6-mediated transport. Although vesicles containing the wild-type PfATP6 also took up SC83288, the extent of uptake was considerably lower than in vesicles containing the PfATP6$^{F972Y}$ variant ($p < 0.001$, two-sided Student's $t$ test) (Fig. 2c). Together, these transport studies demonstrate that PfATP6, particularly its F972Y mutant,

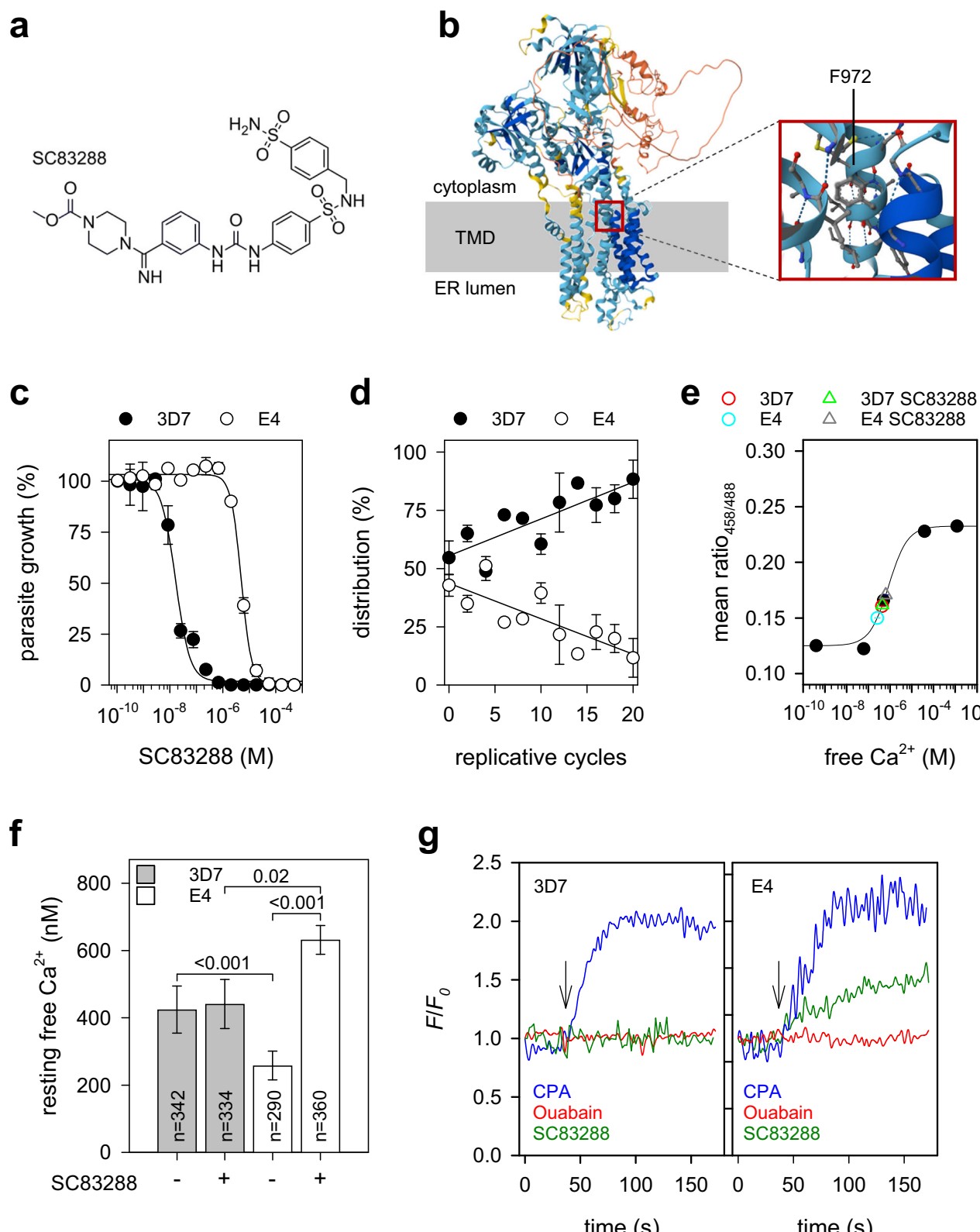

can mediate SC83288 transport and that this process functionally interferes with Ca²⁺ handling.

## PfATP6 diverts SC83288 to the ER

We next examined the accumulation kinetics of tritiated SC83288 from an external concentration of 50 nM by uninfected red blood cells and erythrocytes infected with 3D7 and E4. Both parasite lines accumulated the compound to comparable amounts over time, reaching a plateau accumulation ratio of ~100-fold ($F_{(2,10)} = 5.38$, $p = 0.026$; $F$-test) (Fig. 3a). In comparison, uninfected red blood cells did not take up the compound (Fig. 3a). Uptake of SC83288 by infected erythrocytes occurred independently of glucose and the proton ionophore carbonyl cyanide-p-trifluoromethoxyphenylhydrazone (FCCP, 10 μM), but was susceptible to inhibition by 5-Nitro-2-(3-

**Fig. 1 | Mutation in PfATP6 confers resistance to SC83288. a** Chemical structure of SC83288. **b** Alphafold model of PfATP6 highlighting residue Phe972 (red box) on transmembrane helix 5, replaced by Tyr in SC83288-resistant parasites. **c** Growth inhibition curves for wild-type 3D7 (black circles) and resistant line E4 carrying the PfATP6 F972Y substitution (white circles). Data represent mean ± SEM from five independent biological replicates, with curves fitted using a one-parameter sigmoidal function. **d** Competitive growth assay showing the proportion of 3D7 and PfATP6 F972Y (E4) parasites in mixed culture over time. The means ± SEM of three independent biological replicates are shown. **e** In situ calibration of the ratiometric $Ca^{2+}$ indicator Fura-Red showing fluorescence ratio (458/488 nm) as a function of free $Ca^{2+}$ concentration (black circles and line). Live-cell measurements from 3D7

and E4 parasites, with or without SC83288, are overlaid. **f** Mean cytoplasmic resting $Ca^{2+}$ concentration in untreated and SC83288-treated parasites. Bars show the means ± SEM of n independent single cell measurements; statistical significance determined by Kruskal–Wallis one-way ANOVA on ranks with all pairwise multiple comparison procedures (Dunn's Method; two-sided; Bonferroni adjusted). p-values are shown. **g** Cytoplasmic $Ca^{2+}$ transients in 3D7 and E4 parasites exposed to SC83288 (10 μM, green), ouabain (10 μM, red), or cyclopiazonic acid (10 μM, blue). Trophozoite-stage parasites were loaded with Fluo-4 AM (10 μM) and recorded by live-cell confocal microscopy. Traces show normalized fluorescence relative to baseline ($t = 0$). For subfigures c to g, source data are provided as a Source Data file.

phenylpropylamino) benzoic acid (NPPB, 5 μM), a blocker of parasite-induced new permeation pathways (NPP/PSAC) in the red blood cell plasma membrane[15,16] (Fig. 3b). These findings suggest that SC83288 uptake by infected erythrocytes is facilitated by NPP/PSAC and does not require an energy input or an intracellular proton gradient.

We also investigated the efflux of SC83288 from cells preloaded with the compound, but observed no discernable differences between E4 and the parental 3D7 line ($F(2,8) = 0.37$, $p = 0.7$; F-test) (Fig. 3c). As both 3D7 and E4 displayed similar SC83288 uptake and efflux kinetics, we speculated on PfATP6^F972Y influencing the compound's intracellular distribution, potentially concentrating it within the ER, where the compound might have reduced harmful effects on the parasite.

To localize SC83288 in infected erythrocytes, we synthesized a clickable derivative containing a terminal alkylene group, termed SC106879, designed for Cu-catalyzed conjugation with an azide group present in a rhodamine derivative (Fig. 3d). While SC106879 exhibited lower in vitro activity against wild type 3D7 ($IC_{50}$ value of 74 ± 9 nM) than SC83288, it effectively differentiated between wild type and SC83288-resistant parasites, with E4 showing an $IC_{50}$ value of >1 μM (Supplementary Fig. 2).

Synchronized cultures of 3D7 and E4 were exposed to 10 μM SC106879 during the trophozoite stage for 45 min, followed by fixation with paraformaldehyde and permeabilization with NP40 (Fig. 3e). Subsequently, the rhodamine-azide derivative and $Cu^{2+}$ were added to initiate the click-reaction. Confocal fluorescence microscopy revealed a distinct signal in E4, within an intracellular compartment co-localizing with GFP-tagged BiP, a known marker of the parasite's ER[17] (Fig. 3f, g). In contrast, no fluorescence signals were observed in 3D7 exposed to SC106879 or in untreated cells (Fig. 3f). A quantitative analysis confirmed a statistically significant increase in the ER-associated fluorescence signal intensity in SC106879-treated versus untreated E4 cells and versus treated or untreated 3D7 wild type parasites (Fig. 3h) ($p = 0.004$, Kruskal–Wallis one-way ANOVA on ranks versus 3D7 untreated control). These findings suggest that the muta-ted PfATP6 mediates transport of SC83288 into the parasite's ER to sequester the compound in a compartment away from its molecular target.

### The F972Y mutation affects PfATP6's transport dynamics

To better understand the structure–function relationship between SC83288 and the resistance-associated F972Y mutation, we performed molecular docking and molecular dynamics (MD) simulations. As no experimental structures are available for PfATP6, we generated homology models based on the sarco/endoplasmic reticulum $Ca^{2+}$-ATPase (SERCA1a) from *Oryctolagus cuniculus*[18]. Structural models were created for four transport cycle intermediates: E1 (cytoplasmic-open, $Ca^{2+}$-binding), E1 ~ P·ADP (phosphorylated, ADP-bound transitional state), E2 ~ P (phosphorylated, luminal-open, $Ca^{2+}$-release state), and E2 (dephosphorylated, luminal-open resetting state) (model quality summarized in Supplementary Table 1).

Docking was conducted using AutoDock Vina, Glide, and Boltz-2. Among them, AutoDock Vina produced the most favorable docking scores (below −9 kcal·mol⁻¹) and solvent-accessible surface area

(SASA) values for E1 and E1 ~ P·ADP, whereas Boltz-2 yielded the best results for E2 ~ P and E2 (Supplementary Tables 2, 3). SASA quantifies the degree of ligand exposure. During active transport, a low SASA value indicates that the ligand is well-enclosed by the protein, consistent with substrate occlusion. Notably, AutoDock Vina and Glide treat the protein as rigid, whereas Boltz-2 allows for conformational flexibility upon ligand binding.

The docking experiments predicted that both the wild-type and the F972Y mutant of PfATP6 are capable of binding SC83288 in all four modeled conformations, with no discernable differences between the two PfATP6 variants (Fig. 4 and Supplementary Fig. 3). In particular, the following residues engaged in SC83288 binding: K50, E115, and R1034 in the E1 state (Fig. 4a); K260, L1040, and D1038 in the E1 ~ P state (Fig. 4c); K51, D63, and L343 in the E2 ~ P state (Fig. 4c); and E316 and L1009 in the E2 state (Fig. 4c) (see Supplementary Fig. 3 for wild type PfATP6). 300 ns MD simulations supported these conclusions by demonstrating that these interactions are stable throughout the trajectories (Supplementary Figs. 4–6a–d). Importantly, in none of the docking or MD analyses did SC83288 directly interact with residue Y972 or F972 (Fig. 4b and Supplementary Fig. 6e, f), except for one run performed with AutoDock Vina on the E1 state of PfATP6 wild type where distances below the maximal possible binding distance of ~ 6 Å (for a salt bridge) were transiently observed (Fig. 4b). This suggests that F972Y contributes not to direct ligand binding but rather to conformational transitions and substrate transport across the membrane. This interpretation is consistent with experimental transport assays showing that both wild-type and mutant PfATP6 vesicles can handle SC83288, with the F972Y variant exhibiting enhanced transport efficiency (see Fig. 2c for comparison).

### SC83288 prevents DNA replication and arrests karyokinesis

We next turned to elucidating SC83288's mode of action. We initially assessed the cytological effects of SC83288 on parasite development, by exposing highly synchronized blood cultures of the multi-drug resistant *P. falciparum* strain Dd2 to 30 nM of SC83288 (10 × $IC_{50}$) at different time points across its 48 h replicative cycle and monitored the DNA content, using flow cytometry and SYBRGreen staining, in intervals of 4 h following the addition of the drug. In parallel assays, we investigated untreated cultures and cultures treated with the antifolate drug pyrimethamine (200 μM). The resulting DNA content was then expressed relative to the haploid genome in terms of a C-value and analyzed as a function of the time post-invasion. In untreated controls, the C-value remained constant during ring stage development until it increased around 28 ± 2 h post-invasion (hpi) as DNA replication commenced with the onset of the S/M-like phase[19,20] (Fig. 5a). The C-value peaked around 40 ± 2 hpi, before it fell owing to the rupture of infected red blood cells and the release of the daughter merozoites (Fig. 5a). In comparison, the C-value remained close to 1 throughout the replicative cycle in parasites exposed to SC83288 6 hpi or 24 hpi (Fig. 5a), indicative of a block in DNA replication. When added to tro-phozoites (24–32 hpi) or schizonts (36 ± 2 hpi) some degree of DNA replication occurred but to a lesser extent than in untreated controls (Fig. 5a).

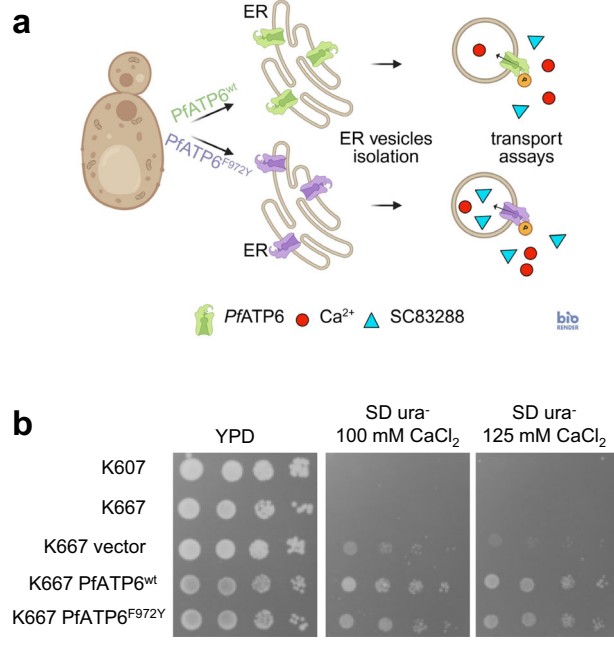

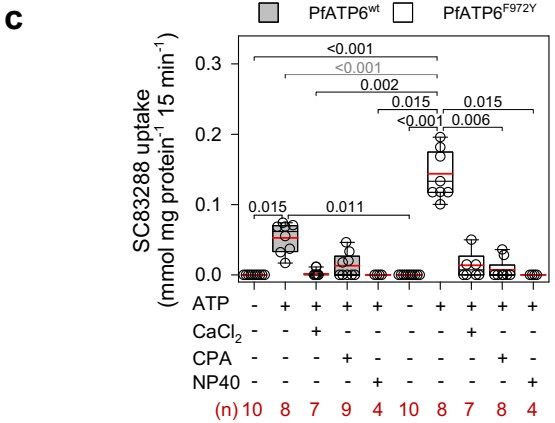

**c**

**Fig. 2 | PfATP6 mediates transport of SC83288. a** Schematic illustration of the experimental strategy to test SC83288 transport by PfATP6. The *Saccharomyces cerevisiae* K667 strain was complemented with either wild-type PfATP6 or the mutant variant PfATP6$^{F972Y}$. Endoplasmic reticulum–derived vesicles containing PfATP6 were isolated and tested for uptake of [$^3$H]-SC83288. Generated in Biorender: https://BioRender.com/knyyacx. **b** Complementation of the Ca$^{2+}$-sensitive yeast strain K667 by wild type PfATP6 and PfATP6$^{F972Y}$. Growth was assessed on rich medium (YPD) and synthetic defined (SD) medium lacking uracil (ura$^-$) and supplemented with 100 mM or 125 mM CaCl$_2$. The uracil-auxotrophic, Ca$^{2+}$-resilient strain K607 served as control. **c** Uptake of [$^3$H]-SC83288 into yeast vesicles containing PfATP6 variants. Vesicles were incubated for 15 min at 30 °C in buffer containing 100 nM [$^3$H]-SC83288 in the presence or absence of ATP (1 mM), CaCl$_2$ (10 µM), cyclopiazonic acid (CPA; 30 µM), or NP-40 (0.1%). Each point represents an independent biological replicate ($n$ = number of replicates); box plots show median (black line), mean (red line) and interquartile range. The error bars above and below the box indicate the 90th and 10th percentiles. Statistical significance determined by Kruskal–Wallis ANOVA on ranks with Dunn's post hoc test (black; two-sided; Bonferroni adjusted) and a targeted two-sided $t$-test for conditions 2 vs 7 (gray). $p$-values are shown. Source data are provided as a Source Data file.

We repeated the experiment using the E4 mutant and its parental 3D7 line. Treatment of the mutant starting at 20 hpi had no effect on life cycle progression; similar to untreated 3D7, the C-value increased steadily, peaking around 44 hpi before declining due to parasite egress (Fig. 5b). In contrast, SC83288-treated 3D7 parasites failed to replicate

their DNA, with the C-value remaining at 1 throughout the observation period (Fig. 5b).

Morphological analysis of Giemsa-stained blood smears from cells exposed to 30 nM SC83288 starting at 8–12 hpi or 30–34 hpi and continuing until the end of the cycle revealed pyknotic forms, characterized by nuclear condensation and a progressive reduction in cell volume, a phenotype not observed in untreated, time-matched controls, which progressed normally through the replicative cycle (Fig. 5c, d). When segmenters (42–46 hpi), a stage post-karyokinesis, were exposed to the drug, no effects were observed; the parasites egressed and reinvaded normally (Fig. 5e). Apparently, cytokinesis, which occurs 42–46 hpi, and egress are not affected by SC83288. These findings suggest that SC83288 specifically prevents DNA replication and arrests karyokinesis, ultimately leading to parasite death.

Pyrimethamine, when administered during early ring stages, also effectively inhibited DNA replication (Fig. 5a), aligning with its mode of action by impeding biosynthesis of dTMP through the inhibition of the parasite's dihydrofolate reductase (DHFR)[21]. Pyrimethamine is usually combined with sulfadoxine, a sulfonamide that also targets the dihydrofolate biosynthesis pathway by inhibiting the parasite's dihydropteroate synthase (DHPS)[21]. The combination of pyrimethamine and sulfadoxine acts synergistically. Since SC83288 shares the sulfonamide moiety with sulfadoxine (Supplementary Fig. 7) and given that its phenotypic effects on DNA replication are comparable to that of pyrimethamine, we speculated that SC83288 might also interfere with pyrimidine synthesis. However, SC83288 interacted antagonistically with pyrimethamine (mean fractional half-maximal inhibitory concentration (meanFIC$_{50}$) = 1.5 ± 0.3), as shown by isobologram analysis (Supplementary Fig. 7). Furthermore, SC83288 failed to demonstrate any appreciable inhibitory activity against the parasite's DHFR, DHPS or hydroxymethyl dihydropteridine pyrophosphokinase (HPPK) in in vitro enzymatic assays (Supplementary Tables 4, 5). HPPK forms a bifunctional enzyme together with DHPS in the parasite. We also explored the possibility of SC83288 blocking DNA replication by intercalating into DNA, but found no supporting evidence (Supplementary Fig. 8).

## SC83288 alters gene expression of nuclear components

To elucidate which aspects of Plasmodium physiology are perturbed by SC83288, we next employed a combination of untargeted transcriptomic, metabolomic, and cellular thermal proteomic (CETSA) approaches. Comparative RNA-seq was performed on trophozoite-stage 3D7 parasites (25 ± 3 hpi) exposed to 0 nM, 5 nM, or 10 nM SC83288 for 4.5 h, a stage chosen for its pronounced susceptibility to SC83288 and hence optimal responsiveness at the transcriptional level. Each condition was analyzed in three biologically independent experiments to ensure reproducibility and statistical robustness.

To ensure stringent RNAseq data quality, low-quality samples and transcripts were excluded, including those with low read counts, rRNA and var gene transcripts, and genes not expressed across any condition. The remaining dataset was normalized and analyzed for differential expression using a generalized linear model with quasi-likelihood (QL) fitting, revealing transcriptomic changes induced by SC83288 treatment.

We found that SC83288 significantly upregulated 9 out of the 3941 unique transcripts detected by RNAseq, with two transcripts being revealed under both conditions ($p < 0.05$, -log$_{10}$(FDR) > 1.3, fold change > 1.5) (Fig. 6). Gene ontology analysis of cellular components revealed that 6 of these 9 transcripts encode proteins with nuclear or partially nuclear localization, associated with chromatin remodeling, transcriptional regulation, or cell division. These include:

- PF3D7_0925700.1 (histone deacetylase 1): involved in chromatin remodeling and transcriptional silencing via histone deacetylation,

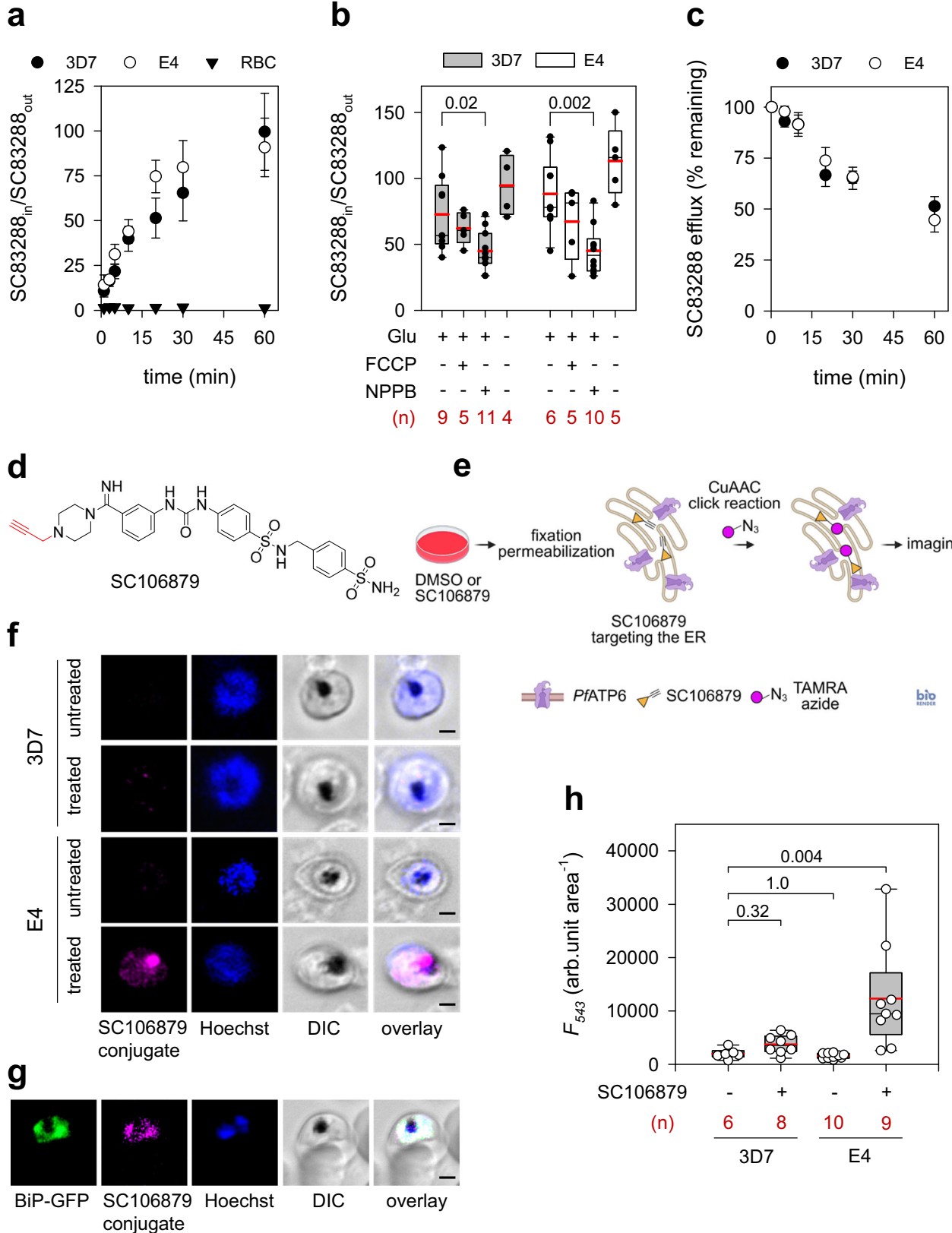

- PF3D7_1362200.1 (RuvB-like helicase 3): linked to chromatin remodeling and transcription regulation,
- PF3D7_1468700.1 (eIF4A): translation initiation factor with potential indirect effects on transcription,
- PF3D7_0605300.1 (aurora kinase ARK1): regulator of mitosis, spindle assembly, and karyokinesis,

- PF3D7_0903700.1 (alpha tubulin 1): a structural component of the microtubule cytoskeleton and mitotic hemispindle and centriolar plaque,
- PF3D7_0821800.1 (SEC61 subunit beta): part of the ER translocon, involved in co-translational protein translocation.

**Fig. 3 | SC83288 flux and subcellular localization. a** Time-course of SC83288 uptake from an external concentration of 50 nM [³H]-SC83288 in wild-type 3D7 parasites (black circles), resistant line E4 (white circles), and uninfected erythrocytes (black inverted triangles). Uptake is expressed as the ratio of intracellular to extracellular drug concentration (SC83288_in/SC83288_out). Data represent mean ± SEM from five independent biological replicates. **b** [³H]-SC83288 accumulation in 3D7 and E4 parasites under indicated conditions: with or without glucose (10 mM glucose or 2-deoxyglucose), protonophore FCCP (10 μM), or NPP inhibitor NPPB (5 μM). Each point represents an independent biological replicate (*n* = number of replicates). Box plots show the interquartile range with median (black line) and mean (red line); whiskers denote the 10th and 90th percentiles. Statistical significance was determined by Holm–Šidák one-way ANOVA with multiple comparison versus the respective glucose-only control. *p* values are shown. **c** Time-course of SC83288 efflux from 3D7 (black circles) and E4 (white circles) parasites preloaded with 50 nM [³H]-SC83288 for 15 min. Data represent mean ± SEM from four independent biological replicates. **d** Chemical structure of the clickable SC83288 analog SC106879. **e** Experimental workflow for subcellular localization of SC106879 using in-cell click chemistry (Biorender: https://BioRender.com/oenavbl). **f** Representative confocal fluorescence images showing intracellular distribution of SC106879. Images derive from three independent biological experiments. Parasites were cultured with or without 10 μM SC106879 for 45 min, processed, and labeled by click reaction with 2.5 mM TAMRA–azide and Cu(I). Scale bar, 2 μm. **g** In-cell click labeling of SC106879 in E4 parasites expressing the ER marker BiP-GFP. Images derive from three independent biological experiments (10 cells). Scale bar, 2 μm. **h** Quantification of SC106879-derived fluorescence intensity (arbitrary units per area) after background subtraction. Box plots as in (**b**). Statistical significance was determined by Kruskal–Wallis one-way ANOVA on ranks with multiple comparisons versus the 3D7 untreated control group. *p*-values and the number of independent biological determinations are shown. source data for (**a**–**c**, **h**), are provided as a Source Data file.

The remaining transcripts encode proteins localized to the ER, cytoplasm, or digestive vacuole and are involved in cellular stress responses and metabolism:

- PF3D7_1032500.1 (DER1-like protein): component of the ER-associated degradation (ERAD) pathway and unfolded protein response (UPR),
- PF3D7_1235600.1 (serine hydroxymethyltransferase): key enzyme in folate metabolism, supplying one-carbon units for nucleotide synthesis and methylation,
- PF3D7_1407800.1 (plasmepsin IV): aspartic protease involved in hemoglobin degradation within the food vacuole.

This transcriptional signature suggests that SC83288 disrupts epigenetic regulation and karyokinesis, triggering stress responses likely due to accumulation of misfolded proteins or impaired protein trafficking, alongside compensatory metabolic and epigenetic adaptations, particularly in methylation and chromatin-associated pathways.

## Perturbations to cellular methylation

In parallel, we conducted untargeted and unbiased global metabolomics analysis to identify changes in metabolic pathways associated with SC83288 treatment. Young trophozoites ($20 \pm 2$ hpi) of 3D7 and E4 were exposed to 40 nM SC83288, or an equivalent volume of DMSO as a negative control, for 12 h. Each experimental condition was reproduced in eight independent biological samples. A total of 781 metabolite-indicating signals were identified by LC-MS. Although additional metabolites were predicted to shift upon SC83288 exposure, only a subset—including 5′-methylthioadenosine (MTA), adenine, phosphoethanolamine, and the ratio of S-adenosyl-L-methionine over S-adenosylhomocysteine (SAM/SAH)—could be confirmed through orthogonal validation, beyond mass-based annotation. Of these, MTA and adenine increased in treated 3D7 lines, while phosphoethanolamine decreased in abundance (Fig. 7a). These changes were characterized by a LOD score exceeding 1.08, a *p*-value below 0.05, and a fold change greater than 2 in either direction. In comparison, no metabolic disturbances were induced by SC83288 in the PfATP6 mutant line E4 (Fig. 7b).

MTA is derived from SAM (AdoMet) (Fig. 7c), which is a key methyl donor in numerous biochemical reactions and is hydrolyzed to adenosine and homocysteine via SAH. SAM is tightly regulated in cellular systems[22]. In SC83288 treated 3D7 lines, the ratio of SAM to SAH, commonly referred to as methylation index, increased by 3.9-fold (*p* = 0.02, two-sided Mann–Whitney Rank Sum Test), indicating a disruption in SAM homeostasis and an accumulation of this key metabolite (Fig. 7a). Possible impacts on cellular methylation in response to drug are thus predicted, consistent with the RNAseq results.

If SC83288 indeed inhibits a SAM-consuming enzyme, such as a methyl transferase, one would expect at least partial phenotypic rescue under conditions of excess SAM. To test this hypothesis, we performed chemical rescue experiments by supplementing the culture medium with increasing concentrations of SAM (0, 0.5, 1.0, and 2 mM) and measuring parasite susceptibility to SC83288 using the standard growth inhibition assay. We observed a significant, concentration-dependent increase in the SC83288 $IC_{50}$ value, from $8.3 \pm 0.8$ nM in the absence of SAM to $53.5 \pm 4.3$ nM at 2.0 mM SAM ($p < 0.001$, Holm–Sidak one-way ANOVA) (Fig. 8a and Supplementary Fig. 9). Similarly, supplementation with 2.0 mM betaine, a methyl donor in the methionine cycle (Fig. 7c), also reduced parasite sensitivity to SC83288 ($IC_{50}$ value = $17.8 \pm 1.8$; $p < 0.001$, Holm–Sidak one-way ANOVA) (Fig. 8a and Supplementary Fig. 9). In contrast, supplementation with 0.5 mM SAH or MTA, both known inhibitors or by-products of methylation reactions[23,24], or 1.0 mM of hypoxanthine had no significant effect on SC83288 responsiveness (Fig. 8a and Supplementary Fig. 9). Although the functional available of these metabolites is suggested by our rescue experiments, facilitated uptake by infected and/or uninfected erythrocytes has thus for only been reported for SAM, betaine, MTA, and hypoxanthine[25–27]. It remains possible that these and other metabolites, including SAH, also gain access via NPP/PSAC[16].

Given the observed effect of SAM on SC83288 responsiveness, along with the drug's impact on DNA replication and karyokinesis, we next examined the impact of SC83288 on both RNA adenosine (m⁶A) methylation and DNA cytosine (m⁵C) methylation. Using a quantitative colorimetric assay, we detected a 45% reduction in RNA m⁶A methylation in RNA isolated from trophozoites treated with 80 nM SC83288 for 4 h, compared to untreated controls ($p < 0.001$, two-sided Mann–Whitney rank rum test) (Fig. 8b). Likewise, total genomic DNA isolated under the same conditions exhibited a 60% decrease in DNA m⁵C methylation, as determined by a quantitative ELISA-based DNA methylation assay ($p < 0.001$, two-sided Welch's *t*-test) (Fig. 8c). Furthermore, nuclear extracts from SC83288-treated trophozoites showed an 80% reduction in DNA methyltransferase activity in a fluorimetric enzyme assay, relative to untreated controls (Fig. 8d). This inhibitory effect was reversible upon supplementation with 2.0 mM SAM (Fig. 8d), suggesting functional competition at the enzymatic level.

Given these findings, we investigated whether SC83288 targets the parasite's DNA methyltransferase, PfDNMT2 (PF3D7_0727300)[28]. We expressed and purified recombinant PfDNMT2 and established a functional enzymatic assay[28]. SC83288 inhibited PfDNMT2 activity in a concentration-dependent manner, with an $IC_{50}$ value of 7 μM, consistent with Michaelis Menten kinetics (Fig. 8e). Although this value is substantially higher than the $IC_{50}$ for parasite killing, the measurements described above have shown that SC83288 accumulates intracellularly, potentially reaching concentrations sufficient to inhibit PfDNMT2 in vivo. To further validate target engagement, we tested SC83288 sensitivity in parasite lines with either PfDNMT2 knock-down or ~12-fold overexpression[28], using the 3D7 parental line as a control.

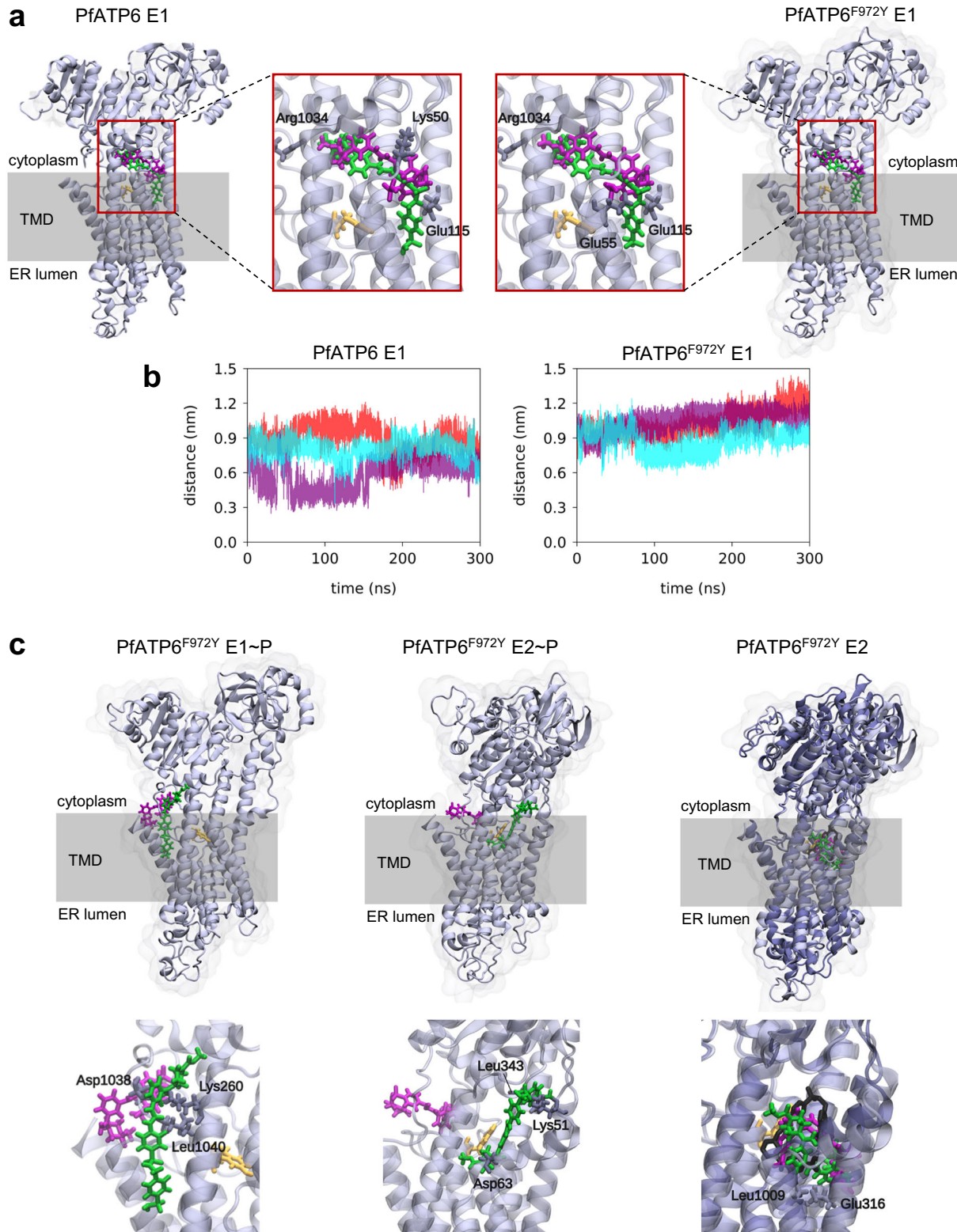

**Fig. 4 | SC83288 binding to PfATP6. a** Molecular docking of SC83288 to PfATP6 in the E1 conformation (cytoplasmic-open, Ca²⁺-binding state). Docking poses predicted by AutoDock Vina (purple) and Glide (green) are shown for wild-type PfATP6 (left) and the PfATP6^F972Y mutant (right). Residues Phe972 and Tyr972 are highlighted in yellow. The PfATP6 structure is shown in light blue. **b** Minimal distance between bound SC83288 and any atom of residue 972 (Phe/Tyr) during 300 ns molecular dynamics (MD) simulations in the E1 state. **c** Predicted binding poses of SC83288 to the PfATP6^F972Y mutant in three additional transport conformations: E1–P (left, phosphorylated, ADP-bound transitional state), E2–P (middle, phosphorylated, luminal-open, Ca²⁺-release state), and E2 (right, dephosphorylated, luminal-open resetting state) (see Supplementary Fig. 3 for data on wild type PfATP6). Binding poses from Boltz2 are shown in black; PfATP6 backbones are shown in dark blue. Color scheme as in (**a**).

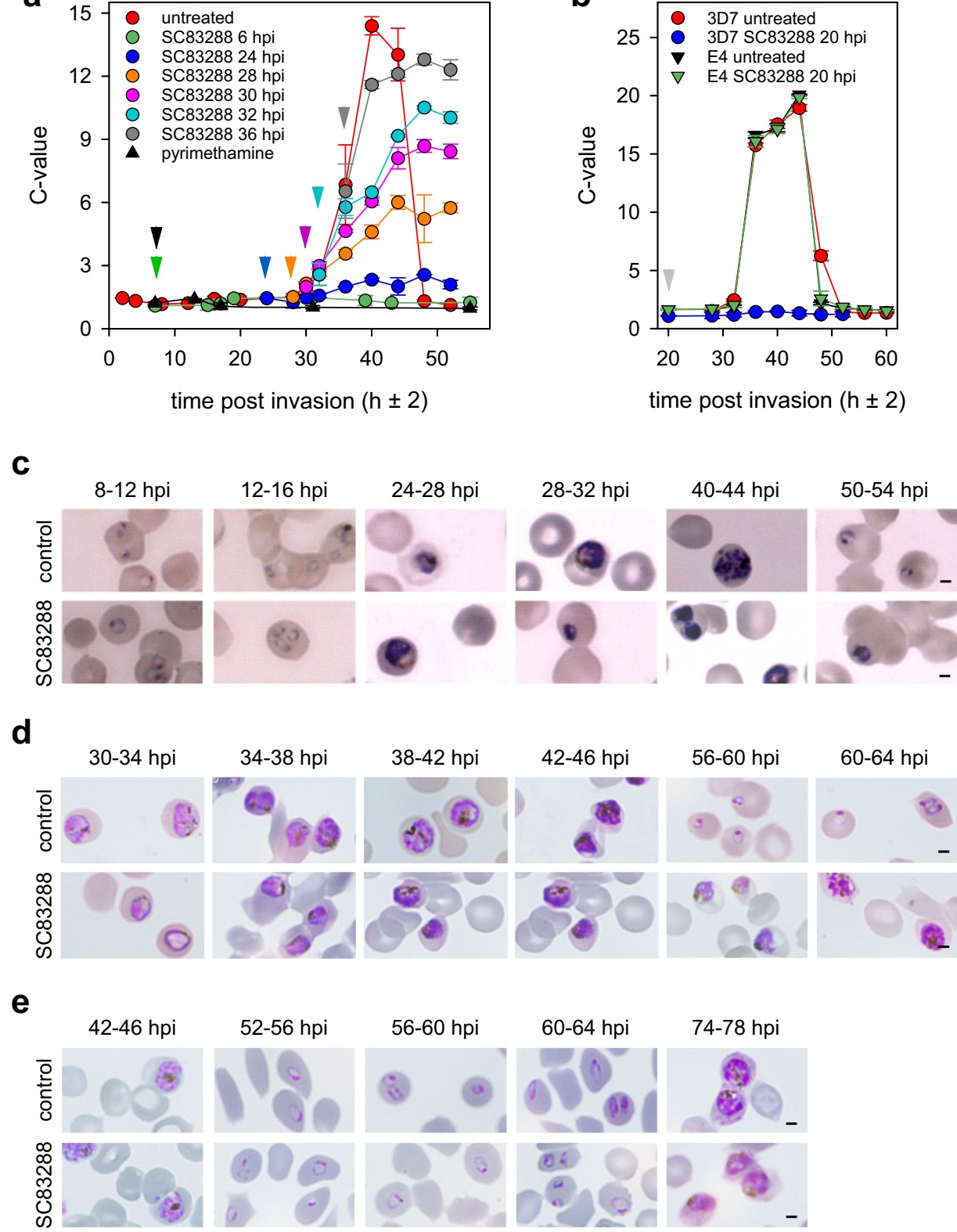

**Fig. 5 | SC83288 inhibits DNA replication and arrests karyokinesis. a** Inhibition of DNA replication by SC83288. *P. falciparum* Dd2–infected erythrocytes were treated with 30 nM SC83288 or 200 μM pyrimethamine at the indicated time points post-invasion (arrowheads). DNA content was quantified by flow cytometry using SYBR Green staining at 4-hour intervals post-treatment and expressed as C-values relative to the haploid genome. Data represent mean ± SEM from 5 independent biological replicates. **b** DNA replication kinetics of wild-type 3D7 and the SC83288-resistant line E4 under identical conditions as in (**a**). The gray arrow head indicates the time point 30 nM SC83288 was added. Data represent means ± SEM from three independent biological replicates. Giemsa-stained smears of highly synchronized 3D7 cultures exposed to 30 nM SC83288 starting at 8–12 hpi (**c**), 30–34 hpi (**d**), or 42–46 hpi (**e**). Untreated, time-matched controls were analyzed in parallel. Images are representative of four independent biological replicates with 30 cells analyzed per condition. Note the appearance of pyknotic parasites with condensed nuclei in the 8–12 hpi and 30–34 hpi treatment groups, but not in the 42–46 hpi group. Bar, 2 μm. For subfigures a and b, source data are provided as a Source Data file.

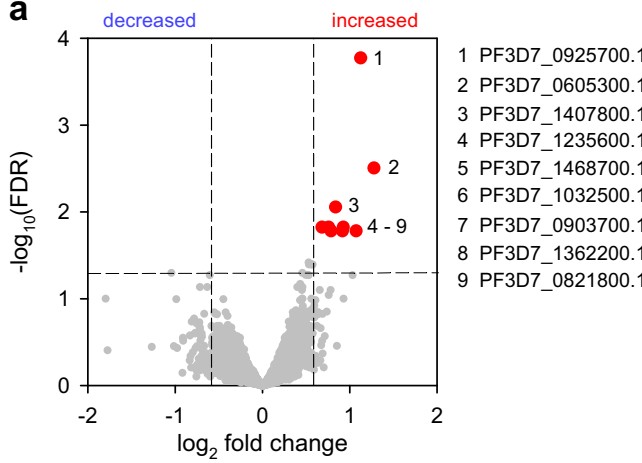

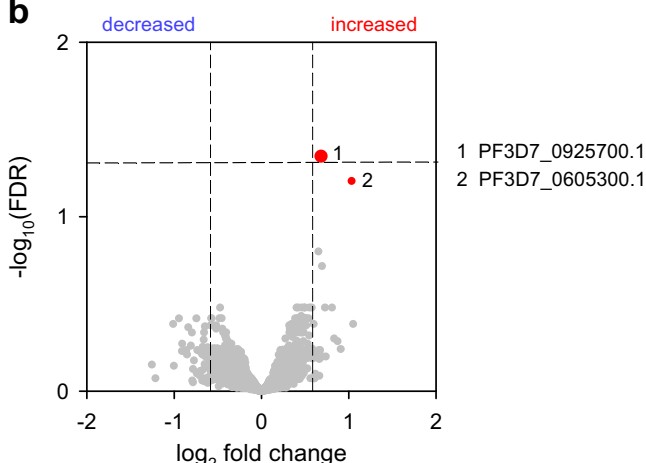

**Fig. 6 | Comparative transcriptomic analysis of SC83288-treated parasites.** Volcano plots showing differentially expressed genes in trophozoites ($25 \pm 3$ h post-infection) treated with 5 nM (**a**) or 10 nM (**b**) SC83288 for 4.5 h, compared with untreated controls. The x-axis represents $\log_2$ fold change and the y-axis shows $-\log_{10}$ false discovery rate (FDR)-adjusted $p$-values. Differential expression was assessed using two-sided quasi-likelihood tests implemented in edgeR, with p-values corrected for multiple comparisons using the Benjamini–Hochberg procedure. Red dots indicate significantly upregulated genes ($p < 0.05$, $-\log_{10}(FDR) > 1.3$, and fold change > 1.5). Data represent three independent biological replicates per treatment condition. Source data are provided as a Source Data file.

While the parental and knock-down lines exhibited similar high sensitivity ($IC_{50}$ values of $2.5 \pm 0.3$ (3) and $4.0 \pm 0.4$ (3), respectively; $p > 0.05$, two-sided Student's $t$ test), the overexpression line displayed significantly reduced sensitivity, indicative of a protective outcompetition effect (Fig. 8f). These data strongly support PfDNMT2 as a critical molecular target of SC83288, with downstream consequences for epigenetic regulation, DNA replication, and karyokinesis. In addition to PfDNMT2, SC83288 may also affect PfMT-A70 (PF3D7_0729500), PfMT-A70.2 (PF3D7_1235500), or its regulatory subunit WTAP (PF3D7_1230800), which together constitute the parasite's principal RNA $N^6$-methyladenosine (m6A) methyltransferase complex[29,30].

### Cellular thermal shift assay reveals additional target candidates

We next employed cellular thermal shift assays in conjunction with mass spectrometry, specifically utilizing the isothermal dose–response variant (ITDR-CETSA), to identify possible molecular targets of SC83288[31,32]. Enriched infected erythrocytes at the trophozoite stage ($25 \pm 3$ hpi) were treated with varying concentrations of SC83288 (7.7 pM, 38.4 pM, 192.0 pM, 969.6 pM, 4.8 nM, 24.0 nM, 120.0 nM, $0.6\,\mu M$, and $3.0\,\mu M$), over a duration of 1 h. Subsequently, sets of exposed samples underwent specific thermal challenges at temperatures of either 37 °C, 50 °C, 55 °C, or 60 °C. A control group treated with DMSO was analyzed in parallel. We then analyzed the resulting shifts in protein thermostability through LC-MS/MS examination.

Changes in a protein's thermostability can result from a direct interaction with SC83288 or a downstream event affecting the protein's conformation, such as posttranslational modifications, complex formation, or the binding of physiological ligands. The thermal responses to the drug were automatically curve fitted, yielding several quantitative parameters, including the $R^2$ value, the area under the curve (AUC), the minimum dose threshold (the lowest effective drug concentration), and the fold protein stability change against the non-denatured control at 37 °C.

In total, we obtained 7366 thermal drug–response profiles, encompassing 1601 proteins from *P. falciparum* and 864 erythrocyte proteins (Supplementary Data Files 1, 2). To identify proteins potentially stabilized or destabilized by SC83288 treatment, we used the following criteria: an $R^2$ value greater than 0.8 and either an AUC differing from the median by two times the median absolute deviation (MAD) or a 30% change in protein stability in the presence of SC83288[33]. None of the red blood cell proteins identified met these criteria (Supplementary Data File 2). In contrast, applying the cutoff values to the *P. falciparum* dataset led to the identification of 21 hits meeting criteria 1 (Fig. 9a) and 13 hits meeting criteria 2 (Fig. 9b), with 6 proteins satisfying both conditions (Supplementary Table 6 and Supplementary Data File 3). We then filtered out hits with low MS spectral confidence, defined as those with fewer than four peptide spectrum matches (PSMs) or fewer than four unique peptide matches (sumUniPeps), resulting in the exclusion of 14 hits (Supplementary Table 6 and Supplementary Data File 3). The remaining candidates underwent visual inspection of their thermal drug–response profiles for plausibility, ultimately yielding four plausible candidates (Fig. 9c). This includes serine repeat antigen 5 (SERA5) and S-adenosylhomocysteine hydrolase (PfSAHH), which were stabilized in the presence of SC83288, and fumarate hydratase and isocitrate dehydrogenase, which were destabilized by the drug treatment (Fig. 9c).

Given the impacts on methylation noted above, we also searched the ITDR-CETSA database for methyltransferases. We identified 28 of the 46 PlasmoDB-annotated methyltransferases, but none exhibited thermal shifts in response to SC83288 (Supplementary Data File 1). Notably, PfDNMT2, PfMT-A70, PfMT-A70.2, and the PfMT-A70 subunit WTAP (PF3D7_1230800) were not detected by ITDR-CETSA.

### SERA5, fumarate hydratase, isocitrate dehydrogenase and PfSAHH are unlikely direct targets of SC83288

SERA5 belongs to a family of nine closely related proteins, most of which are cysteine proteases. SERA5 itself is catalytically inactive and appears to function as a structural or regulatory component during merozoite egress, requiring sequential proteolytic processing for activation[34,35]. In silico docking analyses, using the docking programs Autodock Vina and Glide and the crystallographic structure of the SERA5 pseudo-zymogen[36], revealed only surface-associated binding poses with partly unfavorable docking scores (Supplementary Fig. 10), consistent with weak or transient interactions unlikely to be biologically relevant.

Likewise, computational docking studies performed on fumarate hydratase and isocitrate dehydrogenase, two enzymes of the mitochondrial tricarboxylic acid (TCA) cycle, revealed only weak or negligible surface binding of SC83288, with also unfavorable docking scores, which we considered unspecific (Supplementary Fig. 10 and Supplementary Table 7; for model quality see Supplementary Table 1).

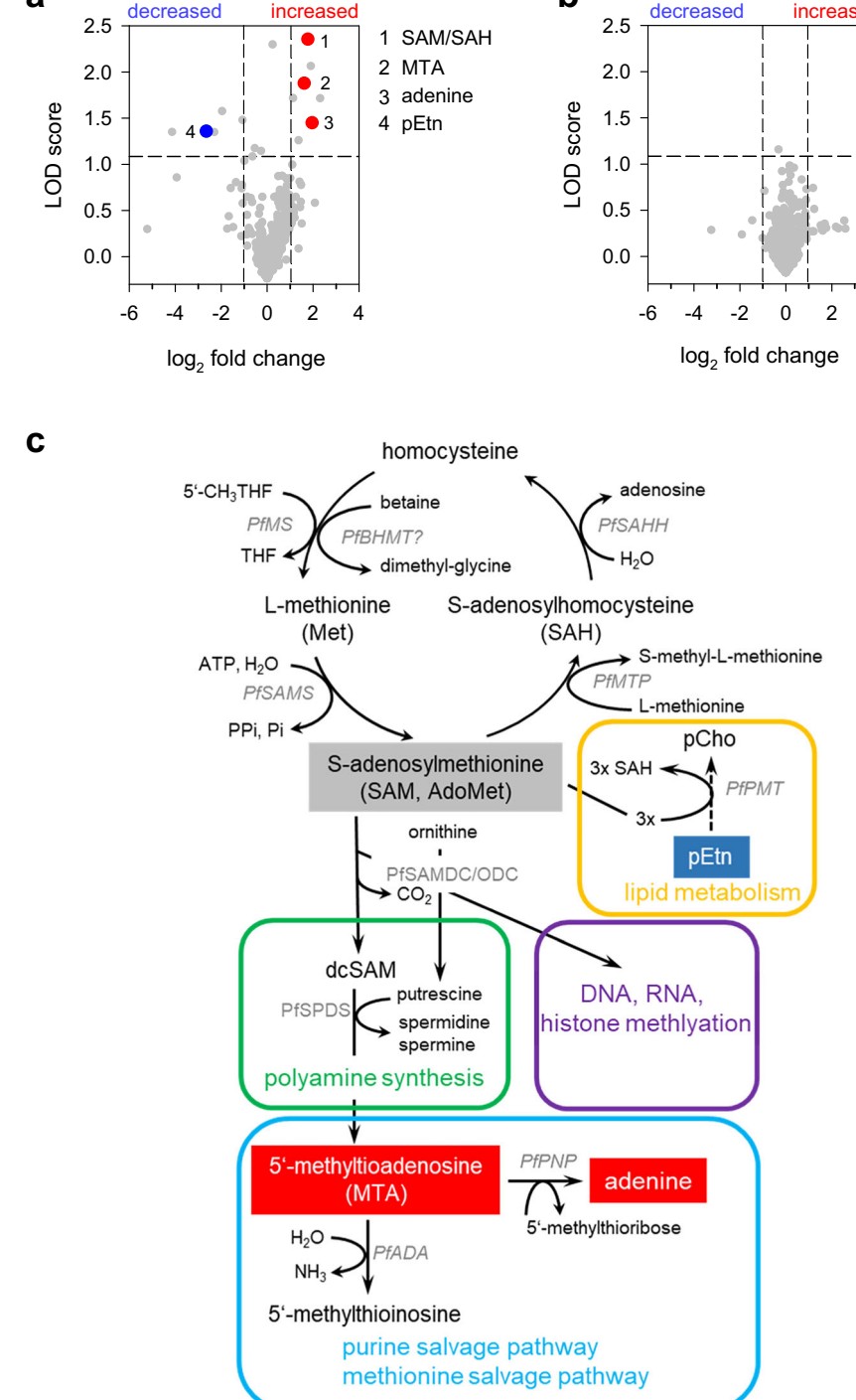

**Fig. 7 | Comparative metabolomic profiling of SC83288-treated parasites.**
**a**, **b** Volcano plots showing the statistical significance (LOD-score) versus the magnitude of metabolite changes (fold change) between SC83288-treated (40 nM, 4 h) and untreated controls in wild-type (**a**) and SC83288-resistant (E4) parasites (**b**). Infected erythrocytes were analyzed at the trophozoite stage (20 ± 2 h post-infection). Each dot represents an individual metabolite, with the exception of the SAM/SAH data point that represents the mean ratio of paired S-adenosyl-L-methionine (SAM) to S-adenosylhomocysteine (SAH) values. Statistical significance was assessed using univariate Pearson correlation analysis comparing treated and control groups, with correlation coefficients converted to p-values and transformed into relative LOD scores defined as $-\log(p_i / \text{mean}(p))$ to rank metabolic perturbations across the datasets. Metabolites meeting significance criteria (LOD > 1.08, $p < 0.05$, and fold change > 2 in either direction) and verified in the organismal context are highlighted − red for increased and blue for decreased metabolites under SC83288 treatment.

Metabolites not meeting these criteria or not verified are shown in gray. Data represent eight independent biological replicates. Source data are provided as a Source Data file. MTA 5′-methylthioadenosine, SAM (AdoMet) S-adenosyl-L-methionine, SAH S-adenosylhomocysteine, pEtn phosphoethanolamine. **c** Proposed model of SAM (AdoMet) metabolism in *P. falciparum*. SAM acts as a precursor and methyl donor in several biochemical pathways. Metabolites increased and decreased upon SC83288 treatment are shown in red and blue, respectively. Relevant enzymes are labeled. pCho phosphocholine, SAH S-adenosylhomocysteine, dcSAM decarboxylated S-adenosylmethionine, THF tetrahydrofolate, PfADA adenosine deaminase, PfBHMT betaine–homocysteine S-methyltransferase (putative), PfPNP purine nucleoside phosphorylase, PfSPDS spermidine synthase, PfSAMDC/ODC bifunctional S-adenosylmethionine decarboxylase/ornithine decarboxylase, PfPMT phosphoethanolamine N-methyltransferase, PfSAMS S-adenosylmethionine synthetase, PfMS methionine synthase, PfSAHH S-adenosylhomocysteine hydrolase.

SAHH maintains cellular methylation capability by reversibly converting SAH to homocysteine and adenosine[37]. Because SAH is a potent product inhibitor of SAM-dependent methyltransferase, SAHH activity is tightly regulated. Most SAHH function as a homotetramer, with each monomer comprising a substrate/product-binding site and a nicotinamide adenine dinucleotide (NAD$^+$) cofactor-binding domain[37]. In *P. falciparum*, PfSAHH is essential for intraerythrocytic development, and direct enzymatic inhibition is lethal to the parasite[38,39].

Molecular docking and MD simulations using the available PfSAHH crystal structure[38], showed no meaningful binding of SC83288 to the catalytically competent, NAD$^+$-bound holo enzyme. Predicted poses were weak, surface-associated, and distant from the active site (Supplementary Fig. 11; Supplementary Table 7). Only under NAD$^+$-free conditions did SC83288 occupy the cofactor-binding pocket throughout the 300 ns MD simulations (Fig. 9d, Supplementary Fig. 11 and Supplementary Table 7). Such a scenario is unlikely to dominate in vivo. Consistent with the in silico findings, SC83288 failed to inhibit purified PfSAHH enzymatic activity in vitro at concentrations up to 500 µM (Supplementary Fig. 12). As PfSAHH expressed in *E. coli* is known to contain tightly bound NAD$^+$ and to be catalytically mature[38], these results strongly argue against PfSAHH being a direct enzymatic target of SC83288.

Collectively, the in silico analyses provide no evidence that SC83288 binds to, or forms stable biologically relevant complexes with, SERA5, fumarate hydratase, isocitrate dehydrogenase, or PfSAHH suggesting that the thermal stability changes observed in ITDR-CETSA are likely indirect effects rather than direct target interactions.

## Discussion

Our findings show that SC83288 disrupts blood-stage development of *P. falciparum* by inhibiting DNA replication and arresting karyokinesis (Fig. 5). We identify the parasite's DNA methyltransferase, PfDNMT2, as a primary molecular target (Fig. 8c–f), linking SC83288 activity to disrupted epigenetic regulation and transcriptional control. Resistance arises through mutations in the SERCA-type Ca$^{2+}$ pump PfATP6, which convert the transporter into a drug redistribution system, sequestering SC83288 into the endoplasmic reticulum and thereby away from its nuclear target(s).

We initially considered PfATP6 as a potential target of SC83288 due to the fact that PfATP6 is under strong selective pressure in the presence of the drug, with mutations and gene copy number amplifications found in resistant lines[8]. PfATP6, the parasite's orthologue of the mammalian sarco-endoplasmic Ca$^{2+}$ ATPase, is crucial for intracellular Ca$^{2+}$ homeostasis, pumping excess cytoplasmic Ca$^{2+}$ into the ER for storage[11,12]. Given that Ca$^{2+}$ triggers the transition from G1 to the S/M-like phase in *P. falciparum*[40], inhibiting PfATP6 by SC83288 appeared to be a plausible explanation for the compound's cytotoxic effects. However, while PfATP6 does interact with SC83288, this interaction is due to its being a substrate for the transporter rather than an inhibitor per se.

CRISPR/Cas9 genome editing confirmed that the PfATP6 F972Y mutation confers a high level of SC83288 resistance, shifting the IC$_{50}$ value by three orders of magnitude from $8 \pm 2$ nM in the wild-type parasite line 3D7 to $5 \pm 0.3$ µM in the mutant line E4 (Fig. 1c). Several non-synonymous single nucleotide polymorphisms have been described for PfATP6, including an L263E mutation that was once implicated in reducing artemisinin responsiveness[12,41]. Notably, the F972Y mutation has not been observed in field isolates and the chances of it establishing itself in the field appear low due to the heavy fitness costs associated with its impact on the Ca$^{2+}$ regulatory function of PfATP6 (Fig. 1f, g). However, prolonged sublethal drug exposure could potentially select compensatory mutations within PfATP6 or secondary factors, thereby restoring intracellular Ca$^{2+}$ homeostasis and cellular Ca$^{2+}$ responses.

Interestingly, both wild-type and PfATP6 mutant parasite lines exhibited comparable SC83288 accumulation and efflux kinetics overall (Fig. 3a, c). Typically, drug transporting systems, such as the multi-drug resistance transporter in cancer cells or PfCRT in *P. falciparum*[4], confer resistance by reducing the overall intracellular drug concentration below toxic levels. In comparison, mutated PfATP6 does not alter the overall intracellular SC83288 concentration, instead redirecting drug by pumping SC83288 into the parasite's ER (Fig. 3f–h). In vitro transport studies using reconstituted vesicles and in situ drug localization experiments using a fluorescently-labeled derivative of SC83288 confirmed this redistribution of drug (Figs. 2c and 3e–h). While both the wild type and F972Y mutant PfATP6 are capable of transporting SC83288, the mutant does so with increased efficiency (Fig. 2c). Molecular docking and MD simulations suggest that this increased transport efficiency is not due to altered drug binding, since no direct interactions of SC83288 with F972 or F972Y were observed during the simulated transport cycle in the wild type or mutant PfATP6 (Fig. 4 and Supplementary Figs. 3, 6e, f). Moreover, both variants exhibited comparable SC83288 binding poses in each of the four simulated transport states (Fig. 4 and Supplementary Fig. 3). Instead, the combined data suggest that the F972Y mutation plays a role in the transport cycle, making it more efficient. Our transport studies further revealed that SC83288 passes the plasma membrane of the infected erythrocyte via parasite-induced new permeation pathways (Fig. 3b).

Several independent lines of evidence converge on the nucleus as the primary site of action of SC83288, implicating nuclear methyltransferases as molecular targets. First, SC83288 arrests karyokinesis during the asexual blood stage cycle, leading to the appearance of pyknotic forms with condensed nuclei and reduced cell volumes (Fig. 5). Importantly, cytokinesis and parasite egress were not affected once karyokinesis had been completed. Second, karyokinesis arrest coincides with stalled DNA replication (Fig. 5). Third, global levels of m$^5$C DNA methylation were reduced by ~80% following SC83288 treatment (Fig. 8c, d), consistent with inhibition of nuclear methyl transferase pathways.

Unlike many eukaryotes, *P. falciparum* appears to possess only a single DNA methyl transferase, PfDNMT2, which also catalyzes methylation of cytosine 38 in tRNA$^{Asp}$ and is localized in both the nucleus and cytoplasm[28,42]. PfDNMT2 has been implicated in epigenetic gene regulation and epitranscriptomic control of translation[28,42], making it the most plausible primary molecular target of SC83288.

Evidence from other systems has shown that pharmacologic or genetic inhibition of DNA methyltransferases induces replication stress and cell-cycle arrest, often at G1/S or within S phase, via DNMT−DNA adduct−mediated fork stalling or hypomethylation-triggered checkpoint activation[43,44]. These precedents align with our findings that SC83288 diminishes DNA m$^5$C methylation and arrests DNA replication in *P. falciparum*. Supporting this model, in vitro enzymatic assays demonstrated direct inhibition of PfDNMT2 by SC83288, with an IC$_{50}$ value of 7 µM (Fig. 8e) - a concentration pharmacologically attainable given the parasite's ability to concentrate the drug. Moreover, a parasite line overexpressing PfDNMT2 by at least 12-fold displayed marked insensitivity to SC83288 (Fig. 8f). This finding suggests that high levels of PfDNMT2 can outcompete nuclear concentrations of SC83288, thereby overcoming the drug's parasiticidal activity. Similarly, excess SAM or alternative methyl donors, such as betaine, can at least partially overcome the inhibitory effect of SC83288 (Fig. 8a), consistent with competitive interference in SAM-dependent methyltransferase reactions. Taken together with the direct inhibition of PfDNMT2 in vitro and the resistance phenotype of PfDNMT2-overexpressing parasites, these results strongly support PfDNMT2 inhibition as a central mechanism of SC83288 action.

However, we cannot exclude the possibility that SC83288 targets other SAM-dependent methyltransferases. A partial knockdown of PfDNMT2 produced phenotypic changes, including increased susceptibility to external stress, prolonged intraerythrocytic development, and elevated merozoite number[28] - that only partly align with the

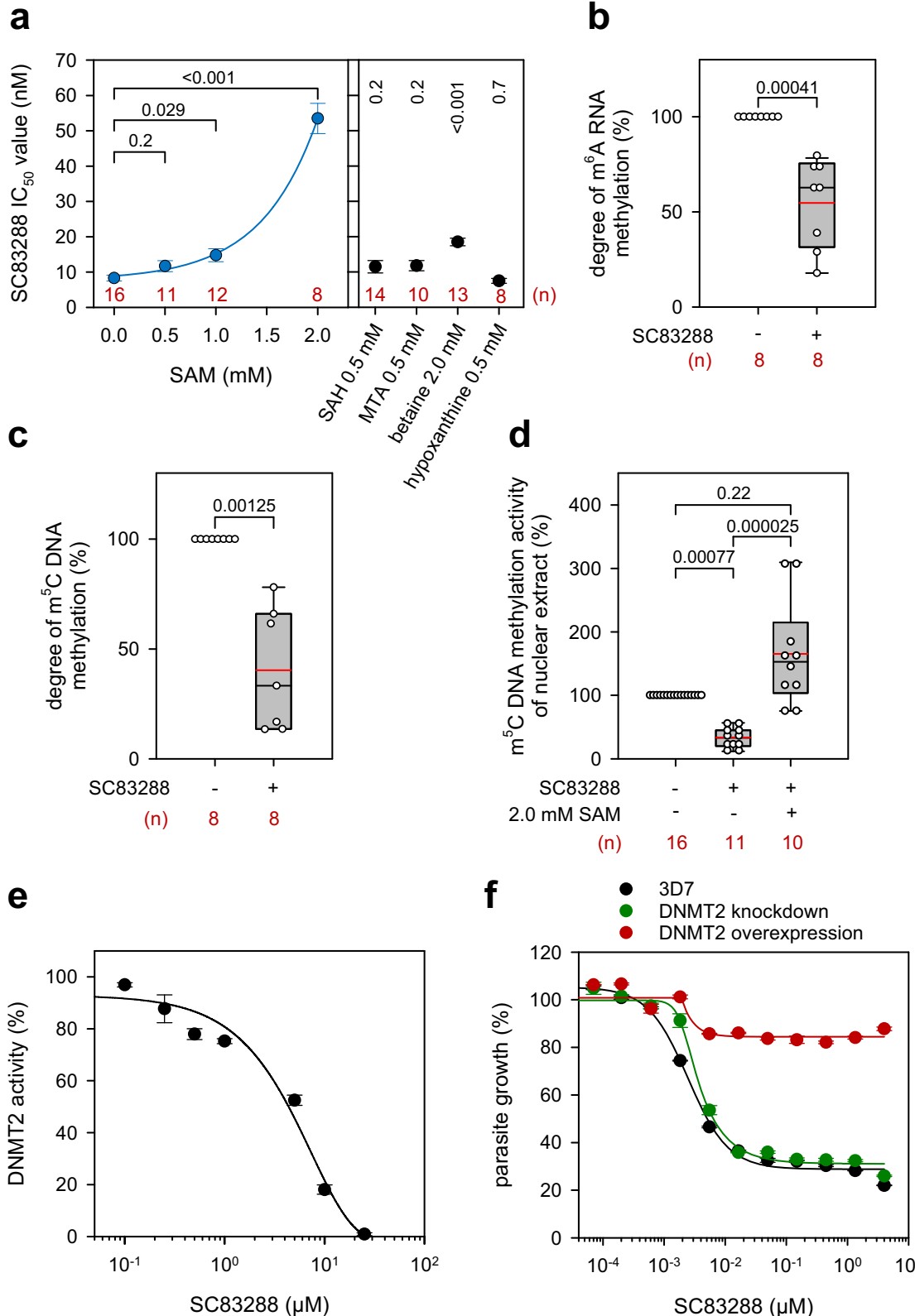

SC83288-induced effects, suggesting the involvement of additional pathways or enzymes.

The ~45% reduction in global m⁶A RNA methylation observed after SC83288 treatment (Fig. 8b), points to the RNA N⁶-methyladenosine (m⁶A) methyltransferase complex as a likely additional target. This observation raises the possibility that SC83288 may interfere with the complex or its subunits - either directly, by binding to the

methyltransferase or its cofactors, or indirectly, through perturbations of SAM metabolism or altered transcriptional regulation of the complex components. However, the precise mechanistic link between SC83288 exposure and m⁶A RNA methylation dysregulation remains to be elucidated.

Metabolomics and RNA-seq analyses provide further, though indirect, support for PfDNMT2 as a primary target of SC83288.

**Fig. 8 | PfDNMT2 as a primary target of SC83288. a** Chemical rescue experiments. Parasite cultures were supplemented with the indicated metabolites at the specified concentrations, and susceptibility to SC83288 was determined as $IC_{50}$ values using the standard growth inhibition assay. Data represent mean ± SEM from n independent biological replicates. Statistical significance was assessed for each panel by Holm–Šidák one-way ANOVA with multiple comparisons versus untreated control. The omnibus ANOVA yielded a significant effect ($F(3,43) = 90.59$, $p < 0.001$). **b** In vivo m⁶A RNA methylation levels in total RNA isolated from trophozoites treated with 80 nM SC83288 for 4 h, relative to untreated controls. Statistical evaluation, two-sided Mann–Whitney rank rum test. **c** In vivo m⁵C DNA methylation levels in total DNA isolated from trophozoites treated with 80 nM SC83288 for 4 h, relative to untreated controls. Statistical evaluation, two-sided Welch's *t* test. **d** In vitro DNA methyltransferase activity of nuclear extracts from trophozoites pretreated with 80 nM SC83288 for 4 h, compared with untreated controls or parasites co-treated with SC83288 and 2.0 mM SAM. Statistical

evaluation, Kruskal–Wallis one-way ANOVA ($H = 22.94$, $p = 1.05 \times 10^{-5}$) followed by Dunn's multiple comparisons test (two-sided; Bonferroni adjusted). **e** In vitro SC83288 dose–response curve for the enzymatic activity of purified PfDNMT2. Data represent mean ± SEM of three independent biological replicates, with the curve fitted using a one-parameter sigmoidal model. **f** In vivo SC83288 dose–response curves of the wild-type parental line 3D7 (black), a partial PfDNMT2 knockdown mutant (green), and a line overexpressing PfDNMT2-12-fold (red). Data represent mean ± SEM of three independent biological replicates. In (**b–d**), each data point corresponds to one independent biological replicate; box plots summarize data distribution. For all box plots in this figure, boxes indicate the interquartile range with the median shown as a black line and the mean as a red line; whiskers denote the 10th and 90th percentiles. For subfigures (**a–d**), *p*-values and the number of independent biological replications (*n*) are indicated in the graph. For all subfigures, source data are provided as a Source Data file.

Comparative, unbiased metabolomics revealed perturbations in cellular methylation, reflected by an altered methylation index (SAM:SAH ratio) and a diversion of SAM fluxes toward the CDP–choline and purine salvage pathways, resulting in increased MTA and adenine levels and decreased phosphoethanolamine in SC83288-treated cells (Fig. 7). Such metabolic rerouting suggests an adaptive response, in which the parasite compensates for reduced demand of macromolecular transmethylation by consuming excess SAM through upregulated polyamine and phosphatidylcholine biosynthesis. This mechanism has been described in other systems as a means of maintaining SAM homeostasis under disrupted DNA, RNA, or protein methylation, leading to accumulation of MTA and its degradation product adenine, accompanied by reduced phosphoethanolamine levels[24,45].

Methyltransferases commonly form complexes with SAHH to ensure efficient removal of SAH, a potent product inhibitor, by converting it to homocysteine and adenosine[23]. By analogy, PfDNMT2 is likely to associate with PfSAHH, raising the possibility that SC83288 binding to PfDNMT2 indirectly contributes to conformational stability of PfSAHH. Such an effect may explain the SC83288-induced stabilization of PfSAHH detected by ITDR-CETSA (Fig. 9c). Alternatively, stabilization may result from elevated adenine and MTA − both known SAHH inhibitors[23,24] − which accumulate in SC83288-treated cells (Fig. 7a). A direct interaction with the active NAD⁺ containing holo enzyme appears unlikely, based on in silico studies and enzymatic activity assays (Fig. 9d and Supplementary Fig. 11). While the holo enzyme is unlikely targeted by the compound, initial evidence from computational analyses suggests that the apo form of PfSAHH can bind SC83288 in its vacant NAD⁺ binding domain (Fig. 9d). Whether this predicted interaction is biologically meaningful – e.g., by delaying enzyme maturation or assembly – remains to be seen. Clarifying the mechanistic basis of PfSAHH stabilization, and its functional relationship to PfDNMT2, will require further study.

The comparative, unbiased RNA-seq analysis revealed a distinct transcriptional signature in SC83288-treated parasites, with six of nine upregulated genes encoding proteins of nuclear or partially nuclear localization (Fig. 6). These include histone deacetylase 1, RuvB-like helicase 3, eIF4A, aurora kinase ARK1, α-tubulin 1, and SEC61 subunit β, which function in epigenetic regulation, chromatin remodeling, transcription, mitosis, spindle formation, and protein translocation, respectively. These transcriptional responses are consistent with nuclear targeting by SC83288 and likely reflect compensatory mechanisms triggered by impaired epigenetic regulation and karyokinesis, particularly via inhibition of PfDNMT2. The remaining three upregulated genes (DER1-like protein, serine hydroxymethyltransferase, and plasmepsin IV) are indicative of stress responses, likely linked to misfolded protein accumulation, impaired protein trafficking, and metabolic adaptation.

ITDR-CETSA also revealed pronounced thermal stability shifts for SERA5, fumarate hydratase, and isocitrate dehydrogenase (Fig. 9c).

SERA5 belongs to a family of cysteine proteases; however, unlike its active paralogs, SERA5 is catalytically inactive and instead appears to function as a structural regulator during parasite egress following sequential proteolytic processing[35]. Molecular docking, using the crystallographic structure of the SERA5 pseudo-zymogen[36], revealed only weak, surface-associated binding poses for SC83288, which are unlikely to represent specific interactions. These results suggest that the observed CETSA shift in SERA5 stability is more likely an indirect consequence, for example, a conformational change induced by perturbed metabolite(s) or a protein–protein interaction effect, rather than direct drug binding. The lack of catalytic activity also precluded development of an in vitro inhibition assay, limiting further investigation of the underlying mechanism.

Fumarate hydratase and isocitrate dehydrogenase function within the mitochondrial tricarboxylic acid (TCA) cycle, catalyzing the interconversion of fumarate to malate and isocitrate to α-ketoglutarate, respectively. Genetic studies indicate that both enzymes are non-essential for asexual intraerythrocytic development, consistent with the parasite's reliance on glycolysis for ATP production and the presence of a largely anabolic or branched TCA cycle[46,47]. Thus, inhibition of these enzymes is unlikely to account for SC83288's antiplasmodial activity, although transient or low-affinity interactions cannot be excluded. Molecular docking using homology models (in the absence of crystal structures) did not support direct binding, revealing only surface-associated poses with unfavorable docking scores. Definitive clarification of whether the observed CETSA-derived thermal shifts represent direct drug binding, indirect metabolic effects, or false positives will require additional experimentation, including in vitro enzyme activity assays, in vivo measurements of NADP(H) redox and α-ketoglutarate/isocitrate ratios, and stable-isotope tracing to quantify flux redistribution under drug pressure.

The antagonistic interaction between SC83288 and pyrimethamine was unexpected. Pyrimethamine inhibits the folate biosynthesis pathway, which supplies the methyl group transferred from 5′-methyl tetrahydrofolate (5-methyl THF) via methionine and SAM to nucleic acids[21] (Fig. 7c). Inhibition of this pathway could be advantageous for parasites under SC83288 treatment, as the reduced synthesis of folate and thymidylate due to pyrimethamine might mitigate cellular stress caused by SC83288-induced disruption in methylation demand. Alternatively, malaria parasites exposed to SC83288 may not rely on DHFR activity. The activation of SAM/polyamine metabolism by SC83288 consumes 5-methyl THF but concurrently generates tetrahydrofolate (THF), which can sustain the dTMP cycle and bypass the requirement for DHFR. Furthermore, *P. falciparum* is capable of salvaging 5-methyl THF from the host, potentially further alleviating the reliance on the folate biosynthesis pathway[48]. High levels of THF have also been reported to reduce the antimalarial activity of pyrimethamine. Together these

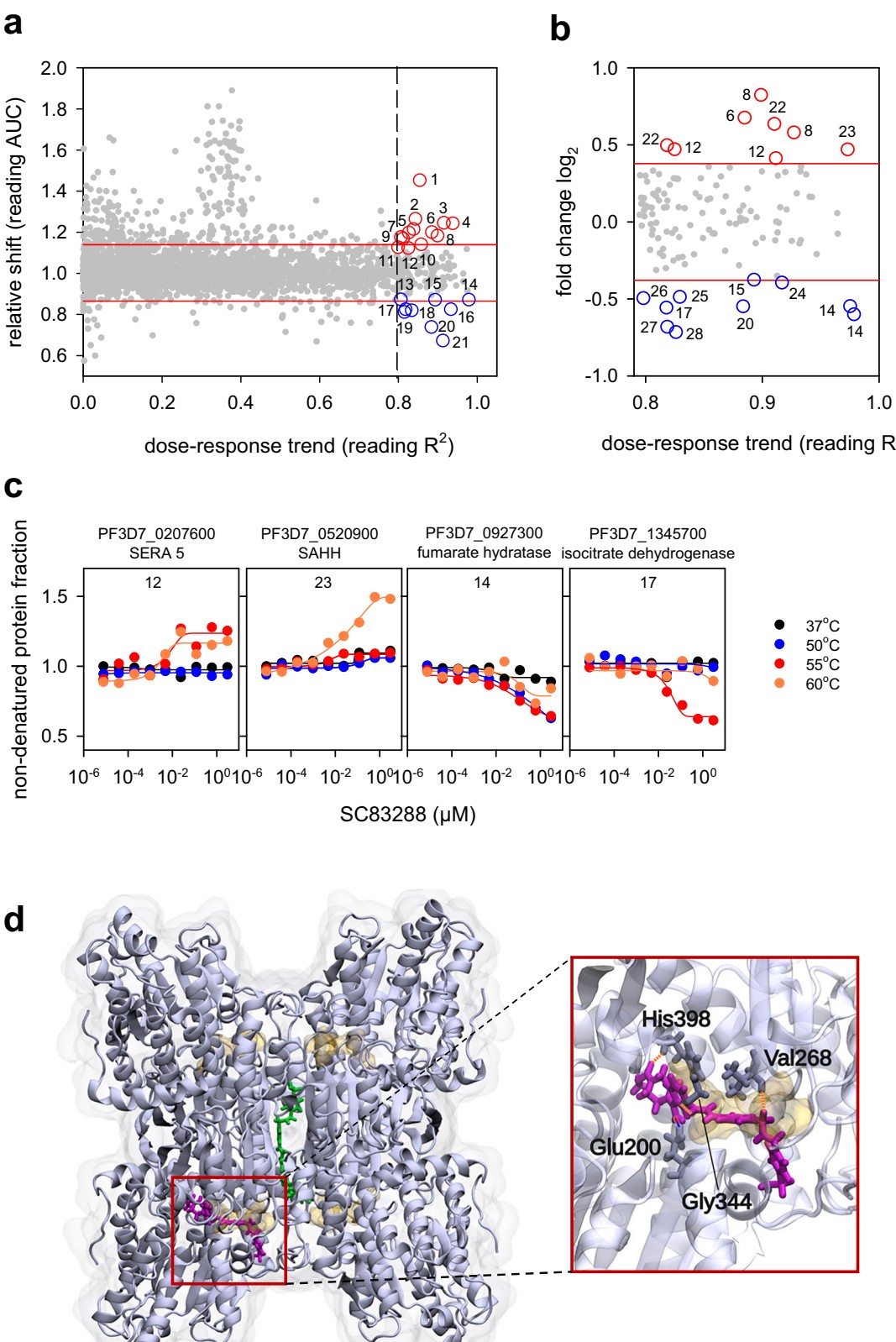

mechanisms may contribute to the antagonistic relationship observed between SC83288 and pyrimethamine (Supplementary Fig. 7).

Drug resistance in *P. falciparum* often arises from mutations within a single primary target, as seen for pyrimethamine (DHFR)[21], sulfadoxine (DHPS)[21], atovaquone (cytochrome bc)[49], MMV390048

(phosphatidylinositol 4-kinase, PI4K)[50], spiroindolines such as KAE609 (PfATP4)[51], and fosmidomycin (DOXP reductoisomerase)[52]. When a single target is not involved, alternative mechanisms emerge, including altered protein degradation, as for artemisinins (PfKelch13), or drug redistribution, such as efflux from the food vacuole mediated by PfCRT[21,53,54].

**Fig. 9 | Identification of putative SC83288 protein targets by intact-cell ITDR-CETSA. a** Protein stabilization plot showing area under the curve (AUC) of thermal response profiles versus coefficient of determination ($R^2$) for all detected proteins. Thermal response curves were generated at 50 °C, 55 °C, 60 °C, and a non-denaturing 37 °C control across increasing SC83288 concentrations and normalized to untreated samples. Red horizontal lines indicate ±2 times median absolute deviation (MAD) of AUC values; the dashed vertical line denotes $R^2 > 0.8$. Proteins exceeding both thresholds are numbered; stabilized and destabilized proteins are shown in red and blue, respectively. Experiments were performed on trophozoites treated with SC83288 (7.7 pM to 3.0 μM) for 1 h. **b** Alternative stabilization plot showing proteins with fold change ≥30% after thermal challenge plotted against $R^2 > 0.8$. Identified hists (see Supplementary Table 6 and Supplementary Data File 3): 1, BRCA2 protein, putative (PF3D7_1328200); 2, carbamoyl phosphate synthetase (PF3D7_1308200); 3, conserved membrane protein, unknown function (PF3D7_0505700); 4, conserved protein, unknown function (PF3D7_0811400); 5, AP-4 complex subunit ε, putative (PF3D7_0904100); 6, 40S ribosomal protein S17, putative (PF3D7_1242700); 7, cytochrome *c* oxidase copper chaperone, putative (PF3D7_1025600); 8, activator of Hsp90 ATPase, putative (PF3D7_0308500); 9, subtilisin-like protease 2 (PF3D7_1136900); 10, PHISTc exported protein, unknown function (PF3D7_0424000); 11, PHISTc exported protein, unknown function (PF3D7_1001800); 12, serine repeat antigen 5 (PF3D7_0207600); 13, cytochrome b5-like heme/steroid-binding protein, putative (PF3D7_0918100); 14, fumarate hydratase (PF3D7_0927300); 15, 1-Cys peroxiredoxin (PF3D7_0729200); 16, prolyl-4-hydroxylase α-subunit, putative (PF3D7_0829400); 17, cytosolic iron–sulfur protein assembly protein 1, putative (PF3D7_1209400); 18, isoleucine-tRNA ligase, putative (PF3D7_1225100); 19, casein kinase 2 α-subunit (PF3D7_1108400); 20, DNA replication licensing factor MCM2 (PF3D7_1417800); 21, kinetochore protein 7, putative (PF3D7_0410900); 22, conserved protein, unknown function (PF3D7_1133200); 23, S-adenosylhomocysteine hydrolase (PF3D7_0520900); 24, eukaryotic translation initiation factor 5, putative (PF3D7_1206700); 25, pseudouridylate synthase, putative (PF3D7_0914000); 26, glutaredoxin-like protein (PF3D7_0606900); 27, isocitrate dehydrogenase [NADP] (PF3D7_1345700); 28, tetratricopeptide repeat protein, putative (PF3D7_1310800). **c** Thermal response curves of the four highest-confidence hits. **d** Docking of SC83288 to the NAD-free (apo) structure of PfSAHH. Glide (green) and Vina (purple) poses are shown. The NAD binding pocket is highlighted in yellow. The Vina pose predicts an ionic interaction with E200 and backbone hydrogen bonds with V268, G344, and H398. Source data for (**a**–**c**) are provided as a Source Data file.

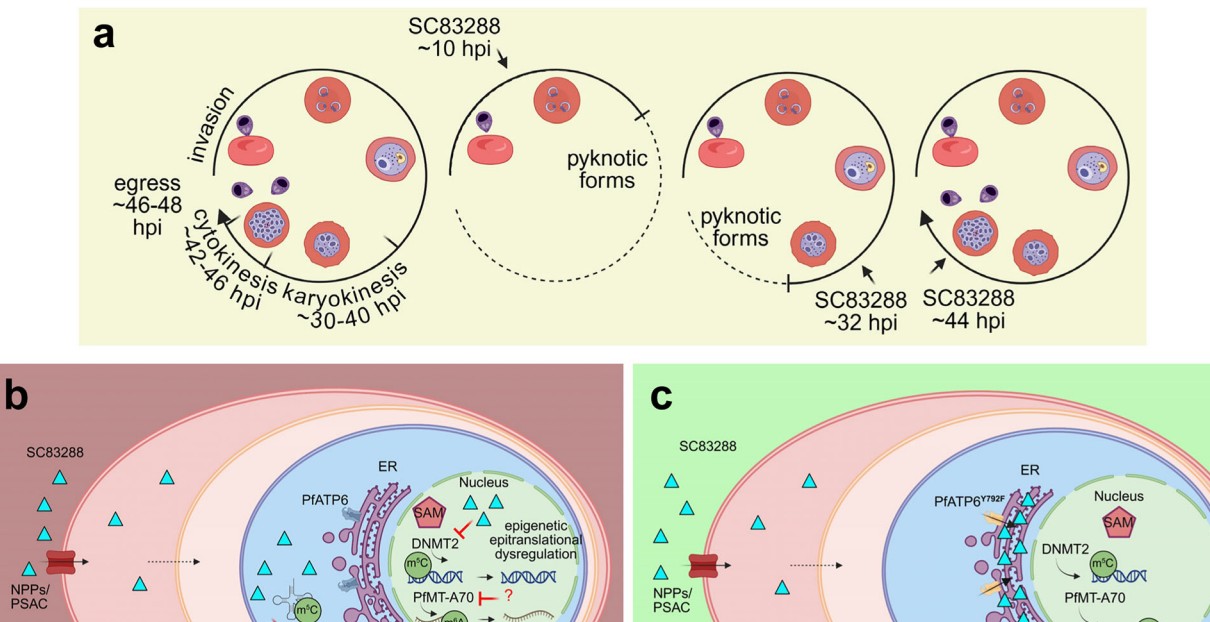

**Fig. 10 | Proposed mode of action and resistance mechanism of SC83288. a** Schematic illustrating the effect of SC83288 on the *P. falciparum* cell cycle. SC83288 arrests karyokinesis by inhibiting DNA replication, resulting in pyknotic parasite forms when added within the first 32 h post-invasion. When applied after completion of karyokinesis, parasites successfully complete schizogony, including cytokinesis and egress, indicating that the compound specifically targets nuclear replication processes. Generated in Biorender: https://BioRender.com/sl5elvi. **b** Proposed model of SC83288's mode of action. After entering infected erythrocytes via parasite-induced new permeation pathways (NPPs/PSAC), SC83288 accumulates within the parasite cytoplasm, where it targets PfDNMT2—implicated in nuclear m5C DNA methylation and cytoplasmic m5C38 tRNA^Asp methylation—and possibly PfMT-A70, the enzyme responsible for m6A RNA methylation. Inhibition of these enzymes leads to epigenetic and epitranslational dysregulation, culminating in replication arrest and parasite death. Generated in Biorender: https://BioRender.com/hg6e2nd. **c** Proposed mechanism of resistance. Mutations in the SERCA-type Ca²⁺ ATPase PfATP6, particularly the F972Y substitution, confer resistance by enabling enhanced transport of SC83288 into the endoplasmic reticulum (ER), where the drug is sequestered away from its nuclear targets. Generated in Biorender: https://BioRender.com/urcrdi7.

The PfATP6-mediated sequestration of SC83288 represents a distinct resistance paradigm that differs fundamentally from both canonical target-based or efflux mechanisms and PfKelch13-mediated artemisinin tolerance. Although both involve adaptive responses that mitigate drug-induced damage rather than altering the drug's primary target, their modes of action diverge. PfKelch13 mutations reduce endocytosis of host hemoglobin into the food vacuole, thereby limiting hemoglobin proteolysis and the release of heme, which is required for artemisinin activation[54]. This attenuation of drug activation diminishes oxidative stress and underlies partial artemisinin resistance[54]. In contrast, PfATP6-mediated resistance redistributes SC83288 into the endoplasmic reticulum, spatially separating the

compound from its nuclear target, PfDNMT2. This form of intracellular drug sequestration, coupled with its high fitness cost and absence of target mutations, defines SC83288 resistance as a mechanistically novel and distinct adaptation. From a translational perspective, this mechanism provides short-term survival but imposes a significant metabolic burden, likely restricting its spread in natural parasite populations.

That PfATP6, and not PfDNMT2, is consistently mutated in resistant lines, despite PfDNMT2 overexpression conferring protection, can be explained by the principle of minimal fitness cost. PfDNMT2 is a vital, non-redundant enzyme that functions in both the nucleus and cytoplasm, forming dynamic complexes with compartment-specific partners[28]. Overexpression likely perturbs this stoichiometric balance, creating aberrant binding sinks that sequester shared cofactors or partners and compromise essential functions. Such disruption would carry a high fitness penalty, rendering stable overexpression unsustainable in vivo. In contrast, PfATP6, as a membrane transporter, is structurally more tolerant to adaptive changes that modify substrate specificity. Consequently, selection favors PfATP6 mutations as the more viable evolutionary solution to counteract SC83288 toxicity, even if associated with substantial growth defects.

Based on the findings presented herein, we propose the following model for the mode of action and mechanism of resistance of SC83288 (Fig. 10). After entering infected erythrocytes through new permeability pathways (NPPs/PSAC), SC83288 accumulates ~100-fold within the parasite compartment. The compound then targets the nucleus, where it arrests DNA replication and karyokinesis through inhibition of PfDNMT2 (Fig. 10a, b), likely leading to DNMT–DNA adduct–mediated fork stalling or hypomethylation-triggered checkpoint activation. This disruption of epigenetic control is accompanied by a reduction in m⁶A RNA methylation (possibly via affecting RNA N⁶-methyladenosine (m⁶A) methyltransferase complex), resulting in loss of epitranscriptomic and translational regulation, collectively causing transcriptional dysregulation and parasite death (Fig. 10b). At present we cannot exclude the possibility of SC83288 targeting other methyltransferases. Resistance to SC83288 is conferred by point mutations in PfATP6, which enhance its ability to transport and sequester the drug into the endoplasmic reticulum, thereby diverting it away from its nuclear target (Fig. 10c). The dual-acting mechanism, together with a fitness-costly resistance phenotype, underscores the therapeutic promise of SC83288 as a next-generation antimalarial candidate with a novel and multifaceted mode of action.

## Methods
### Compounds
SC83288 (methyl 4-(imino(3-(3-(4-(N-(4-sulfamoylbenzyl)sulfamoyl) phenyl) ureido) phenyl)methyl) piperazine-1-carboxylate) was synthesized by Almac, Ireland, under preGMP conditions. Purify: >95%; $[M + H]^+$ calculated for $C_{27}H_{31}N_7O_7S_2$, 630.180; found for $[M + H]^+$: 630.1; ¹H-NMR (300 MHz, DMSO-$d_6$) δ = 3.28–3.32 (m, 2 H), 3.55–3.62 (m, 6H), 3.64 (s, 3H), 4.03 (d, J = 6.1 Hz, 2H), 7.21 (d, J = 7.5 Hz, 1H), 7.29 (bs, 2H), 7.44 (d, J = 8.3 Hz, 2H), 7.54 (t, J = 7.9 Hz, 1H), 7.63–7.78 (m, 7H), 7.86–7.89 (m, 1H), 8.08 (t, J = 6.3 Hz, 1H), 9.39 (bs, 1H), 9.90 (s, 1H), 9.96 (s, 1H). ¹³C-NMR (75 MHz, DMSO-$d_6$) δ = 42.23 (2x), 45.55 (3x), 52.58, 117.43, 117.74 (2x), 121.46, 121.60, 125.56 (2x), 127.80 (4x), 129.63, 129.81, 132.97, 140.37, 142.01, 142.91, 143.47, 152.41, 154.92, 164.39. element. anal.: element: (calc. comp. incl. add. (%)/found (%)): C: (48.68/48.25), Cl: (5.32/4.77), H: (4.84/5.6), N: (14.72/14.51), O: (16.81/18.1), S: (9.63/7.99). [³H]-SC83288 was purchased from Quotient Bioresearch. SC196879 was synthesized by 4SC AG.

### Primers
Primers are listed in Supplementary Table 8 and were purchased from Eurofins Genomics.

### Parasite culture
*P. falciparum* parasites were cultured in human A⁺ red blood cells at a hematocrit of ~3.5% in RPMI 1640 medium supplemented with 25 mM HEPES, 2mM L-glutamine, 0.2 mM hypoxanthine, 5 % (v/v) human serum, 5 % (v/v) albumax and 20 μg ml⁻¹ gentamycin at 37 °C under atmospheric conditions of 5% $O_2$, 3% $CO_2$, 92% $N_2$ and 95% humidity[55]. The parasitemia was assessed using Giemsa-stained thin blood smears. Parasite cultures were synchronized within a window of 4 hours using a combination of 5% sorbitol[56] and 5 units ml⁻¹ heparin[57]. Trophozoites were enriched from highly synchronized parasite cultures using the magnet cell sorting method[58].

### Site-directed mutagenesis of *pfatp6* in *P. falciparum*
The desired F972Y mutational change was introduced into the genomic *pfatp6* sequence, using CRISPR/Cas9 genome editing technology[10]. The pL6 transfection vector was digested with the restriction endonuclease BtgZI and the PfATP6 guide sequence was cloned into the vector using the In Fusion cloning approach and primers ATPase6-guide 7-for and ATPase6-guide 7-rev. The pfatp6 homology region containing the single nucleotide polymorphism coding for the mutation F972Y, as well as shield mutations was amplified from cDNA from the *P. falciparum* line Dd2 with the primers ATPase6-HRB-2269-SpeI-for, ATPase6-HRB-3409-BssHII-rev, ATPase6-shield(guide7)-for and ATPase6-shield(guide7)-rev and cloned into the transfection vector using the restriction enzymes SpeI and BssHII. Ring stages of the *P. falciparum* line 3D7 were transfected by electroporation with 100 μg each of the pL6 and pUF1-Cas9 transfection vectors[10,59]. Transfectants were selected with 5 nM of WR99210 and 1.5 μM of DSM1. Clonal lines of the CRISPR/Cas9 edited parasites were obtained by limiting dilution, without drug pressure. The sequence of the edited *pfatp6* gene was confirmed for each selected clone by sequencing the entire gene following amplification from genomic DNA, using the primer pair PfATP6-amp-for and PfATP6-amp-rev (Supplementary Table 8).

### Growth assays
Growth inhibition assays were performed according to standard protocol based on the detection of parasitic DNA by fluorescent SYBR® green staining[60]. Briefly, a synchronized culture of ring-stage parasites was incubated in the presence of decreasing drug concentrations, at the following final conditions: 100 μl per well, 0.5% parasitemia, 2% hematocrit, and 72 hours incubation at 37 °C. After incubation, the plates were frozen at −80 °C overnight. On the day of the measurement, plates were thawed for 1 hour at RT. Subsequently, 100 μl per well of a 1x SYBRGreen® (ThermoFisher Scientific Inc.) solution in lysis buffer ($H_2O$, 20 mM Tris base pH 7.4, 5 mM EDTA, 0.008% (w/v) Saponin, 0.08% (w/v) Triton X-100) was added, briefly shaken and incubated at room temperature for 1 h in the dark. Fluorescence was measured in a fluorescence plate reader (FluoStar Optima, BMG Labtech GmbH) (ext/em: 485/520 nm, gain 1380, 10 flashes/well, top optic). Parasite development in the presence of the drug was normalized to untreated control and graphed as a function of the drug concentration. A four-parameter Hill function was fitted to the data points and the $IC_{50}$-value was calculated, using SigmaPlot 16 (Systat Software Inc.).

### Fitness assay
The relative fitness of parasites was determined in a growth competition assay. The F972Y PfATP6 clone and the parental *P. falciparum* 3D7 line were carefully counted, mixed 1:1 and seeded at an initial parasitemia of 3% of trophozoite stages. The parasitemia was maintained between 0.5 and 6.0% to ensure optimal growth conditions. The mixed parasites were kept in culture for 20 cycles. To determine the ratio of both strains in the mixture overtime, saponin-lysed parasite pellets of the mixed cultures were collected every 4 days, and the genomic DNA

was extracted from the pellets using the DNeasy Tissue Kit (Qiagen). The gDNA was then used as a template for amplification of the *pfatp6* sequence using the forward primer PfATP6-for (Fit/PCR) and the biotinylated reverse primer PfATP6- rev2 (Fit/Biotin). 37 μl of the resulting PCR product were mixed with 3 μl of Streptavidin Sepharose HP (GE Healthcare) and 40 μl of binding buffer (PBS 10 mM Tris, 2 M NaCl, 1 mM EDTA, 0.1% Tween 20, pH 7.6). The mixture was transferred to a 96 well microtiter plate and shaken for 10 min at 1400 rpm. Each sample was analyzed in duplicates by pyrosequencing, using the sequencing primer PfATP6-for (Fit/seq) and the PyroMark Q96 workstation (Qiagen), as recommended by the manufacturer.

### Determining the resting free $Ca^{2+}$ concentration with FuraRed

The ratiometric dye FuraRed was used to determine the resting cytosolic free $Ca^{2+}$ concentration in live parasites at the single cell level[11]. To this end, trophozoites were incubated with 9 μM of FuraRed-AM in Ringer solution (122.5 mM NaCl, 5.4 mM KCl, 1.2 mM CaCl2, 0.8 mM MgCl2, 11 mM D-glucose, 10 mM HEPES, 1 mM NaH2PO4, pH 7.4) with 0,1% (v/v) Pluoronic F-127 and 40 μM of probenecid for 45 min at 37 °C. Dye-loaded parasites were washed two times before settling onto a poly-L-lysine-coated coverslip in an 8-well microperfusion chamber (ibidi). Confocal laser scanning fluorescence microscopy was performed using a confocal Leica TCS SP5 (Leica Microsystems CMS GmbH). Fura-Red stained samples were alternately excited with the 458 nm and 488 nm argon laser lines. Fura-Red fluorescence was measured using a 570-nm-long pass filter, resulting in a pair of images (F458 nm and F488 nm). Single images were obtained using a 63× objective, 1024 × 1024 pixels. Single cell fluorescent images were further analyzed using Fiji and regions of interest were defined to measure the mean fluorescent intensity. A region of interest was also defined to allow for background subtraction. The mean fluorescence ratio (R458/488 nm) was then calculated. Calibration of the FuraRed fluorescence was achieved using the Calcium Calibration Buffer Kit (ThermoFisher Scientific). Cells were permeabilized with 10 μM of the $Ca^{2+}$ ionophore ionomycin, adjusted to varying free $Ca^{2+}$ concentration, and imaged. A Hill function was fitted to the data points, with the Hill coefficient fixed to unity reflecting the 1:1 stoichiometry of the FuraRed binding to $Ca^{2+}$ ions. A Grubbs test was performed to identify outliers in the data sets (at the $p = 0.01$ level). These outliers were omitted for further analysis.

### Live cell $Ca^{2+}$ imaging using Fluo-4

Trophozoite-stage *P. falciparum* 3D7 parasites were pre-treated with 6 nM of XR-9576 for 10 min at 37 °C, in order to avoid the accumulation of Fluo-4-AM into the parasite´s digestive vacuole[61]. The parasites were further incubated with 10 μM of Fluo-4-AM in RPMI 1640 medium (Life Technologies) with Pluoronic F-127 (0.1% v/v) and 40 μM of probenecid for 45 min at 37 °C[11]. Dye-loaded parasites were washed two times before settling onto a poly-L-lysine-coated glass slides in a microperfusion chamber (ibdi). Confocal laser scanning fluorescence microscopy was performed using a Leica TCS SP8 (Leica Microsystems CMS GmbH). Fluo-4-AM was excited at 488 nm (argon laser, 0.03%) and the emitted fluorescence was collected from 505 to 520 nm. Single images were obtained using a 63× objective, with a 5-fold software zoom, 1024 × 1024 pixels, every 2 s over a time span of 180 s. The $Ca^{2+}$ ATPase inhibitor CPA, the $Na^{+}/K^{+}$ ATPase inhibitor oubain, and SC83288 were added to a final concentration of 10 μM at 40 s. The calcium signal was further analyzed using Fiji.

### Yeast complementation assay

The yeast codon-optimized versions of *pfatp6*$^{wt}$ and the mutant *pfatp6*$^{F972Y}$ (Geneart) were cloned into the yeast expression vector pNEV-N vector under the control of the PMA1 promoter and terminator regions. pNEV-N contains the ura3 gene conferring uracil autotrophy. The resulting vectors were transformed into the yeast strain K667 (MATa ade2-1 can1-100 his3-11,15 leu2-3,112 trp1-1 ura3-1

cnb1::LEU2 pmc1::TRP1 vcx1Δ), which exhibits an intolerance to high $Ca^{2+}$ concentrations[62]. Transformants were selected for uracil prototrophy by plating on synthetic medium minus uracil (SC media minus uracil: 6.7 g $L^{-1}$ yeast nitrogen base without ammonium sulfate and amino acids, 1.92 g $L^{-1}$ drop-out mix without uracil, 2% glucose, and 2% agar). Single colonies were grown in liquid SC media minus uracil to mid-log phase before cells were centrifuged and washed three times with water. The yeast cells were serially diluted with water to obtain $10^{-1}$, $10^{-2}$, and $10^{-3}$-fold dilutions of the initial culture with an $OD_{600}$ of 0.5. 10 μl of each serial dilution was dotted on an YPD and an SC minus uracil plate containing 100 or 125 mM CaCl2. Plates were incubated at 30 °C for 3 days. The yeast strain K607 (MATa ade2-1 can1-100 his3-11,15 leu2-3,112 trp1-1 ura3-1 cnb1::LEU2 pmc1::TRP1)[62] served as a negative control.

### Functional characterization of PfATP6 containing yeast vesicles

A P2 vesicle fraction was prepared as previously described with minor modifications[63]. Briefly, K667 yeast clones expressing *pfatp6*$^{wt}$ and pfatp6$^{F972Y}$ were grown in SD minus uracil medium to an $OD_{600}$ of 0.5, centrifuged, washed with 10 mM NaN3, and subsequently converted to spheroplasts using 300U lyticase per 10 $OD_{600}$ cell units in buffer A (140 mM potassium phosphate, 1.2 M sorbitol, 0.2% ß-mercaptoethanol, 10 mM NaN3, pH 7.5) for 30-45 min at 30 °C. Spheroplasts were subsequently resuspended in buffer B (10 mM triethylamine-acetate, 0.8 M sorbitol, 1 mM EDTA, 1 mM PMSF pH 7.2) at a density of 1 g of cells per 1 ml and broken by drawing the cell suspension rapidly several times through a 27 G ¾ needle. Cell breakage was not complete under these conditions, but vesicle integrity was very well preserved. The lysate was centrifuged at 1500 × *g* for 20 min at 4 °C to give a first pellet, termed P1, and a supernatant. This supernatant was centrifuged at 100,000 × *g* for 1 h at 4 °C to obtain the P2 vesicle fraction. This P2 pellet was gently resuspended in buffer A at 5–10 mg protein $ml^{-1}$, aliquoted, and immediately frozen in liquid nitrogen. Each aliquot was thawed only once to preserve vesicle integrity. The luminal marker guanosine diphosphatase (GDPase) was assayed as an indicator of vesicle latency using the Sensolyte MG Phosphate assay kit for inorganic phosphate measurements, as described[64]. The vesicle latency was consistently greater than 90%. The vesicles exhibited activity of the ER marker enzyme NADPH-cytochrome C reductase (0.06−0.09 nmol $mg^{-1}$ of protein $min^{-1}$), as determined using the Cytochrome c Reductase (NADPH) Assay Kit (Sigma-Aldrich) according to the manufacturer's instruction. Uptake of [$^{3}$H]-SC83288 by isolated membrane vesicles was determined using the filtration method[65]. Vesicles were incubated in buffer A (10 mM Tris-HCl, 160 mM KCl, 5 mM MgCl2 pH 7.2) for 15 min at 30 °C (10 μg protein per assay and run in triplicates). The reaction was stopped by adding 100× volume of ice-cold stop buffer (10 mM MOPS-KOH, 150 mM KCl, 5 mM MgCl2 pH7.2), filtered on 0.45 μM nitrocellulose filters (Millipore) and washed 3 times with 1 ml stop buffer. Radioactivity was measured by scintillation counting. The final concentrations used were: 0.1 μM [$^{3}$H]-SC83288, 1 mM ATP, 10 μM CaCl2, 15 μM CPA, and 0.05% NP40. Each sample was analyzed in triplicates.

### SC83288 uptake assay

Uptake assays were performed as described[66]. Briefly, magnetically enriched trophozoite-infected erythrocytes were resuspended in reaction buffer A (bicarbonate-free RPMI 1640, 11 mM glucose, 25 mM HEPES-Na, 2 mM glutamine, pH 7.3 at 37 °C) and hematocrit adjusted between 20,000 and 30,000 infected erythrocytes per ul, using a Neubauer counting chamber. Cells were resuspended in prewarmed reaction buffer A containing 50 nM [$^{3}$H]-SC83288. The reaction was held at 37 °C, and the time course of [$^{3}$H]-SC83288 accumulation was monitored. Every 5 min, 2 μl of 0.5 M glucose was added to the cells. Duplicative 75 mL aliquots were removed from the reaction at various time points, diluted with an equal volume of ice-cold reaction buffer A,

and spun down at $15,000 \times g$ for 1 min through a layer of separation oil (a 5:4 mixture of dibutylphthalate and dioctylphthalate) to separate the cells from the aqueous medium, which contained the unincorporated [³H]-SC83288. The aqueous phase was collected, and its radioactivity was determined to obtain an accurate measurement of the extracellular [³H]-SC83288 concentration. The cell pellets were recovered by cutting the reaction tubes through the oil layer. The tips of the tubes containing the cell pellets were placed in a larger 1.5 ml Eppendorf tube and incubated with 100 μl of tissue solubilizer (2:1 mixture of ethanol and tissue solubilizer from Pharmacia) overnight at 55 °C. The lysates were decolorized by the addition of 25 μL of 30% $H_2O_2$ and acidified by the addition of 25 μL of 1 N HCl. The lysates were transferred to scintillation vials, and the radioactivity was measured using a liquid scintillation counter (TRI-CARB 2100 TR, Packard). The intracellular SC83288 concentration was calculated from the amount of [³H]-SC83288 taken up by the cells and by assuming that the volume of a trophozoite-infected erythrocyte is 75 fl[66]. SC83288 accumulation was expressed as the ratio of the intracellular versus the extracellular SC83288 concentration ($SC83288_{in}/SC83288_{out}$). Uninfected erythrocytes were investigated in parallel. Where indicated, experiments were performed in the presence of the proton ionophore carbonyl cyanide-p-trifluoromethoxyphenylhydrazone (10 μM, FCCP), and the new permeation pathways inhibitor 5-Nitro-2-(3-phenylpropylamino) benzoic acid (5 μM, NPPB) or under glucose-free conditions in reaction buffer B (bicarbonate-free RPMI 1640, 25 mM HEPES-Na, 2 mM glutamine, pH 7.3 at 37 °C) for 20 min. Cells were held in reaction buffer B for 10 min to deplete them of intracellular ATP prior to the commencement of the experiment.

## SC83288 efflux assay

*P. falciparum*-infected erythrocytes at the trophozoite stage were prepared as described above and incubated in reaction buffer A containing 50 nM of [³H]-SC83288 at 37 °C for 15 min (hematocrit of 25,000 cell per ml). Cells were then washed twice in ice-cold reaction buffer A (pH 7.3 at 4 °C) and resuspended in pre-warmed reaction buffer A. The reaction was held at 37 °C and the time-course of [³H]-SC83288 was monitored by removing duplicate 75 μl aliquots at the time points indicated and processing it as described above.

## Localization of subcellular SC83288 accumulation using click chemistry

*P. falciparum* cultures of 3D7 and the PfATP6$^{F972Y}$ mutant at the early trophozoite stages were incubated with 10 μM of the clickable SC83288 derivative, SC106879, for 45 min under standard culture conditions. Cells were washed with PBS and fixed in 4% paraformaldehyde and 0.0075% glutaldehyde in PBS for 20 min at RT before cells were washed with PBS and permeabilized with 0.025% NP40 in PBS for 10 min at RT. Cells were subsequently washed with 10 mM glycine in PBS and incubated in 3% BSA in PBS for 30 min. After washing, a buffer containing 2.5 μM TAMRA azide (Click-IT cell reaction buffer, Thermo Fisher Scientific) was added and reactions were incubated for 30 min at RT in the dark. Afterward, samples were washed 3 times with 3% BSA in PBS and PBS alone. Cells were subsequently visualized by confocal laser scanning microscopy (Zeiss LSM510) equipped with UV and visible laser lines and a C-Apochromat objective with a 63× magnification. Fluorescence signals were collected in Multi-Track mode, using the following laser lines and filter settings: Nuclear staining was carried out using Hoechst 33342 (5 μM) and images were captured using a 385–470 band pass filter upon excitation at a wavelength of 364 nm, TAMRA: excitation wavelength 543 nm; emission detected using an LP 560 nm filter. The PfATP6$^{F972Y}$ mutant line E4 was transfected with the pARL vector containing the ER marker BiP fused to citrine. Confocal images were recorded using excitation and emission wavelengths of 488 nm and 505–550 nm (band pass filter), respectively. Images were processed using Fiji.

## C-value determination

Tightly synchronized parasite cultures (Dd2) were exposed to 30 nM SC83288 or 200 μM pyrimethamine at various timepoints during the 48 h replicative cycle. Samples were collected every 2–4 h from this timepoint onwards, until 52 h post-invasion, washed once with PBS, and then fixed (PBS, 4% paraformaldehyde, 0.0075% glutaraldehyde) for 2 h at room temperature. The samples were then washed once with PBS, resuspended in PBS, and kept at 4 °C until further analysis. The DNA content was assessed as described[19]. Briefly, the parasites were permeabilized (PBS 0.1% Triton X-100) for 8 min at room temperature before being washed two times with PBS. The samples were subsequently subjected to a light RNase treatment (0.3–0.5 mg.ml⁻¹ in PBS) for 30 min at 37 °C. The treated samples were then washed two times with PBS, and their DNA content was stained with SYBRGreen (1:2000 dilution in PBS) for 20 min at room temperature protected from direct light. After two PBS-washes, the parasites were transferred into tubes adapted for flow cytometry measurements. Fluorescence was measured by flow cytometry using a Beckton and Dickinson FACS Canto. A minimum of 2000 infected erythrocytes were analyzed per condition and independent biological replicate.

## Isobologram analysis

The fixed-ratio isobologram method was used to investigate the in vitro interaction between two compounds[9,67]. Briefly, a synchronous culture of ring-stages of the *P. falciparum* strain Dd2 was incubated in 96-well black microtiter plates for 72 h in the presence of decreasing drug concentrations alone (A or B) and in combination with the other drug at fixed concentration ratios (4:1, 3:2, 2.5:2.5, 2:3, and 1:4). Technical triplicates were performed for each condition and each independent biological replicate. Parasite multiplication was assessed using the fluorescent DNA stain SYBRGreen and the FLUOstar Optima fluorescence plate reader (BMG Labtech) ($\lambda_{exc} = 485$ nm, $\lambda_{em} = 520$ nm, gain 1380, 10 flashes/well, top optic). The half-maximal inhibitory concentration ($EC_{50}$) and the fractional half-maximal inhibitory concentration ($FIC_{50}$) (ratio of $EC_{50}$ of drug combination to $EC_{50}$ of drug alone) were calculated for each condition[9,67].

## Enzymes of the folate pathway

The enzymatic activity of DHFR was assessed as described[68]. Briefly, serial dilutions of SC8322 and established DHFR inhibitors from stocks of 10 mM were made in DMSO and pipetted in 96-well microtiter plates containing the appropriate amount of 1× reaction buffer (50 mM TES pH 7.0, 75 mM β-mercaptoethanol, 1 mg ml⁻¹ BSA, 100 μM NADPH, and 100 μM dihydrofolate. The reaction was initiated by adding an appropriate amount of 5–20 nM DHFR enzyme (PfDHFR$^{WT}$, PfDHFR$^{QM}$, or human DHFR). The enzymatic reaction was followed by recording the optical density at 340 nm ($OD_{340}$) for 1 min using a microplate reader (MULTISKAN GO, Thermo Scientific). The results were expressed as the half-maximal inhibitory concentration ($IC_{50}$) or percentage of inhibition at the highest concentration of the compound. To investigate the effect of SC83288 on PfDHPS and PfHPPK, previously described protocols were followed[69]. Briefly, activity buffer (100 mM Tris, pH 9, 100 mM β-mercaptoethanol, 10 mM magnesium sulfate) was mixed with HMDP-PP and pABA (final concentrations of 100 μM each) and the test compound (final concentration of 1 mM). The reaction was initiated by the addition of wild-type PfDHPS enzyme, and was incubated at 37 °C for 20 min. The reaction was then stopped by incubation at 95 °C for 5 min, followed by a 5 min incubation on ice. The samples were then centrifuged, filtered, and analyzed by Ultra High-Performance Liquid Chromatography (UHPLC) using a 0–50% ACN gradient in 0.1% formic acid in water over 10 min. In the case of the PfHPPK enzyme assay, 2 μg of PfHPPK-GFP enzyme was added to the reaction buffer containing 100 mM Tris pH 9, 10 mM β-mercaptoethanol, 10 mM MgSO₄, 0.01% w/v BSA, 10 μM HMDP, 10 μM ATP and 1 mM of the test compound. The reaction was carried out in white 96-

well microtiter plates and allowed to proceed for 20 min at RT under shaking at 300 rpm. The reaction was then quenched by the addition of 50 µL KinaseGlo reagent (Promega) and allowed to equilibrate for 10 min at RT under shaking at 300 rpm. Luminescence was recorded on a Biotek synergy H1 plate reader using an integration time of 1 s per well. For both the PfDHPS and the PfHPPK enzyme, the percentage of inhibition was calculated in relation to positive controls (no enzyme added) and negative controls (no inhibitor added).

## DNA binding assay

Different amounts of salmon sperm DNA or *P. falciparum* DNA (1–2 kb in size), ranging from 0 to 5 µg, were added to buffer containing 50 mM NaCl, 10 mM Tris-HCl, 10 mM MgCl$_2$, 100 µg ml$^{-1}$ BSA, and 0.5 µM [$^3$H]-SC83288 (pH 7.9 at 25 °C), yielding a total volume of 50 µl. The reaction was incubated for 30 min at RT before 50 µl 20% TCA was added. After 30 minutes of incubation on ice, the precipitate was filtered through Whatman glass microfiber filters, washed with 10% TCA, and then with ethanol, and let air dry for at least 10 min under a fume hood. Dry filters were placed in plastic scintillation counter vials, to which 2 ml of Ultima Gold (Perkin Elmer) liquid scintillation cocktail was added. Samples were measured in the Tri-Carb 4910 TR liquid scintillation counter (Perkin Elmer). The fraction of DNA-bound [$^3$H]-SC83288 was calculated in relation to the total input of [$^3$H]-SC83288. DNA binding of DAPI (4,6-diamidino-2-phenylindole, a minor groove binder) and TP3 (TO-PRO-3 iodide, an intercalator) (both purchased from Thermo Fisher Scientific) was conducted as described[70]. Briefly, [$^3$H]-SC83288 was replaced by 1 µM of the respective dye in the reaction. DAPI fluorescence was measured using the plate reader FLUOstar Optima (BMG Labtech) ($\lambda_{exc} = 358$ nm, $\lambda_{em} = 461$ nm, gain 1380), and TP3 fluorescence using the Cytation3 Reader (BioTEK) ($\lambda_{exc} = 633$ nm, $\lambda_{em} = 661$ nm, gain 100). Each sample was analyzed in triplicates.

## Untargeted transcriptomics analysis

An unbiased global transcriptomics analysis was conducted as described[71]. Briefly, highly synchronized trophozoites (25 ± 3 h post-invasion) were exposed to different concentrations of SC83288, encompassing 0 nM, 5 nM, and 10 nM, for a duration of 4.5 h. 190 µl of cultures were transferred post-treatment to a new hard-shell 96-well PCR plate (BioRad). Plates were centrifuged for 1 min at 1000 × *g* and the supernatant was aspirated and discarded. 60 µl of TRIzol reagent (Thermo Fisher Scientific) was added to 4 µl of remaining infected packed red blood cells. Plates were sealed and vortexed at 7000 RPM for 1 min. Subsequently plates were centrifuged for 1 min at 2000 × *g*. 50 µl of supernatant was aspirated and transferred to a new 2 ml 96-deep well plate (Corning). RNA was extracted using ZYMO DirectZol-96 MagBead RNA kit (ZYMO) following manufacturer instructions with minor modifications. In brief, 50 µl of 100% ethanol (Merck) was added to the supernatant and mixed by shaking at 1500 RPM for 30 s. Subsequently 10 µl of beads were added to each sample and incubated for 10 min with shaking. Beads were separated on the magnet (Alpaqua) and cleared supernatant was discarded. Beads were washed with 500 µl of wash buffer following manufacturer instructions. RNA was eluted in 17 µl of RNAse/DNAse free water, and its purity was assessed by spectrometry on Nanodrop (ThermoScientific). The RNA concentration was estimated using RNA-specific Qubit fluorometric assays (Thermo Fisher Scientific). RNA integrity was assessed on Bioanalyzer RNA Nano Chip (Agilent). RNA quality cut-off metrics were as follow: 260/280 ratio ≥1.8, 260/230 ratio ≥1.5, RIN ≥ 8, RNA concentration ≥5 ng µl$^{-1}$. RNA samples that passed quality metrics were sequenced. All steps were automated on Hamilton STAR liquid handling platform (Hamilton Robotics). 25 ng of total RNA was reverse-transcribed and amplified using a polyA-specific primer as previously described[71]. Resulted cDNA was processed using Nextera XT library prep kit (Illumina) to obtain short sequence reads according to manufacturer instructions with minor modifications. Briefly, 0.25 ng of cDNA was fragmented and amplified using downscaled library preparation protocol with 1/4 volume of recommended reagents used. PCR amplification cycles were increased from 12 to 15 cycles. PCR reactions were purified using 0.6× volumes of AMPure XP magnetic beads (Beckman Coulter) following manufacturer's instructions on Hamilton STAR liquid handling platform. Samples were eluted in 17 µl of 10 mM Tris-Cl, pH 8.5 (Qiagen) and average library sizes were estimated on Bioanalyzer High Sensitivity chips (Agilent). Library molarity was assessed with qPCR and libraries were pooled in equimolar ratios and sequenced on Novaseq S4 platform (Illumina) with pair-end 150 bp long reads protocol, generating on average 2.5 Gb of data per sample (Novogene Co.). Raw reads obtained from the sequencer were first assessed for quality and trimmed to remove adapters, amplification primers, and low-quality bases from the 3′ ends using Trim Galore with the parameters --quality 20 --phred33 --stringency 5 --trim-n --error 0.2 --length 35 –paired (https://github.com/FelixKrueger/TrimGalore). HISAT2 was used to perform alignment to the 3D7 genome v.68[72] with additional options --min-intronlen 20 --max-intronlen 3000. Paired reads with proper orientation mapped to unique locations of genome were considered for counting. Unique read pairs overlapping exon features were counted using featureCounts tool using unstranded protocol[73,74]. Read counts were scaled to TPM and correlated with the reference 3D7 intraerythrocytic developmental cycle to verify age similarity between samples. Raw counts were then imported into R and analyzed using the edgeR package[75]. Genes with zero expression across samples and overabundant ribosomal RNA and var gene family transcripts were filtered out. Samples with fewer than 1 million raw reads were removed. Remaining genes were filtered using --filterByExpr, and counts were normalized between samples using the Trimmed Mean of M-values (TMM) method to account for differences in library size and composition bias. Principal component analysis (PCA) was used to detect outlier samples. Differential expression analysis was performed in edgeR, comparing drug-treated samples at each concentration ($n = 3$) to DMSO controls ($n = 5$), using TMM-normalized counts with quasi-likelihood F-tests and likelihood ratio tests. P-values were adjusted for multiple testing using the Benjamini–Hochberg false discovery rate (FDR). All samples were processed within the same batch.

## Untargeted metabolomics analysis

Metabolites from the *P. falciparum* wild-type strain 3D7 or the PfATP6$^{F972Y}$ mutant line E4, in the absence or presence of SC83288 were extracted for quantitative analysis by mass spectrometry. Briefly, cultures at a parasitemia in trophozoites (20 ± 2 h post-invasion) of 5% were treated with a concentration of SC83288 corresponding to 5× IC$_{50}$ value (40 nM), or with the respective amount of DMSO, for 12 h. These conditions were pre-determined to achieve a killing of half of the parasite´s population. Parasites were then magnetically purified using the MACS system[58]. For each sample, 3 ×10$^7$ parasites were collected and washed with ice-cold PBS. The complete supernatant was removed, and the pellet was quickly resuspended in 150 µl of extraction buffer (chloroform/methanol/H$_2$O at a 1/3/1 ratio). Samples were incubated in a vortex for 1 h at 4 °C, and centrifuged for 10 min at 16,000 × *g* at 4 °C. 100 µl of each sample was transferred into a fresh tube, sealed, and kept at −80 °C until further analysis. The remaining 20 µl of each sample were pooled together and centrifuged for 10 minutes at 16,000 × *g* at 4 °C. The supernatant was transferred into a fresh tube, sealed, and kept at −80 °C until analysis. Quantitative mass spectrometry analysis was performed as previously described[76,77]. Briefly, samples were analyzed on a Thermo Scientific Q-Exactive Orbitrap mass spectrometer running in positive/negative switching mode. This was connected to a Dionex UltiMate 3000 RSLC system (Thermo Fisher Scientific) using a ZIC-pHILIC column (150 mm × 4.6 mm, 5 µm column, Merck Sequant). The column was maintained at 30 °C and samples were eluted with a linear gradient (20 mM ammonium carbonate in water, solution A and acetonitrile,

solution B) over 26 min at a flow rate of 0.3 mL min$^{-1}$ as follows: 0–20 min 20–80% solution A, 15–17 min 95% solution A, 17–26 min 20% solution A. The injection volume was 10 μL and samples were maintained at 5 °C prior to injection. Mass spectrometry data was processed using a combination of XCMS 3.2.0[78] and MZMatch.R 1.0–4III[79]. Briefly, data were converted from Thermo proprietary raw files to the open format mzXML. Unique signals were extracted using the centwave algorithm and matched across biological replicates based on mass-to-charge ratio and retention time. These grouped peaks were then filtered based on relative standard deviation and combined into a single file. The combined sets were then filtered on signal-to-noise score, minimum intensity and minimum detections. The final peak set was then gap-filled and converted to text for use with IDEOM v18[80]. Putative metabolite identification corresponds for the most part to Metabolite Standards Initiative (MSI) level 2 (mass only), whereas metabolites matching in retention time to an included standard correspond to level 1, as indicated in the underlying data. Peaks having an area with root squared deviation across pooled samples > 50% were excluded, as were those with a retention time < 4 min (due to poor resolution). Comparative analysis was, as described[81], using a Pearson correlation, with the first variable being the signal intensities of the identified metabolites and the second variable being 1 and 0 for the SC83288 treatment group (consisting of eight independent biological replicates) and the non-treated control group (consisting of eight independent biological replicates), respectively. The eight independent biological replicates were collected in two batches of four over the course of one year. All eight independent biological replicates were analyzed simultaneously on the same machine and by the same operator. The resulting Pearson coefficient was converted into a $p$-value. Subsequently, a log-transformed $p$-value adjustment was performed by computing the negative logarithm of the ratio between a metabolite-specific $p$-value and the mean of all $p$-values, resulting in the LOD score. The fold change for each metabolite was calculated by dividing the mean signal intensity obtained for the treated group by the mean signal intensity of the untreated control.

### Nuclear isolation and DNA methylation

Highly synchronized cultures of the *P. falciparum* line 3D7 at the trophozoite stage were incubated with 80 nM SC83288 for 4 h under regular cell culture conditions. An untreated control culture was investigated in parallel. Cells were harvested in PBS, lysed with 0.01% saponin inPBS and washed with PBS. Nuclear proteins were extracted as described[82], with some modifications. Parasite pellets were resuspended in lysis buffer (20 mM HEPES-KOH, 10 mM KCl, 1 mM EDTA, 1 mM DTT, 0.5 % NP40, 1× HALT protease inhibitor (Thermo Fisher Scientific) pH 7.8), and incubated 5 min on ice. Extracts were centrifuged at 2500 × $g$ for 5 min at 4 °C, washed twice with PBS and resuspended in extraction buffer (20 mM HEPES_KOH, 800 mM KCl, 1 mM EDTA, 1 mM DTT, 1× HALT protease inhibitor (Thermo Fisher Scientific) pH 7.8) for 30 min while rotating at 4 °C. Nuclear extracts were cleared by 30 min of centrifugation at 13,000 × $g$ at 4 °C and diluted 1:1 volume with dilution buffer (20 mM HEPES-KOH, 1 mM EDTA, 1 mM DTT, 30% glycerol, pH 7.8). Protein content was quantified by Bradford assay and DNMT activity was measured using the EpiQuik™ DNA methyltransferase activity kit (Epigentek cat # P-3004, fluorometric) following the manufacturer's instructions. Activity was measured every minute for 20 min. Each sample was analyzed in technical duplicates.

### In vitro PfDNMT2 assays

In vitro PfDNMT2 assays were performed as described[28]. PfDNMT2 was expressed episomally as a GFP-tagged fusion protein in *P. falciparum*. -10$^9$ trophozoite stage parasites were isolated with 0.1% (v/v) saponin and lysed in 5 volumes of the lysis buffer (300 mM NaCl, 20 mM HEPES pH7.9, 20% v/v glycerol, 2 mM MgCl2, 0.2 mM EDTA, 0.1% NP40,

0.5 mM DTT) containing a protease inhibitor cocktail (Roche). The parasites were mechanically lysed using a homogenizer with 80–100 strokes. The supernatant was collected and incubated with 100 μl beads slurry (GFP-Trap®, Cat# gta-20, RRID: AB_2631357, ChromoTek) for 4 h at 4 °C with gentle rotation to pulldown GFP-tagged PfDNMT2. Subsequently, the beads were washed with washing buffer (300 mM NaCl, 20 mM HEPES, pH 7.9, 2 mM MgCl2, 0.2 mM EDTA, 0.1% NP40, 0.5 mM DTT) three times. The beads were incubated with Pierce™ Gentle Elution Buffer (ThermoFisher Cat# 21027) for 10 min at room temperature to directly elute the immune complex for downstream enzymatic assays. Western blot analysis following SDS denaturation of immunoprecipitated eluate was performed using the anti-Protein C antibodies (1:1000, GenScript) to confirm the presence of the PfDNMT2-GFP enzyme. The DNMT activity of the eluate was measured using the EpiQuik™ DNA Methyltransferase Activity Ultra Kit (Fluorometric) (EpigenTek) following the manufacturer's instructions. Assays were performed in triplicate by incubating 0.1 μg of protein eluate from -1 ×10$^8$ trophozoite stage PfDNMT2::GFP parasite-infected red blood cells for 75 min. SC83288 was added to the reaction at final concentrations of 0, 0.1, 0.25, 0.5, 1, 5, 10, and 25 μM. A buffer-only blank was used for background subtraction. DNMT activity was expressed in relative fluorescence units per hour and per mg of proteins (RFU h$^{-1}$ mg$^{-1}$).

### In vivo RNA N6-adenosine methylation assay

Highly synchronized cultures of the *P. falciparum* line 3D7 at the trophozoite stage were incubated in the presence or absence of 80 nM SC83288 for 4 h under regular cell culture conditions. Cells were harvested in PBS, lysed with 0.01% saponin in PBS and washed with PBS. Cell pellets were resuspended in Trizol (Thermo Fisher Scientific) and kept at −80 °C until the RNA was extracted. RNA extraction was done according to the manufacturer's instructions. RNA quality was verified by electrophoresis on formaldehyde/agarose gels. m6A RNA methylation was quantified using the EpiQuik™ RNA methylation kit (Epigentek cat # P-9005, colorimetric). Samples were analyzed in technical duplicates.

### Cellular thermal shift assay

Intact-cell isothermal dose–response cellular thermal shift assay (ITDR-CETSA) was performed as described[31,32]. 150 mL highly synchronized trophozoite-stage culture (25 ± 3 hpi) of around 10% parasitemia was subjected to MACS purification[58] under sterile conditions and using a prewarmed culture medium for elution. After centrifugation at 1200 × $g$ for 5 min at room temperature, the supernatant was replaced with fresh pre-warmed medium, and the cells were allowed to recover for 1 h under in vitro culture conditions. The number of cells recovered and the level of parasitemia were measured using a hemocytometer and a Giemsa-stained thin blood smear, respectively. After the recovery step, the culture was centrifuged at 1200 × $g$ for 5 min at room temperature. The supernatant was discarded, and the concentration was adjusted to 10 × 10$^6$ cells mL$^{-1}$. Subsequently, 4 mL aliquots of the cell suspension were treated with serial dilutions of SC83288 (covering a range from 7 pM to 3 μM as indicated, and a DMSO control). Samples were incubated for 1 h at 37 °C. Each sample was centrifuged at 1200 g for 5 min at room temperature. The supernatant was removed before resuspension in 100 mL of room-temperature PBS. A 50 μL aliquot of each sample was added to a separate well of two different 96-well PCR plates (20 ×10$^6$ cells per well). The plates were then placed in two thermocyclers, one with the 37 °C control temperature and the other with a denaturing condition (e.g. 50, 55, or 60 °C) and subjected to the thermal challenge for 3 min. Finally, 50 μL of 2× lysis buffer (100 mM HEPES, 10 mM β-glycerophosphate, 0.2 mM sodium orthovanadate, 20 mM magnesium chloride, 2 μL mL$^{-1}$ EDTA-free protease inhibitor cocktail and 2 μL mL$^{-1}$ 1 M Tris-(2-carboxyethyl)-phosphine hydrochloride (TCEP). Samples

were lysed by 3 cycles of liquid $N_2$ flash-freeze thawing, followed by mechanical sheering with 31 g needle and soluble protein isolation through centrifugation ($20,000 \times g$, 4 °C, 20 min). Soluble phases were then reduced, alkylated and digested with trypsin and lysC, labeled with TMT10 isobaric mass tags, pulled together and desalted and analyzed by quantitative mass spectrometry.

## PfSAHH enzymatic assay

PfSAHH (PF3D7_0520900) was amplified from 3D7 gDNA using the following primers: SAHH-pET-for-2 and SAHH-pET-rev-2 (Supplementary Table 8). The PCR product was subsequently cloned into the pET 151 D-TOPO vector (Thermo Fisher Scientific), expressing a PfSAHH-His fusion protein. The sequence was verified and the PfSAHH-HIS tag coding sequence was expressed in *E. coli* BL21 (DE). Precultures were grown at 37 °C overnight and diluted in fresh LB medium to OD of 0.2 at an absorption of 600 nm. At an OD of 0.4, 0.4 mM IPTG was added and the culture was incubated at 30 °C for 12 h. The bacterial pellet was collected by centrifugation and cells were lysed on ice for 10 min in buffer A (20 mM potassium phosphate, 0.2 M NaCl, 10 mM imidazole, pH 7.5) containing 1 mM PMSF, 1 mg ml$^{-1}$ lysozyme, 0.1 mg ml$^{-1}$ DNAse, 5 mM $MgCl_2$ and 0.02% NP40, followed by $2 \times 30$ s sonication. Extracts were centrifuged at $10,000 \times g$ for 20 min and the expressed PfSAHH was purified by affinity chromatography using a HisPur cobalt spin column (Thermo Fisher Scientific) equilibrated in buffer A. After washing with buffer A, proteins were eluted with buffer B (20 mM NaPi, 0.2 M NaCl, 400 mM imidazole, pH 7.5). Fractions containing PfSAHH protein were dialyzed overnight at 4 °C in buffer C (20 mM potassium phosphate, 1 mM EDTA, 10% glycerol pH 8) and concentrated using a centricon 30 spin column (Amicon). The PfSAHH enzymatic activity was spectrophotometrically determined using a previously published protocol[83,84], with some modification. The enzyme reaction mixture contained 50 μM S-adenosylhomocysteine, 4 U adenosine deaminase, 250 mM 5,5-Dithio-bis-(2-nitrobenzoic acid) (DNTB), 1 mM DADin, and 50 mM potassium phosphate, pH 7.5. Reactions were started by the addition of 5–10 μg purified PfSAHH and incubated for 15 min at 37 °C. Enzymatic activity was spectrophotometrically detected at an absorption of 412 nm. The inhibitory activity of SC83288 on PfSAHH was evaluated by preincubating the PfSAHH with the indicated concentrations of SC83288 for 10 min at 37 °C. Controls included reactions without enzyme and with the recombinant human SAHH (prospec, # enz-532). All reactions were done by technical duplicates.

## Modeling

The crystallographic structures of the SERA5 pseudo-zymogen and the SAHH were retrieved from the Protein Data Bank (PDB 6×44 and PDB 1V8B)[36,38]. Fumarase, isocitrate dehydrogenase and PfATP6 did not have experimental structures available. Models for fumarase and isocitrate dehydrogenase were built using homology modeling, the web server SWISS-MODEL[85] and the crystal structures from PDB 6UP9 (fumarate hydratase from *Leishmania major*) and 5I95 (human mitochondrial isocitrate dehydrogenase) as templates, respectively[85–87]. Residues 254–310 in fumarase did not align with any region in the sequence of the template structure used, and were modeled as a disordered region. The sequence identity between the template and fumarase or isocitrate dehydrogenase were 56% and 63%, respectively. The use of homology modeling, instead of AlphaFold2[88], facilitated the placement of metal centers (fumarase), ions and co-factors ($Mg^{2+}$ and NADPH for isocitrate dehydrogenase) in the modeled proteins.

Four models were built for PfATP6 in order to represent the four different conformations known for the protein chosen as a template, the calcium ATPase of skeletal muscle sarcoplasmic reticulum (SERCA1a) from *Oryctolagus cuniculus*[18,89]. The four states (E1, E1 - P·ADP, E2 - P and E2) represent different conformational states during the calcium transport cycle[89]. The models were built using homology modeling, the web server SWISS-MODEL[85] and the crystal structures

from PDB 1SU4 (state E1)[90], 2ZBD (state E1 - P·ADP)[91], 2ZBG (state E2 - P)[92] and 2AGV (state E2)[93] as templates. The sequence identity between the template and PfATP6 was 48%. In the sequence alignments between the templates and PfATP6, it was noted that residues 408–497 and 562–674 in PfATP6 did not align with any region in the sequences of the template structures used. Therefore, only residues 1–365 and 802–1217 (which include the complete transmembrane region, and part of the region outside the membrane) were included in the final models used for docking and simulations. Na$^+$ present in the template structures of PDB 1SU4 and 2AGV were kept in the modeled structures. The use of homology modeling, instead of AlphaFold2[88], enabled the modeling of PfATP6 in different conformations. Structures of the mutant F972Y of PfATP6 were modeled replacing Phe by Tyr in the models of the wild type PfATP6 using Pymol[94]. A quality assessment of the final models is provided in Supplementary Table 1. The models are provided as supporting information.

## Docking

The protein structures were processed using the Protein Preparation tool from Maestro (Schrödinger) at pH 7.4. The canonical SMILES code of the ligand (SC83288) was obtained from the PubChem database[95] and used as input in Maestro to generate the 3D structure of the molecule. The ligand was protonated at the pH of 7.4 using Epik Classik (Schrödinger), and presented a net charge of +1 e. The ligand was then prepared and energy minimized using the LigPrep tool in Maestro (Schrödinger).

Using the prepared protein and ligand structures, docking was performed by using AutoDock Vina[96] (exhaustiveness level of 32 for SAHH and 8 for the other proteins) as well as the Glide tool from Maestro (Schrödinger)[97]. For SERA5, in both programs, docking was performed locally, building the grid around Ser596 (which is catalytically inactive in SERA5 but catalytically active in closely related enzymes), as well as globally, building the grid around the entire protein. For the other proteins, only local docking was performed, since the catalytic site was known from experimental structures. Docking with Vina was not performed for fumarase due to the lack of parameters to describe the metal center. Docking for SAHH and isocitrate dehydrogenase was performed in the presence and absence of the co-factors. As the grid center, we chose residues close to the substrate or product in the experimental structure, or in the experimental structure used as a template to model the protein (residues Glu200, Asn125 and Thr580 for SAHH, isocitrate dehydrogenase and fumarase, respectively). In PfATP6, we chose residue Phe972 as the grid center due to experimental evidence that the mutation F972Y perturbs transport of SC83288 by PfATP6.

Bond rotations were allowed in the ligand, while the protein structure was kept rigid. Only the top pose from each procedure (the one with the lowest docking score) was selected for further analysis. Details of the docking protocols (grid center and grid size) can be found in Supplementary Table 9.

We also utilized Boltz version 2.2.0 tool[98] via Tamarind Bio's API (https://www.tamarind.bio) to dock SC83288 to PfATP6. As Boltz-2 is a co-folding tool, we used it to represent potential protein conformational changes due to interactions with the inhibitor. We used as input the primary sequence of PfATP6 (Uniprot ID: Q76NN8), in the wild type or mutated (F972Y) form, and the SMILES code of the inhibitor. Only the top protein-ligand complex (the one with the highest confidence score) was selected for further analysis.

## Molecular dynamics simulations

After docking, MD simulations of selected protein-ligand complexes were performed using GROMACS V2024.2[99], the AMBER99SB-ILDN force field to describe the protein and the general Amber force field (GAFF) to describe SC83288. Partial charges for the atoms of SC83288 were obtained from quantum mechanical calculations (single point

calculation) performed by Gaussian 16 Rev. C.01 using Hartree−Fock and the 6-31G* basis set and then converted to gromacs file formats using ACPYPE[100]. For selected complexes, 3 replicates of 300 ns were performed to check whether the ligand would show a stable binding mode when interacting with the protein. The protein-ligand complexes were centered in a cubic box with at least 1 nm distance to the edges of the simulation box. The box was then filled with TIP3P water molecules[101]. Then, using the steepest descent algorithm, the system was energy minimized until the maximum force was less than 10 kJ/mol.nm. Subsequently, the temperature and the pressure of the systems were equilibrated at 310 K and 1 bar using the Berendsen thermostat and barostat[102]. After temperature and pressure equilibration, 4 steps of 500 ps simulations were performed to reduce the positional restraints over the backbone atoms of the protein (500, 100, 10 kJ/mol.nm$^2$). A time step of 2 fs was used for the equilibration runs and during the 300 ns production runs. For production runs, Nose-Hoover thermostat with a time constant of 0.1 ps was used for temperature coupling[103,104] and Parrinello-Rahman barostat with a time constant of 5 ps was used for the pressure coupling[105,106]. The LINCS algorithm was used to constraint the covalent bonds to hydrogens and the SETTLE algorithm was used to constraint the bond lengths for the solvent. The Particle Mesh Ewald (PME) method was used to calculate electrostatic forces with a Fourier grid spacing of 1.2 Å, real-space cutoff of 1.2 nm and PME order of four. The cutoff for calculating the Van-der-Waals forces was set to 1.2 nm.

For simulations with PfATP6, which is a membrane protein, the systems were prepared by the CharmmGUI web server[107]. A total of six systems were simulated: wild type and F972Y mutant of PfATP6 in the apo form, wild type and F972Y mutant of PfATP6 (E1 state) bound to SC83288, with docked poses generated by AutoDock Vina, and wild type and F972Y mutant of PfATP6 bound to SC83288, with docked poses generated by Boltz-2. The lipid bilayer was composed of palmitoyl-oleoyl-phosphatidylcholine (POPC) and palmitoyl-oleoyl-phosphatidylethanolamine (POPE) with a 2:1 ratio, respectively, in order to reproduce the lipid composition of the endoplasmic reticulum[108]. The protein-ligand complexes were put in the center of a rectangular box with a system size of 139 Å along the X and Y axis. Then the box was filled with TIP3P water molecules[101], and a concentration of 150 mM of NaCl was added to the solvent to neutralize the system. The parameters from AMBER FF19SB[109] and the AMBER Lipid21 force fields[110] were used to describe the protein and the lipid molecules, respectively. For the ligand, the same force field and parameters for the MD simulations of soluble proteins mentioned previously were used. The simulations were performed using the GROMACS V2024.2 software[99]. The final prepared system was energy minimized utilizing the steepest descend method with a maximum number of 5000 steps while using positional restraints on the heavy atoms of the protein (4000 kJ/mol/nm$^2$ over backbone atoms, 2000 kJ/mol/nm$^2$ over side chain atoms) and membrane (1000 kJ/mol/nm$^2$). Next, for the equilibration, we used the default six-step equilibration suggested by CharmmGUI before moving to the production runs. In the two first steps, the system was heated to 310 K using the V-rescale thermostat[111]. The positional restraints were reduced to 2000, 1000 and 400 kJ/mol/nm$^2$ over backbone atoms of the protein, side chain atoms and the lipid heavy atoms, respectively, during these two steps. Then, in the next steps of the equilibration, a pressure of 1 bar was maintained using the C-rescale barostat[112] while the positional restraints over backbone atoms of the protein and side chain atoms were further reduced to 1000 and 500 kJ/mol/nm$^2$, respectively. In the last steps of the equilibration, the positional restraints were gradually reduced to zero. For the production runs, we performed 3 replicas for each system with a duration of 300 ns and a time step of 2 fs. The same thermostat and barostat were used with a time constant of 1 ps. The barostat was set to semiisotropic with a compressibility of 4.5 $10^{-5}$ bar$^{-1}$. The LINCS algorithm[113] was used to constraint the covalent bonds to hydrogen atoms in the solute and the

bond lengths for the solvent were constrained using the SETTLE algorithm[114]. The Particle mesh Ewald (PME) method[115,116] was used to calculate the electrostatic forces with a real-space cutoff of 1.2 nm, PME order of four, and a Fourier grid spacing of 1.2 Å, and the Van-der-Waals forces were computed using a cutoff of 1.2 nm.

## In silico data analysis

Backbone root mean square deviation (RMSD) values between structures predicted by Boltz-2 and the modeled structures of PfATP6, presented in Supplementary Table 3, were calculated by MDAnalysis[117]. The SASA values of the docked poses were calculated by the FreeSASA software[118]. Other analyses for the MD simulations, such as RMSD, root mean square fluctuation (RMSF), and distances, were carried out by Gromacs Utilities[99].

## Statistics and reproducibility

Statistical analyses were performed using Sigma Plot (v.16.0, Systat) software and the statistical test used is indicated in the main text and/or figure legends. $p$ values < 0.05 were considered significant, if not indicated otherwise. The number of biologically independent samples is indicated in the main text and/or the figure legends. If data from biologically independent samples were averaged, then the mean ± SEM is shown. Biologically independent samples are defined as experiments conducted with different source materials. The number of technical replicates is indicated in the respective Methods section. Box plots show the median (black line), the mean (red line), and the 25 and 75% quartile range. The error bars above and below the box indicate the 90th and 10th percentiles.

## Illustrations

Schematic illustrations were prepared using BioRender; all such figures are licensed under CC-BY 4.0.

## Reporting summary

Further information on research design is available in the Nature Portfolio Reporting Summary linked to this article.

## Data availability

All data supporting the findings of this study are available within the article and its Supplementary Information files. *P. falciparum* lines are available upon reasonable request and contingent upon the recipient's provision of appropriate institutional approvals and legally required permits for transport, handling, and storage of genetically modified organisms. Source data are provided with this paper. The in silico datasets generated in this study have been deposited in the zenodo database under accession code 17361603, https://zenodo.org/records/17361603. RNA-seq sequencing data files have been deposited in the Sequence Read Archive (SRA) repository under accession code PRJNA1417599, https://www.ncbi.nlm.nih.gov/sra/PRJNA1417599. ITDR-CETSA proteomics data: The thermostability profiles of identified human and *P. falciparum* proteins have been deposited in the figshare data repository under accession code 30620744 [119]. The raw MS data are available at JPOSTrepo, a member of ProteomeExchange Consortium, under accession numbers JPST004409 and PXD074497 (https://repository.jpostdb.org/entry/JPST004409 and https://proteomecentral.proteomexchange.org/?pxid=PXD074497). The metabolomics data have been deposited to MetaboLights repository with the study identifier MTBLS13911, https://www.ebi.ac.uk/metabolights/MTBLS13911. Source data are provided with this paper.

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

## Acknowledgements

We thank Marina Müller, Atdhe Kernaja, and Stefan Prior for excellent technical assistance and Stefanie Fehler for the synthesis of SC106879. The work was partly funded by the Bundesministerium für Bildung und Forschung under the portfolio of the German Centre for Infectious Research (DZIF), project TTU 03.803) (M.L.), the National Science and Technology Development Agency, Thailand, project P1850116 (S.K.), the Deutsche Forschungsgemeinschaft (DFG) under Germany's Excellence Strategy – EXC 2008/1-390540038 – UniSysCat (F.S. and A.N.A.), the state of Baden-Württemberg through bwHPC (S.D.B. and M.L.), and the German Research Foundation (DFG) through grant INST 35/1597-1 FUGG (S.D.B.). J.M.D. acknowledges support by an NTU President's Postdoctoral Fellowship (NTU-PPF-2019).

## Author contributions

M.L., C.P.S., M.P.B., S.K., A.N.A., and Z.B. designed the study; C.P.S., M.D., R.V.C., M.K., M.H., T.S., F.S., S.D.B., A.N.A., A.B.L., J.M., J.M.D. performed the experiments; C.P.S., M.D., R.V.C., M.K., M.H., T.S., S.K., F.S., S.D.B., A.N.A., A.B.L., J.M., C.R., J.M.D., Z.B., M.P.B., and M.L. analyzed results; M.L. wrote the manuscript, with the help of M.P.B. All authors participated in discussion and manuscript editing.

## Funding

## Competing interests

The authors declare no competing interests.
