## [Peer Review file · Nature Communications]

Mechanisms of PfDNMT2 inhibition and PfATP6-mediated resistance to the antimalarial candidate SC83288 in *Plasmodium falciparum*

Corresponding Author: Professor Michael Lanzer

Version 0:

Reviewer comments:

Reviewer #1

(Remarks to the Author)

Sanchez et al. explored the mode of action and the mechanism of resistance of a novel anti-P. falciparum compound SC83288. Previous work has suggested the association of PfATP6 mutations with resistance to SC83288. Through CRISPR mutagenesis, yeast complementation, live-cell Ca^{2+} imaging, click-chemistry localization, transcriptomics, metabolomics, and CETSA, their work suggested SC83288 being a competitive substrate rather than an inhibitor of PfATP6. In addition, SC83288 showed a polypharmacological activity of disrupting DNA replication, transcription, methylation, and parasite egress. These results demonstrated a novel mode of action of SC83288 in inducing a substantial fitness cost in parasites and its potential for malaria treatment. Although the study found a potentially novel drug resistance mechanism and multi-target effects of SC83288, however, the insufficient depth of mechanism interpretation, the lack of validation of key targets, the weak clinical translation data and the limitations of methodology significantly affected the reliability of the conclusion. Therefore, I recommend a major revision of the manuscript. The comments are listed below:

1. ITDR-CETSA analysis: The authors should explain why they chose an in-cell rather than a lysate CETSA experiment. In the figure legend of Fig.7, the hits of ITDR-CETSA should be in Supplementary Table 6, not Supplementary Table 4. Given the concentration used, the authors are advised to present the MDT concentration in the unit of "nM" but not "uM". In addition, the authors should explicitly indicate which proteins meet each criterion (criteria 1, criteria 2, and spectral confidence). In the main manuscript, only 4 out of 28 CETSA hits (SERA5, SAHH, and two TCA enzymes) were studied as plausible target candidates in detail. However, there is a lack of evidence for direct binding or inhibition. Given the negative result in the docking and MD simulations of SC83288 to the extended cleft domain of SERA5 and PfSAHH enzymatic assays, the primary target remains elusive. This unfortunately impacts the quality of this study.

2. Metabolomics analysis: In Fig.6A, is the ratio of AdoMet to SAH increased? Or SAH is a typo for SAM? In a volcano plot, each dot should typically represent one metabolite. There was a mismatch between this figure to the associated text. The authors should specify all the dysregulated metabolites and expand the relevant result description (which only talks about MTA, adenine, and phosphoethanolamine).

3. Despite CETSA "hits" (SERA5, SAHH, TCA enzymes), direct binding was not confirmed (e.g. PfSAHH enzymatic assays negative). A bona fide primary target or high-affinity binding partner remains elusive.

4. Improve Data Clarity. Simplify and quantitatively support complex figures; provide co-localization statistics for click-chemistry (Figure 4)

5. Although SC83288 is reported to disrupt DNA replication, transcription, and metabolism, no conditional knockdown/complementation or co-treatment experiments were performed to verify which pathways are true on-target effects. Functional studies linking specific candidate targets to the observed phenotypes are required.

6. Broaden Impact Discussion. Frame the ER-sequestration mechanism in the context of other resistance pathways, and discuss translational implications for next-generation antimalarials.

7. The similarities and differences with the artemisinin resistance mechanism (Kelch13 mutation) were not deeply compared, which weakened the uniqueness argument of it as a "new mechanism drug".

8. Although ITDR-CETSA identified potential targets such as SERA5 and SAHH, the necessity of these proteins in drug effects was not verified through functional experiments (such as target-specific inhibitors and gene editing).

Reviewer #2

(Remarks to the Author)

This manuscript describes characterization of mechanisms of resistance of malaria parasites to SC83288, a compound whose excellent antimalarial and drug-like properties and mechanism of resistance (due to mutations in the Ca-ATPase PfATP6) were described in 2017. This new report adds an in depth characterization of resistance mechanisms, with the interesting observation that a PfATP6 mutation that mediates high level resistance does not alter transport of SC83288 into the infected erythrocyte, but rather alters its intracellular distribution, with concentration in the ER, presumably sequestering the compound apart from its site of action. Further, the manuscript explores other effects of SC83288, with demonstration of impacts on various processes via quantification of cellular DNA, transcriptomic and metabolomic studies, and thermal shift assays. These assays led to interesting results, with intriguing suggestions for processes and proteins potentially impacted by SC83288, but it is difficult to distinguish direct effects of the compound from secondary responses in parasites inhibited in some manner by the compound. There was certainly no "smoking gun", as different approaches identified different potential targets, and each of the 4 proteins that made it through multiple filters after thermal shift assays was not particularly supported as a target by additional considerations (although the authors are to be praised for providing descriptions of these careful considerations). Overall, the report describes a large, well-done series of experiments attempting to better describe mechanisms of action and resistance to a promising drug lead. The results add importantly to our understanding of the action of and resistance to SC83288, but, as is often the case with compounds with high barriers to resistance, the mechanism of action of the compound and the full mechanism of resistance remain incompletely understood. Some changes to the manuscript are suggested, as follows.

1) Line 39. The term "polypharmacological" is not very familiar, and it seems synonymous in this context with "multiple drug targets", so it is not clear why both terms are used here.

2) Line 51. It should be noted that these are the numbers estimated, not reported, e.g. "...estimated in 2023 alone."

3) Line 54. Life cycle, not life style.

4) Line 55. Omit "chilling" or change to "chills, fevers...."

5) Line 59. This statement is inaccurate, as the most widely used antimalarial in the world (artemether-lumefantrine) includes lumefantrine, which is not in any of these categories (unless it is considered "quinoline-like", though lacking a quinoline ring?).

6) Line 65. "uncomplicated falciparum malaria"

7) Line 67. It is not clear what is meant by "accessibility". Was the intended meaning "availability"?

8) Line 87-99. This summary is helpful, but it appears to be fully a summary of the results of this report. If that is the case, it belongs in the Discussion, not the Introduction. Inclusion of 1-2 sentences briefly explaining the approach of this study and perhaps a "teaser" regarding results would be appropriate, but this 13 line summary seems excessive, and also a bit confusing, as readers will not be sure whether these data are background from prior studies or the results of this study. Of note, in the first paragraph of the Discussion, where one expects to see a summary of new results, the authors mostly summarized previously published results (see comment 13 below). Reorganization will be helpful and will make it easier for readers to appreciate what is new in this report.

9) Line 127. It will be slightly clearer to describe the assay before describing the results of the assay, i.e. to place the last sentence earlier in the paragraph.

10) Lines 204 and 390. The short descriptions of the mechanism of resistance to artemisinins is a bit misleading. Although this is only peripherally relevant to this report, it would be helpful to note that artemisinin partial resistance seems primarily mediated by mutations in K13 that alter hemoglobin transport to the food vacuole, as nicely demonstrated by Spielmann's group (Birnbauer, et al., Science 2020).

11) Line 205. Chloroquine is a quinoline, not a quinolone.

12) Line 228-230. It is unclear how Fig. 5A demonstrates a block in RBC egress. A photomicrograph seems a better way of

demonstrating this, and in addition it would provide general information by showing the morphology of treated parasites. Has such a block been demonstrated morphologically in older reports on SC83288?

13) Line 377. The first paragraph of the Discussion is nearly completely a review of old data, mostly already provided in the Introduction. It would be more value to use this paragraph to summarize the new data provided in this report and to explain how that adds to our understanding. In fact, it is not convincing that mutations in PfATP6 are unlikely to emerge in human infections; a more reasonable interpretation seems to be that selection of resistance to SC83288 occurs less readily compared to some other antimalarials, but one can't be sure how readily resistance will emerge if the compound is widely used to treat malaria.

14) Line 428-431. This sentence has grammatical problems. Since each approach yielded different "targets" for SC83288, it seems more likely that some of these approaches did not actually identify a target rather than to assume that each identified different valid targets. Overall, although the varied data are of interest and worth reporting (and discussing in subsequent paragraphs), it does not clearly follow that, because it has been difficult to identify a target, the compound has a polypharmacological mechanism.

15) Line 475-491. This discussion addresses potential interaction between SC83288 and SERA5 leading to a block in parasite egress from the erythrocyte. This is interesting, but key information is not mentioned in the discussion. Does SC83288 cause an obvious block in egress, as can be studied simply by looking at treated parasites (by Giemsa-stained smear) over the course of the life cycle? This point was also brought up above. If information is already available on the morphology of SC83288-treated parasites over the course of the life cycle, this information should be included in the discussion suggesting specific action against parasite egress. If not, a simple experiment showing the morphology of treated parasites over the course of the life cycle should be added.

Reviewer #3

(Remarks to the Author)

The authors have previously conducted an amicarbalide-based medicinal chemistry program for development of novel antimalarials, and screening of these chemicals showed that two compounds with optimized pharmacological and antiparasitic properties. Particularly, Comprehensive pre-clinical evaluation revealed that the compound SC83288 harbor potential as a clinical development candidates for malaria therapy. Interestingly, they also found that the sarco/endoplasmic reticulum Ca²⁺ transporting PfATP6 might be a putative drug resistant marker gene to this compound. Therefore, understanding the underlying mechanisms of parasite-killing action and drug resistance is crucial for the further development of this compound as a new-generation antimalarial. In this study, the authors have comprehensively investigated the pharmacological activity and drug targets via combinations of various approaches. They found that SC83288 is a substrate for PfATP6, in which the mutations may sequester the compound into the compartment of ER thereby blocking the potential interaction between the drug and its targets. This part is convincing. It is notable that the mutations of PfATP6 found in the in vitro selected drug-resistant clones may result in heavy fitness-cost, and are not previously identified in field isolates. This finding further confer this compound high potential as new drug candidate for malaria treatment. In general, this work is interesting and important for development of new-generation antimalarials, which is urgently required when the artemisinin-based therapies (ACTs) are now encountering challenge of drug resistance including artemisinin itself. While the phenotypic data are solid and convincing, the mechanistic interpretations are relatively weak and elusive. Though polypharmacological activity and multiple drug targets may be preferable for drug development against malaria parasites, but it is difficult to justify this conclusion with the data presented in the current version of manuscript. Here are some concerns still need to be addressed by the authors.

Major concerns:

1. the F972Y mutation of PfATP6 was identified in SC83288 resistant clones selected with lower drug concentration in vitro, which is also validated by gene editing of this gene. As shown in the Fig. 2A, this site located in the fifth transmembrane domain. While no evidence of the direct interaction between SC83288 and PfATP6 is available, it is unclear about the role of this position in PfATP6's function. What's mechanism of this mutant protein in enhancing the local distribution of SC83288 in the compartment of ER? What's the mechanism of heavy fitness cost conferred by this mutation in PfATP6?
2. the authors claim that the SC83288-induced rise in cytosolic free Ca²⁺ would be a result of a competition reaction between Ca²⁺ and this compound. Though they tested this phenotype in *S. cerevisiae* in vivo and in vitro and the results are convincing, what's the possible mechanism of the competition between them, as questioned in the first concern that this compound does not directly interact with PfATP6.
3. For the differential transcriptome analysis of parasites treated with SC83288 and control, the PfATP6-mutant line, e.g., E4, should be included. This may help validate the drug-responsive genes or pathways, and also interpret the heavy fitness cost due to the site mutation in PfATP6.
4. SC83288 treatment showed significant alteration of enzymes involved in DNA and RNA methylation. For the DNA methyltransferase PfDNMT2, Fig. 6D provided solid evidence of inhibition of this enzyme by SC83288. However, it is likely not clear how this drug acts on the RNA m⁶A writer complex. It is known that the two key writers of mRNA m⁶A modification are PfMT-A70 and PfMT-A70.2 corresponding to the conserved METLL3 and METLL14 in model eukaryotes, but not WTAP?
5. How to interpret the observation that the pronounced changes in RNA levels occurred independent of the SC83288 dose (Fig. 5C)? Does it indicate that a majority of differential expression came from indirect effect by drug treatment, or represents a result of cell death pathways? Though the main stage with potent inhibitory impact of SC83288 was observed in thophozoites, a timecourse of transcriptomic dynamics with lower drug concentrations, eg. pmol grade, may be helpful.
6. It is difficult to understand the result of ITDR-CETSA analysis. This work points to four target candidates of SC83288, but

no proof was shown for the direct interaction between this compound and each of them. Moreover, Fumarase and isocitrate dehydrogenase are not essential for the asexual development of parasites. Perhaps other methods such as Activity-based Protein Profiling (ABPP) may produce different result. But, at least the authors should demonstrate that the four target candidates do not localize in the ER by IFA or other assays, otherwise the proposed mechanism of PfATP6-mediated resistance will not work in these cases.

7. Both differential transcriptomics and metabolism analysis point to cellular methylation involving DNA and RNA methyltransferases. I believe this metabolic pathway would be a key information responsible for cell death under SC83288 treatment. However, no any experiment evidence was shown in the manuscript, eg. gene manipulation of any m5C/m6A writer or other enzymes of methyl metabolism to rescue the phenotype?

Minor concerns

1. in the previous paper published in NC (Pegoraro et al. 2017), it looks that the F972Y mutation was found in SC81458 instead of SC83288?
2. in Fig. 2A, the F972Y position looks not consistent with the previous version shown in the Fig. S11C published in NC 2017.
3. line 225: six should be 6 h
4. The duration of drug treatment varied in transcriptomic analysis (4.5 h) and metabolomic analysis (12 h). What's the rationality of these conditions?
5. Why Pyrimethamine was shown in Fig. 1 and also used in the analysis of Fig 5B? I saw the structure of SC83288 likely from amicarbalide and sulfadoxine, but not Pyrimethamine?
6. The majority data of this manuscript likely focus on the pharmacological action or molecular targets instead of ER-enrichment-based resistance. The title may be modified to reflect the main part of this work.

Reviewer #4

(Remarks to the Author)

Version 1:

Reviewer comments:

Reviewer #2

(Remarks to the Author)

The authors made substantial changes to the manuscript based on reviewer comments, and in fact these changes considerably improved the manuscript. It now has a clearer message, and the argument that SC83288 most likely acts via inhibition of PfDNMT2 is well-supported by the exhaustive data provided. Although the modes of action and resistance of the compound are complex, and not fully explained by all available data, this large manuscript makes major advances in explaining drug action. The authors are praised for comprehensive responses to reviews and a very clear and comprehensive (albeit very long) manuscript. No additional changes are suggested.

Reviewer #3

(Remarks to the Author)

The authors have addressed part of the previous concerns, and the manuscript has been significantly improved. However, three major issues in my previous review comments remain unresolved in the revised version, which weaken the novelty and integrity of this work.

1. The transcriptome analysis remains incomplete without data from the PfATP6-mutant line (e.g., E4). This comparison is needed to confirm the specificity of the drug-responsive signature and to investigate the fitness cost of the mutation. I urge the authors to provide this data.
2. The claim that SC83288 impacts the RNA m6A writer complex is currently weak and lacks direct experimental support. The authors have only discussed the involved enzymes (PfMT-A70/PfMT-A70.2) without providing any functional evidence to demonstrate how the drug alters m6A methylation.
3. The claim that SAHH is a direct target of SC83288 relies heavily on computational docking, which is not conclusive evidence. To validate this interaction, direct experimental proof is required. We urge the authors to perform in vitro binding assays (e.g., SPR, ABPP or ITC) with purified SAHH and the compound. Furthermore, an enzymatic activity assay demonstrating that SC83288 directly inhibits SAHH's function would provide the functional validation necessary to support this claim.

Nature Communications manuscript

We thank the reviewers for their stimulating and constructive comments. By challenging several of our interpretations, their feedback encouraged us to revisit key experiments and refine our mechanistic model, which has led to a substantially strengthened and more coherent study.

Reviewer #1 (Remarks to the Author):

Sanchez et al. explored the mode of action and the mechanism of resistance of a novel anti-*P. falciparum* compound SC83288. Previous work has suggested the association of PfATP6 mutations with resistance to SC83288. Through CRISPR mutagenesis, yeast complementation, live-cell Ca²⁺ imaging, click-chemistry localization, transcriptomics, metabolomics, and CETSA, their work suggested SC83288 being a competitive substrate rather than an inhibitor of PfATP6. In addition, SC83288 showed a polypharmacological activity of disrupting DNA replication, transcription, methylation, and parasite egress. These results demonstrated a novel mode of action of SC83288 in inducing a substantial fitness cost in parasites and its potential for malaria treatment. Although the study found a potentially novel drug resistance mechanism and multi-target effects of SC83288, however, the insufficient depth of mechanism interpretation, the lack of validation of key targets, the weak clinical translation data and the limitations of methodology significantly affected the reliability of the conclusion. Therefore, I recommend a major revision of the manuscript. The comments are listed below:

We fully understand the reviewer's concern regarding the lack of a validated molecular target in the initial submission. As stated in our original manuscript, the main focus was on the mechanism of resistance, where we provided biochemical and genetic evidence implicating PfATP6 as a critical resistance determinant that mediates redistribution of SC83288 into the parasite's endoplasmic reticulum. This part of the study, we believe, was received positively.

We also acknowledge that elucidating a drug's mode of action can be particularly challenging when the resistance determinant and the molecular target are distinct entities, as is the case for SC83288. Nevertheless, following the reviewers' recommendation, we have conducted additional experiments aimed at identifying and validating the primary target. These include chemical rescue assays, expanded RNA-seq analyses, and characterization of PfdNMT2 knockdown and overexpressing parasite lines. The results of these new experiments converge to identify PfdNMT2 as a key target of SC83288. Furthermore, we now demonstrate that SC83288 arrests karyokinesis by inhibiting DNA replication, which we link mechanistically to loss of epigenetic and epitranslational control resulting from inhibition of PfdNMT2.

1. ITDR-CETSA analysis: The authors should explain why they chose an in-cell rather than a lysate CETSA experiment. In the figure legend of Fig.7, the hits of ITDR-CETSA should be in Supplementary Table 6, not Supplementary Table 4. Given the concentration used, the authors are advised to present the MDT concentration in the unit of "nM" but not "uM". In addition, the authors should explicitly indicate which proteins meet each criterion (criteria 1, criteria 2, and spectral confidence). In the main manuscript, only 4 out of 28 CETSA hits (SERA5, SAHH, and two TCA enzymes) were studied as plausible target candidates in detail. However, there is a lack of evidence for direct binding or inhibition. Given the negative result in the docking and MD simulations of SC83288 to the extended cleft domain of SERA5 and PfSAHH enzymatic assays, the primary target remains elusive. This unfortunately impacts the quality of this study.

We thank the reviewer for these insightful comments regarding the CETSA experiments and data presentation.

Choice of in-cell vs lysate CETSA: We performed both in-cell and lysate ITDR-CETSA experiments. However, the lysate-based assays yielded inconsistent and non-reproducible results, likely due to the loss of native protein-protein and protein-ligand interactions upon cell disruption. By contrast, the in-cell CETSA

produced highly reproducible thermal profiles. We therefore focused on the in-cell dataset, which better reflects physiologically relevant conformational changes and compound engagement.

Nature of CETSA shifts: As stated in the revised manuscript, the in-cell ITDR-CETSA detects changes in protein thermostability that may arise from direct drug binding or indirect effects, such as post-translational modifications, altered complex formation, or binding of endogenous ligands (page 16, first paragraph). Each case requires independent validation.

PfSAHH validation: We consider PfSAHH a bona fide CETSA hit. The observed thermal stabilization is fully consistent with our metabolomics data showing elevated intracellular levels of MTA and adenine—two known product inhibitors of SAHH (Fig. 7a and Fig. 9c). Moreover, our new molecular docking and MD simulations indicate that SC83288 can bind to the apo form of PfSAHH by occupying the vacant NAD⁺ binding site, whereas the NAD⁺-bound holo enzyme does not accommodate the compound (Fig. 9d). This explains why recombinant PfSAHH purified from *E. coli*—which contains tightly bound NAD⁺—showed no inhibition by SC83288 in vitro. We therefore interpret the observed stabilization as the combined outcome of a direct interaction with apo-PfSAHH and indirect effects mediated by elevated MTA/adenine levels. These findings reinforce the validity of the in-cell CETSA approach and are discussed accordingly in the revised manuscript.

Selection of CETSA hits for validation: Candidate selection was based on manual curation of the most statistically robust and biologically relevant stabilizations. Specifically, hits were required to satisfy an $R^2 > 0.8$ and either (i) an AUC differing from the median by more than twice the median absolute deviation (MAD; i.e., $AUC < 0.8654$ or > 1.1395) or (ii) a $\geq 30\%$ change in soluble protein abundance (fold change < 0.77 or > 1.3). These thresholds are now clearly stated in the revised Supplementary Table 6, where data points meeting or not meeting these criteria are indicated in black and red, respectively.

SERA5, fumarate hydratase, and isocitrate dehydrogenase: We performed additional molecular docking analyses for these proteins, which revealed only weak, surface-associated binding poses with unfavorable docking scores (Supplementary Fig. 12). We therefore conclude that the thermal shifts observed in CETSA most likely reflect indirect effects rather than direct drug–protein interactions. This interpretation has been integrated into the revised Results and Discussion sections (page 17 and 24).

Finally, as suggested by the reviewer, the MDT concentration is now reported in nM instead of μM , and the figure legend has been corrected to refer to Supplementary Table 6 (not Table 4).

2. Metabolomics analysis: In Fig.6A, is the ratio of AdoMet to SAH increased? Or SAH is a typo for SAM? In a volcano plot, each dot should typically represent one metabolite. There was a mismatch between this figure to the associated text. The authors should specify all the dysregulated metabolites and expand the relevant result description (which only talks about MTA, adenine, and phosphoethanolamine).

We thank the reviewer for these helpful comments.

SAM/SAH ratio: The volcano plot indeed shows the ratio of S-adenosylmethionine (SAM) to S-adenosylhomocysteine (SAH), rather than the individual metabolites. We acknowledge that depicting a ratio is atypical for metabolomic volcano plots, where each data point usually represents a single metabolite. However, in this case, inclusion of the SAM/SAH ratio is justified because it represents a well-established indicator of the cellular methylation potential and thus provides biologically meaningful context for interpreting methylation-related perturbations. To clarify this, we have modified the legend of Figure 7a to read:

“Each dot represents an individual metabolite, with the exception of the SAM/SAH data point, which represents the mean ratio of paired values for S-adenosylmethionine and S-adenosylhomocysteine.”

Dysregulated metabolites: In addition to MTA, adenine, phosphoethanolamine, and the SAM/SAH ratio, our metabolomic dataset indicated alterations in eight additional metabolites. However, these could not

be independently verified in *P. falciparum* extracts through orthogonal validation experiments (addition of authentic standards and reanalysis). We have therefore restricted our discussion to those metabolites whose identity and perturbation could be confidently confirmed. This is now clarified in the Results section by the following statement:

“Although additional metabolites were predicted to shift upon SC83288 exposure, only a subset—including MTA, adenine, phosphoethanolamine, and the SAM/SAH ratio—could be confirmed through orthogonal validation, beyond mass-based annotation.” (page 13)

3. Despite CETSA “hits” (SERA5, SAHH, TCA enzymes), direct binding was not confirmed (e.g. PfSAHH enzymatic assays negative). A bona fide primary target or high-affinity binding partner remains elusive.

We refer the reviewer to our detailed response to Comment 1. As elaborated there, we consider PfSAHH a bona fide in-cell ITDR-CETSA hit. The observed thermal stabilization can be mechanistically explained by two complementary processes: (i) conformational changes induced by the accumulation of the physiological product inhibitors MTA and adenine, both elevated in SC83288-treated parasites, and (ii) direct binding of SC83288 to the NAD-free (apo) form of the enzyme, as supported by our molecular docking and MD simulations. The NAD-bound (holo) form, in contrast, does not accommodate SC83288, which explains the lack of inhibition in the in-vitro enzyme assays that used recombinant, NAD-saturated PfSAHH purified from *E. coli*. Together, these findings account for the observed CETSA stabilization and support PfSAHH as a true hit reflecting physiologically relevant compound engagement.

4. Improve Data Clarity. Simplify and quantitatively support complex figures; provide co-localization statistics for click-chemistry (Figure 4)

We thank the reviewer for this helpful suggestion. To improve data clarity, we have revised formerly Figure 4 to include a schematic overview of the experimental workflow (now Fig. 3e) for easier interpretation. In addition, we have quantified the click-chemistry-derived fluorescence intensities of SC106879 in PfATP6 F972Y (E4) parasites and corresponding control samples, providing statistical analysis of the co-localization signal (now Fig. 3h). These additions clarify the experimental design and quantitatively substantiate the localization data.

5. Although SC83288 is reported to disrupt DNA replication, transcription, and metabolism, no conditional knockdown/complementation or co-treatment experiments were performed to verify which pathways are true on-target effects. Functional studies linking specific candidate targets to the observed phenotypes are required.

We thank the reviewer for this important comment. From the outset, our phenotypic and biochemical evidence suggested PfDNMT2 as a likely molecular target of SC83288. Treated parasites showed clear hallmarks of DNA replication arrest (Fig. 5a, b) and substantial reductions in global m⁵C DNA methylation (Fig. 8c, d). Moreover, purified PfDNMT2 was inhibited by SC83288 in a concentration-dependent manner in vitro (Fig. 8e).

To functionally validate this link, we have now performed additional experiments using PfDNMT2 knockdown and overexpressing parasite lines. While the partial knockdown line exhibited SC83288 sensitivity comparable to the parental 3D7 strain, the ~12-fold overexpressing line displayed a pronounced loss of drug responsiveness (Fig. 8f), consistent with an out-competition mechanism.

These new genetic data provide direct functional evidence connecting PfDNMT2 to the observed phenotypic effects of SC83288 and, we believe, address the reviewer’s request for experiments linking specific targets to the drug’s mode of action.

6. Broaden Impact Discussion. Frame the ER-sequestration mechanism in the context of other resistance pathways, and discuss translational implications for next-generation antimalarials.

We thank the reviewer for this valuable suggestion. In the revised manuscript, we have substantially expanded the discussion of SC83288 resistance to situate it within the broader framework of antimalarial resistance mechanisms. We now explicitly contrast the PfATP6-mediated redistribution of SC83288 with canonical target-based resistance seen for PfDHFR, PfDHPS, PfATP4, and cytochrome bc1, as well as with efflux- or degradation-based mechanisms such as PfCRT-mediated chloroquine resistance and PfKelch13-associated artemisinin tolerance (page 26).

This comparison highlights that SC83288 resistance represents a distinct paradigm in which the resistance determinant (PfATP6) and the primary drug target (PfDNMT2) are separate molecular entities. Rather than lowering target affinity, resistance arises through spatial sequestration of the drug into the endoplasmic reticulum, effectively removing it from its nuclear site of action. We also added a short section discussing the evolutionary and translational implications of this mechanism, emphasizing that while PfATP6-mediated sequestration enables temporary survival under drug pressure, it imposes a significant fitness cost. Consequently, such resistance is unlikely to spread widely in natural populations, underscoring SC83288's potential durability as a next-generation antimalarial candidate.

These additions appear in the final part of the Discussion (page 26).

7. The similarities and differences with the artemisinin resistance mechanism (Kelch13 mutation) were not deeply compared, which weakened the uniqueness argument of it as a "new mechanism drug".

We thank the reviewer for this insightful comment. We have now expanded the discussion to provide a clearer comparison between PfKelch13-mediated artemisinin resistance and the PfATP6-mediated SC83288 resistance mechanism. The new paragraph (page 26) outlines both similarities and key distinctions between the two systems. While both involve adaptive responses that mitigate drug toxicity without directly modifying the primary target, artemisinin tolerance relies on altered proteostasis and reduced drug activation, whereas SC83288 resistance operates through active intracellular sequestration of the compound into the ER, spatially removing it from its nuclear target, PfDNMT2. This comparison reinforces the uniqueness of SC83288 resistance as a previously unrecognized form of spatial drug resistance in *P. falciparum*.

8. Although ITDR-CETSA identified potential targets such as SERA5 and SAHH, the necessity of these proteins in drug effects was not verified through functional experiments (such as target-specific inhibitors and gene editing).

We thank the reviewer for this valuable comment. As noted above, PfSAHH represents a bona fide hit in the in-cell ITDR-CETSA, with its SC83288-induced thermal stabilization explained by conformational changes arising from two sources: (i) the accumulation of the endogenous product inhibitors MTA and adenine, which are markedly elevated in the metabolomics data, and (ii) direct binding of SC83288 to the NAD-free (apo) form of the enzyme, as supported by docking and MD simulations (Fig. 9d).

For SERA5, fumarate hydratase (FH), and isocitrate dehydrogenase (IDH), the evidence for direct drug engagement is weaker. Molecular docking revealed only low-affinity, surface-associated binding poses with unfavorable scores (Supplementary Fig. 12). Given that both FH and IDH are dispensable for intraerythrocytic growth and function within a largely anabolic TCA cycle, their contribution to SC83288's mode of action is likely minimal. We explicitly acknowledge these limitations in the revised discussion and propose further validation—such as in vitro enzymatic assays and in vivo metabolic flux analyses—as important next steps beyond the scope of the current study.

“Genetic studies indicate that both fumarate hydratase and isocitrate dehydrogenase are non-essential for asexual intraerythrocytic development, consistent with the parasite’s reliance on glycolysis for ATP generation and the presence of a largely anabolic or branched TCA cycle^{47,48}. Thus, inhibition of these enzymes is unlikely to account for SC83288’s antiplasmodial activity, although transient or low-affinity interactions cannot be excluded. Molecular docking using homology models (in the absence of crystal structures) did not support direct binding, revealing only surface-associated poses with unfavorable docking scores. Definitive clarification of whether the observed CETSA-derived thermal shifts represent direct drug binding, indirect metabolic effects, or false positives will require additional experimentation, including in vitro enzyme activity assays, in vivo measurements of NADP(H) redox and α -ketoglutarate/isocitrate ratios, and stable-isotope tracing to quantify flux redistribution under drug pressure.” (page 25)

Reviewer #2 (Remarks to the Author):

This manuscript describes characterization of mechanisms of resistance of malaria parasites to SC83288, a compound whose excellent antimalarial and drug-like properties and mechanism of resistance (due to mutations in the Ca-ATPase PfATP6) were described in 2017. This new report adds an in depth characterization of resistance mechanisms, with the interesting observation that a PfATP6 mutation that mediates high level resistance does not alter transport of SC83288 into the infected erythrocyte, but rather alters its intracellular distribution, with concentration in the ER, presumably sequestering the compound apart from its site of action. Further, the manuscript explores other effects of SC83288, with demonstration of impacts on various processes via quantification of cellular DNA, transcriptomic and metabolomic studies, and thermal shift assays. These assays led to interesting results, with intriguing suggestions for processes and proteins potentially impacted by SC83288, but it is difficult to distinguish direct effects of the compound from secondary responses in parasites inhibited in some manner by the compound. There was certainly no “smoking gun”, as different approaches identified different potential targets, and each of the 4 proteins that made it through multiple filters after thermal shift assays was not particularly supported as a target by additional considerations (although the authors are to be praised for providing descriptions of these careful considerations). Overall, the report describes a large, well-done series of experiments attempting to better describe mechanisms of action and resistance to a promising drug lead. The results add importantly to our understanding of the action of and resistance to SC83288, but, as is often the case with compounds with high barriers to resistance, the mechanism of action of the compound and the full mechanism of resistance remain incompletely understood. Some changes to the manuscript are suggested, as follows.

We sincerely thank the reviewer for their thoughtful and encouraging assessment of our work. We greatly appreciate their recognition of the experimental breadth and rigor with which we have characterized the mechanisms of SC83288 action and resistance. We agree that defining the precise molecular target of a new compound is inherently challenging, and we value the reviewer’s acknowledgment that our data substantially advance understanding of both its mode of action and the associated resistance mechanisms, even if some aspects remain to be fully resolved.

1) Line 39. The term “polypharmacological” is not very familiar, and it seems synonymous in this context with “multiple drug targets”, so it is not clear why both terms are used here.

As suggested by the reviewer, we have removed the term polypharmacological from the manuscript. It is no longer required, as new experimental data now clearly identify PfDNMT2 as the primary molecular target of SC83288. This conclusion is supported by multiple independent lines of evidence: (i) phenotypic observations showing inhibition of DNA replication and arrest of karyokinesis (Fig. 5), (ii) a marked

reduction in total m⁵C DNA methylation following treatment (Fig. 8c and d), (iii) direct, concentration-dependent inhibition of recombinant PfDNMT2 in vitro (Fig. 8e), and (iv) loss of drug responsiveness in a PfDNMT2-overexpressing parasite line (Fig. 8f), consistent with an out-competition mechanism.

2) Line 51. It should be noted that these are the numbers estimated, not reported, e.g. “...estimated in 2023 alone.”

We agree with the reviewer and have clarified the text as follows:

“Malaria remains a major infectious disease, with an estimated 263 million cases and approximately 597,000 deaths in 2023 alone, primarily in sub-Saharan Africa and mainly affecting children under the age of five.” (page 3)

3) Line 54. Life cycle, not life style.

The term life style has been replaced with life cycle.

4) Line 55. Omit “chilling” or change to “chills, fevers....”

The phrase has been corrected to “chills, fevers, and...” in accordance with the reviewer’s suggestion.

5) Line 59. This statement is inaccurate, as the most widely used antimalarial in the world (artemether-lumefantrine) includes lumefantrine, which is not in any of these categories (unless it is considered “quinoline-like”, though lacking a quinoline ring?).

We thank the reviewer for this clarification. Lumefantrine is an aryl aminoalcohol compound, like other quinoline-related antimalarials such as chloroquine, quinine, mefloquine, and halofantrine. To reflect this accurately without listing specific drugs, the sentence now reads:

“The treatment of malaria primarily relies on three major drug classes: artemisinins, aryl aminoalcohol compounds, including quinoline derivatives and quinoline-like drugs, and antifolates.” (page 3)

6) Line 65. “uncomplicated falciparum malaria”

We now refer explicitly to uncomplicated falciparum malaria, as suggested:

“A particularly worrying trend is the emergence and spread of *P. falciparum* lines that are partially resistant to artemisinin-based combinations, the first line treatment against uncomplicated falciparum malaria and the cornerstone of current malaria control strategies.” (page 3)

7) Line 67. It is not clear what is meant by “accessibility”. Was the intended meaning “availability”?

We agree that availability better conveys the intended meaning. The revised sentence now reads:

“Hence, there is a need for novel drugs that not only offer improved efficacy but also ensure safety and availability, given that malaria predominantly affects vulnerable populations in resource-limited settings.”
“Hence, there is a need for novel drugs that not only promise heightened efficacy but also prioritize safety and availability given that malaria is a disease that predominantly affects vulnerable populations in resource-limited settings.” (page 3)

8) Line 87-99. This summary is helpful, but it appears to be fully a summary of the results of this report. If that is the case, it belongs in the Discussion, not the Introduction. Inclusion of 1-2 sentences briefly explaining the approach of this study and perhaps a “teaser” regarding results would be appropriate, but

this 13 line summary seems excessive, and also a bit confusing, as readers will not be sure whether these data are background from prior studies or the results of this study. Of note, in the first paragraph of the Discussion, where one expects to see a summary of new results, the authors mostly summarized previously published results (see comment 13 below). Reorganization will be helpful and will make it easier for readers to appreciate what is new in this report.

We appreciate the reviewer's constructive feedback. The introduction has been restructured and shortened to provide only essential background and a brief rationale for the study. The detailed summary of results has been removed from the Introduction and replaced with a concise statement outlining the study's approach. The first paragraph of the Discussion now contains a focused summary of the new findings reported here.

9) Line 127. It will be slightly clearer to describe the assay before describing the results of the assay, i.e. to place the last sentence earlier in the paragraph.

We have adjusted the sentence order so that the assay is described before presenting the corresponding results, as recommended.

10) Lines 204 and 390. The short descriptions of the mechanism of resistance to artemisinins is a bit misleading. Although this is only peripherally relevant to this report, it would be helpful to note that artemisinin partial resistance seems primarily mediated by mutations in K13 that alter hemoglobin transport to the food vacuole, as nicely demonstrated by Spielmann's group (Birnbaum, et al., Science 2020).

We thank the reviewer for this valuable clarification and have expanded the discussion to more accurately compare artemisinin resistance with the mechanism observed for SC83288. The revised paragraph now reads as follows:

"The PfATP6-mediated sequestration of SC83288 represents a distinct resistance paradigm that differs fundamentally from both canonical target-based or efflux mechanisms and PfKelch13-mediated artemisinin tolerance. Although both involve adaptive responses that mitigate drug-induced damage rather than altering the drug's primary target, their modes of action diverge. PfKelch13 mutations reduce endocytosis of host hemoglobin into the food vacuole, thereby limiting hemoglobin proteolysis and the release of heme, which is required for artemisinin activation⁵⁵. This attenuation of drug activation diminishes oxidative stress and underlies partial artemisinin resistance⁵⁵. In contrast, PfATP6-mediated resistance redistributes SC83288 into the endoplasmic reticulum, spatially separating the compound from its nuclear target, PfDNMT2. This form of intracellular drug sequestration—coupled with its high fitness cost and absence of target mutations—defines SC83288 resistance as a mechanistically novel and distinct adaptation. From a translational perspective, this mechanism provides short-term survival but imposes a significant metabolic burden, likely restricting its spread in natural parasite populations." (page 26)

11) Line 205. Chloroquine is a quinoline, not a quinolone.

We thank the reviewer for catching this oversight. The typo has been corrected; chloroquine is now correctly referred to as a quinoline compound in the revised manuscript.

12) Line 228-230. It is unclear how Fig. 5A demonstrates a block in RBC egress. A photomicrograph seems a better way of demonstrating this, and in addition it would provide general information by showing the

morphology of treated parasites. Has such a block been demonstrated morphologically in older reports on SC83288?

We are very grateful for this insightful comment, which helped us clarify the developmental stage at which SC83288 exerts its effect. In the revised manuscript, we now include microscopic images of Giemsa-stained blood smears from synchronized parasite cultures exposed to SC83288 during different intraerythrocytic intervals (starting 8–12 hpi, 30–34 hpi, and 42–46 hpi onwards). These new data (now presented in Fig. 5c–e) demonstrate that SC83288 arrests karyokinesis, resulting in pyknotic parasites characterized by nuclear condensation and progressive reduction in cell volume. When the compound is applied after completion of karyokinesis (42–46 hpi), parasites proceed normally through schizogony, including cytokinesis and egress. Thus, SC83288 specifically targets karyokinesis rather than egress.

We apologize for the confusion caused by our earlier phrasing and have corrected the description accordingly in the Results (page 10) and Discussion (page 21 and 27) sections.

13) Line 377. The first paragraph of the Discussion is nearly completely a review of old data, mostly already provided in the Introduction. It would be more value to use this paragraph to summarize the new data provided in this report and to explain how that adds to our understanding. In fact, it is not convincing that mutations in PfATP6 are unlikely to emerge in human infections; a more reasonable interpretation seems to be that selection of resistance to SC83288 occurs less readily compared to some other antimalarials, but one can't be sure how readily resistance will emerge if the compound is widely used to treat malaria.

We thank the reviewer for this valuable feedback. In response, we have revised the opening of the Discussion to focus on the novel findings of this study and their significance, while removing material that merely repeated background information from the Introduction. The revised first paragraph now reads as follows:

“Our findings show that SC83288 disrupts blood-stage development of *P. falciparum* by inhibiting DNA replication and arresting karyokinesis (Fig. 5). We identify the parasite's DNA methyltransferase, PfDNMT2, as a primary molecular target (Figs. 8c–f), linking SC83288 activity to disrupted epigenetic regulation and transcriptional control. Resistance arises through mutations in the SERCA-type Ca²⁺ pump PfATP6, which convert the transporter into a drug redistribution system, sequestering SC83288 into the endoplasmic reticulum and thereby away from its nuclear target(s).” (page 20)

14) Line 428-431. This sentence has grammatical problems. Since each approach yielded different “targets” for SC83288, it seems more likely that some of these approaches did not actually identify a target rather than to assume that each identified different valid targets. Overall, although the varied data are of interest and worth reporting (and discussing in subsequent paragraphs), it does not clearly follow that, because it has been difficult to identify a target, the compound has a polypharmacological mechanism.

We thank the reviewer for this constructive comment. We agree that the original paragraph was both grammatically unclear and conceptually overstated. In the revised manuscript, the paragraph has been removed entirely. The Discussion has been restructured and expanded to focus on the new data, which converge on PfDNMT2 as the primary molecular target of SC83288. This updated section now integrates evidence from enzymatic inhibition assays, DNA methylation analyses, and genetic perturbation studies (PfDNMT2 knockdown and overexpression), thereby resolving the earlier ambiguity and eliminating the need to invoke a polypharmacological mechanism.

15) Line 475-491. This discussion addresses potential interaction between SC83288 and SERA5 leading to a block in parasite egress from the erythrocyte. This is interesting, but key information is not mentioned

in the discussion. Does SC83288 cause an obvious block in egress, as can be studied simply by looking at treated parasites (by Giemsa-stained smear) over the course of the life cycle? This point was also brought up above. If information is already available on the morphology of SC83288-treated parasites over the course of the life cycle, this information should be included in the discussion suggesting specific action against parasite egress. If not, a simple experiment showing the morphology of treated parasites over the course of the life cycle should be added.

We appreciate the reviewer for again drawing attention to this critical point. Indeed, it touches on what we now recognize as our most significant oversight in the original submission. Our earlier interpretation of the DNA content data led us to suggest a potential block in egress; however, after revisiting this issue carefully, we realized that the phenotype was misread.

To resolve this, we performed new microscopic analyses of Giemsa-stained blood smears from highly synchronized parasite cultures treated with SC83288 at defined intraerythrocytic intervals (8–12 hpi, 30–34 hpi, and 42–46 hpi onwards). These new data (now included in Fig. 5c–e) clearly show that SC83288 arrests karyokinesis, producing pyknotic forms with condensed nuclei and reduced cytoplasmic volume, while cytokinesis and egress remain unaffected.

We have corrected this interpretation throughout the manuscript, and the results are now fully described in the Results and Discussion sections. We are genuinely grateful to the reviewer for prompting this important clarification.

Reviewer #3 (Remarks to the Author):

The authors have previously conducted an amicarbalide-based medicinal chemistry program for development of novel antimalarials, and screening of these chemicals showed that two compounds with optimized pharmacological and antiparasitic properties. Particularly, Comprehensive pre-clinical evaluation revealed that the compound SC83288 harbor potential as a clinical development candidates for malaria therapy. Interestingly, they also found that the sarco/endoplasmic reticulum Ca²⁺ transporting PfATP6 might be a putative drug resistant marker gene to this compound. Therefore, understanding the underlying mechanisms of parasite-killing action and drug resistance is crucial for the further development of this compound as a new-generation antimalarial. In this study, the authors have comprehensively investigated the pharmacological activity and drug targets via combinations of various approaches. They found that SC83288 is a substrate for PfATP6, in which the mutations may sequester the compound into the compartment of ER thereby blocking the potential interaction between the drug and its targets. This part is convincing. It is notable that the mutations of PfATP6 found in the in vitro selected drug-resistant clones may result in heavy fitness-cost, and are not previously identified in field isolates. This finding further confer this compound high potential as new drug candidate for malaria treatment. In general, this work is interesting and important for development of new-generation antimalarials, which is urgently required when the artemisinin-based therapies (ACTs) are now encountering challenge of drug resistance including artemisinin itself. While the phenotypic data are solid and convincing, the mechanistic interpretations are relatively weak and elusive. Though polypharmacological activity and multiple drug targets may be preferable for drug development against malaria parasites, but it is difficult to justify this conclusion with the data presented in the current version of manuscript. Here are some concerns still need to be addressed by the authors.

We thank the reviewer for their encouraging and insightful comments. We greatly appreciate their recognition of the robustness of our phenotypic data and of our findings regarding the PfATP6-mediated mechanism of resistance. We also fully agree with the reviewer that, in the initial submission, our mechanistic interpretation of SC83288's mode of action was incomplete.

In the revised manuscript, we have therefore concentrated our efforts on clarifying the drug's cytotoxic mechanism. New experimental data now converge on PfDNMT2 as the principal molecular target of SC83288. This conclusion is supported by multiple independent lines of evidence:

(i) phenotypic observations demonstrating inhibition of DNA replication and arrest of karyokinesis (Fig. 5);
(ii) a marked reduction in total m⁵C DNA methylation following treatment (Fig. 8c);
(iii) direct, concentration-dependent inhibition of recombinant PfDNMT2 in vitro (Fig. 8e); and
(iv) pronounced loss of drug responsiveness in a PfDNMT2-overexpressing parasite line (Fig. 8f), consistent with an out-competition mechanism.

Together, these data now establish PfDNMT2 as a key molecular target underlying SC83288's antiparasitic activity, thereby addressing the reviewer's concern about the previously insufficient mechanistic interpretation.

Major concerns:

1. the F972Y mutation of PfATP6 was identified in SC83288 resistant clones selected with lower drug concentration in vitro, which is also validated by gene editing of this gene. As shown in the Fig. 2A, this site located in the fifth transmembrane domain. While no evidence of the direct interaction between SC83288 and PfATP6 is available, it is unclear about the role of this position in PfATP6's function. What's mechanism of this mutant protein in enhancing the local distribution of SC83288 in the compartment of ER? What's the mechanism of heavy fitness cost conferred by this mutation in PfATP6?

We thank the reviewer for these important mechanistic questions. We interpret them as referring to (i) the functional role of the PfATP6 F972Y mutation in mediating SC83288 transport and ER sequestration, and (ii) the origin of the associated fitness cost.

Our in vitro vesicle transport assays (Fig. 2c) and in-cell localization studies (Fig. 3f–h) clearly demonstrate that PfATP6 actively transports SC83288, and that the F972Y mutation enhances this transport efficiency, leading to drug accumulation in the ER. To better understand the molecular basis of this phenotype, we performed molecular docking and molecular dynamics (MD) simulations for both wild-type and mutant PfATP6 across four conformational intermediates of the Ca²⁺-ATPase transport cycle: E1 (cytoplasmic-open, Ca²⁺-binding), E1~P~ADP (phosphorylated transitional state), E2~P (luminal-open, Ca²⁺-release state), and E2 (resetting state).

SC83288 was predicted to bind to both PfATP6 forms in all conformations, but in no case did the interaction involve residue F972/Y972 (Fig. 4). These results indicate that the mutation does not alter drug binding directly, but rather enhances the kinetic efficiency of transport, likely by subtly modifying the dynamics of the transmembrane region that gates substrate passage. This interpretation aligns with our experimental data showing higher SC83288 uptake by the mutant transporter (Fig. 2c).

The fitness cost associated with the F972Y mutation appears to stem from competition between SC83288 and Ca²⁺ for transport through PfATP6. Our transport assays revealed complete inhibition of SC83288 uptake at elevated Ca²⁺ concentrations, and live-cell Ca²⁺ imaging confirmed altered cytoplasmic Ca²⁺ responses in the mutant line E4 (Fig. 1f–g). Thus, the mutation perturbs Ca²⁺ homeostasis, which in turn affects multiple essential signaling pathways, explaining the pronounced growth defect observed in the resistant parasites.

The new molecular docking and MD results are presented in the Results section (page 8 and 9) and discussed in detail in the Discussion (page 21).

2. the authors claim that the SC83288-induced rise in cytosolic free Ca²⁺ would be a result of a competition reaction between Ca²⁺ and this compound. Though they tested this phenotype in *S. cerevisiae* in vivo and in vitro and the results are convincing, what's the possible mechanism of the competition between them, as questioned in the first concern that this compound does not directly interact with PfATP6.

We thank the reviewer for this excellent question.

Interaction between PfATP6 and SC83288: Transport of any solute by an ATP-driven pump inherently requires a transient, specific interaction between the transporter and its substrate (Stein, Transport and Diffusion Across Cell Membranes, 1986). Because our in-vitro vesicle assays demonstrate active, ATP-dependent uptake of SC83288 mediated by PfATP6 (Fig. 2c), the data necessarily imply that SC83288 interacts directly with the transporter as a transported substrate. Molecular docking and molecular-dynamics (MD) simulations further support this conclusion, identifying specific transmembrane residues involved in transient contacts with SC83288 throughout the transport cycle.

Mechanism of competition between SC83288 and Ca²⁺: Our experimental data indicate that SC83288 and Ca²⁺ compete for access to the PfATP6 transport pathway. In the vesicle transport assay, excess Ca²⁺ abolishes SC83288 uptake (Fig. 2c), while in live parasites, addition of SC83288 triggers a transient rise in cytosolic Ca²⁺ (Fig. 1f–g). Together, these observations suggest that SC83288 and Ca²⁺ share overlapping translocation routes within the PfATP6 catalytic cycle. Binding of SC83288 likely transiently displaces Ca²⁺ or interferes with its release, thereby perturbing Ca²⁺ homeostasis and producing the observed increase in cytoplasmic free Ca²⁺.

This mechanistic model—direct substrate recognition of SC83288 by PfATP6, coupled with interference with Ca²⁺ transport—is now described in the Results (page 7) and elaborated in the Discussion (page 26).

3. For the differential transcriptome analysis of parasites treated with SC83288 and control, the PfATP6-mutant line, e.g., E4, should be included. This may help validate the drug-responsive genes or pathways, and also interpret the heavy fitness cost due to the site mutation in PfATP6.

We thank the reviewer for this valuable suggestion. We agree that comparative transcriptomic analysis of the PfATP6^{F972Y} mutant (E4 line) would provide important complementary insights—particularly for distinguishing drug-responsive pathways from those related to the mutation’s intrinsic fitness cost.

However, due to time constraints and the substantial additional effort required to generate and sequence sufficient biological replicates for this line, we prioritized during revision:

- (i) experimental validation of PfDNMT2 as the primary molecular target of SC83288,
- (ii) mechanistic characterization of the F972Y mutation in PfATP6 using biochemical transport assays, molecular docking, and MD simulations, and
- (iii) replication and statistical reinforcement of the original RNAseq analysis in wild-type parasites.

We fully recognize the value of extending the transcriptomic comparison to the PfATP6^{F972Y} mutant and plan to pursue this in future work as part of a broader systems-level analysis of the fitness cost and adaptive response associated with SC83288 resistance.

4. SC83288 treatment showed significant alteration of enzymes involved in DNA and RNA methylation. For the DNA methyltransferase PfDNMT2, Fig. 6D provided solid evidence of inhibition of this enzyme by SC83288. However, it is likely not clear how this drug acts on the RNA m⁶A writer complex. It is known that the two key writers of mRNA m⁶A modification are PfMT-A70 and PfMT-A70.2 corresponding to the conserved METLL3 and METLL14 in model eukaryotes, but not WTAP?

We thank the reviewer for this insightful comment. We agree that clarifying how SC83288 affects the RNA m⁶A writer complex is an important next step, particularly given the observed reduction in global m⁶A RNA methylation following drug exposure.

While, due to time constraints, we prioritized experimental validation of PfDNMT2 as the primary target, we have taken the reviewer’s suggestion into account and now explicitly discuss the potential involvement of the PfMT-A70/PfMT-A70.2–WTAP complex—the parasite’s canonical m⁶A writer machinery, homologous to the METTL3/METTL14–WTAP system in higher eukaryotes.

To acknowledge this in the manuscript, we have added the following sentence to the Results section:

“In addition to PfDNMT2, SC83288 may also affect PfMT-A70 (PF3D7_0729500), PfMT-A70.2 (PF3D7_1235500), or its regulatory subunit WTAP (PF3D7_1230800), which together constitute the parasite’s principal RNA N⁶-methyladenosine (m⁶A) methyltransferase complex^{29,30}.” (page 15)

We plan to investigate this potential link further in future studies using biochemical and genetic approaches to dissect how SC83288 may modulate PfMT-A70 complex function.

5. How to interpret the observation that the pronounced changes in RNA levels occurred independent of the SC83288 dose (Fig. 5C)? Does it indicate that a majority of differential expression came from indirect effect by drug treatment, or represents a result of cell death pathways? Though the main stage with potent inhibitory impact of SC83288 was observed in thophozoites, a timecourse of transcriptomic dynamics with lower drug concentrations, eg. pmol grade, may be helpful.

We greatly appreciate the reviewer’s thoughtful question, which prompted us to re-examine our original RNA-seq dataset. Upon critical re-evaluation, we identified technical inconsistencies that warranted repeating the entire experiment under refined and better-controlled conditions. In the revised analysis, we placed particular emphasis on maintaining parasite viability during exposure and employed improved analytical pipelines for normalization and statistical modeling.

These new data—now presented in Figure 6—show clear and biologically coherent transcriptional responses to SC83288 treatment. In contrast to the earlier dataset, the magnitude of transcriptomic changes now correlates with drug concentration, and the differentially expressed genes display functional enrichment consistent with nuclear targeting and disruption of DNA replication and mitotic progression. As discussed in the revised manuscript:

“The comparative, unbiased RNA-seq analysis revealed a distinct transcriptional signature in SC83288-treated parasites, with six of nine upregulated genes encoding proteins of nuclear or partially nuclear localization (Fig. 6). These included histone deacetylase 1, RuvB-like helicase 3, eIF4A, aurora kinase ARK1, α -tubulin 1, and SEC61 subunit β , which function in epigenetic regulation, chromatin remodeling, transcription, mitosis, spindle formation, and protein translocation, respectively. These transcriptional responses are consistent with nuclear targeting by SC83288 and likely reflect compensatory mechanisms triggered by impaired epigenetic regulation and karyokinesis, particularly via inhibition of PfDNMT2. The remaining three upregulated genes—DER1-like protein, serine hydroxymethyltransferase, and plasmepsin IV—are indicative of stress responses, likely linked to misfolded protein accumulation, impaired protein trafficking, and metabolic adaptation.” (page 24)

We agree that future time-course RNA-seq analyses at lower SC83288 concentrations would further illuminate early versus late transcriptional responses, and we intend to pursue this in subsequent studies.

6. It is difficult to understand the result of ITDR-CETSA analysis. This work points to four target candidates of SC83288, but no proof was shown for the direct interaction between this compound and each of them. Moreover, Fumarase and isocitrate dehydrogenase are not essential for the asexual development of parasites. Perhaps other methods such as Activity-based Protein Profiling (ABPP) may produce different result. But, at least the authors should demonstrate that the four target candidates do not localize in the ER by IFA or other assays, otherwise the proposed mechanism of PfATP6-mediated resistance will not work in these cases.

We thank the reviewer for these thoughtful comments and agree that clarifying the interpretation of the ITDR-CETSA results is important.

Subcellular localization: None of the four CETSA hits are localized to the endoplasmic reticulum. PfSAHH displays both cytoplasmic and nuclear localization (Brzezinski et al., 2020), SERA5 associates with the parasitophorous vacuole membrane and the host erythrocyte plasma membrane (Collins et al., 2017), and fumarate hydratase and isocitrate dehydrogenase are mitochondrial enzymes of the tricarboxylic acid (TCA) cycle (Ke et al., 2015; Rajaram et al., 2022). Thus, their spatial distribution supports our model that PfATP6-mediated sequestration of SC83288 into the ER separates the compound from its true nuclear and cytoplasmic targets.

Nature of CETSA shifts: As noted in the revised manuscript, the in-cell ITDR-CETSA detects protein thermostability changes that can arise from either direct drug binding or indirect effects such as altered post-translational modifications, protein–protein interactions, or ligand occupancy. Each case requires independent validation, which we have pursued selectively.

PfSAHH validation: Among the identified proteins, we consider PfSAHH a bona fide CETSA hit. The observed thermal stabilization aligns with our metabolomics data showing increased intracellular levels of methylthioadenosine (MTA) and adenine—both established product inhibitors of SAHH. Furthermore, new molecular docking and MD simulations show that SC83288 can bind to the NAD⁺-free apo form of PfSAHH by occupying the cofactor binding site, whereas the NAD⁺-bound holo enzyme does not accommodate the compound. This explains why recombinant PfSAHH purified from *E. coli*, which retains tightly bound NAD⁺, showed no inhibition by SC83288 *in vitro*. We therefore interpret the CETSA stabilization as resulting from a combination of (i) direct binding of SC83288 to apo-PfSAHH and (ii) indirect effects mediated by elevated MTA and adenine levels.

SERA5, fumarate hydratase, and isocitrate dehydrogenase: For these proteins, additional molecular docking analyses revealed only weak, surface-associated binding poses with unfavorable docking scores. Given their non-essential roles in asexual intraerythrocytic development and the absence of metabolic perturbations in the TCA cycle, we conclude that their CETSA shifts likely reflect indirect effects rather than direct interactions with SC83288.

We have incorporated these clarifications into the revised Results and Discussion sections to ensure that the interpretation of CETSA data is both transparent and consistent with the overall mechanistic model.

7. Both differential transcriptomics and metabolism analysis point to cellular methylation involving DNA and RNA methyltransferases. I believe this metabolic pathway would be a key information responsible for cell death under SC83288 treatment. However, no any experiment evidence was shown in the manuscript, eg. gene manipulation of any m5C/m6A writher or other enzymes of methyl metabolism to rescue the phenotype?

We fully agree with the reviewer that functional genetic evidence is essential to establish the causal link between SC83288 activity and methylation-dependent pathways. From the outset, our phenotypic and biochemical data strongly suggested PfDNMT2 as a primary molecular target of SC83288: treated parasites exhibited clear inhibition of DNA replication and karyokinesis (Fig. 5a, b), a marked reduction in global m⁵C DNA methylation (Fig. 8c, d), and direct, concentration-dependent inhibition of recombinant PfDNMT2 *in vitro* (Fig. 8e).

To functionally validate this relationship, we performed additional experiments using PfDNMT2 knockdown and overexpressing parasite lines. While the partial knockdown line displayed SC83288 sensitivity comparable to the parental 3D7 strain, the ~12-fold overexpressing line showed a pronounced loss of drug responsiveness (Fig. 8f), consistent with an out-competition mechanism.

These new genetic data provide direct functional evidence linking PfDNMT2 inhibition to the phenotypic effects of SC83288 and, we believe, fully address the reviewer's request for experiments connecting specific targets to the compound's mode of action.

Minor concerns

1. in the previous paper published in NC (Pegoraro et al. 2017), it looks that the F972Y mutation was found in SC81458 instead of SC83288?

The reviewer is correct that the F972Y mutation in PfATP6 was first identified in parasites selected under increasing concentrations of SC81458. However, these lines exhibited strong cross-resistance to SC83288, which is fully consistent with the close structural and pharmacological relatedness of the two compounds. In subsequent independent selection experiments, we also confirmed the emergence of the F972Y mutation in parasites selected directly with SC83288, validating its role as a shared resistance determinant for both analogs.

2. in Fig. 2A, the F972Y position looks not consistent with the previous version shown in the Fig. S11C published in NC 2017.

We thank the reviewer for this careful observation. The apparent discrepancy arises from differences in the structural models of PfATP6 used at the two time points. Since our 2017 publication, the understanding of PfATP6 and related SERCA-type Ca^{2+} ATPases has significantly advanced, owing to the availability of high-resolution crystallographic data from orthologous systems and improved structural predictions from AlphaFold2. In the revised manuscript, the residue F972 is mapped according to the AlphaFold2 model of PfATP6 (Fig. 1b), which provides a more accurate representation of its transmembrane topology and conformational context.

3. line 225: six should be 6 h

We thank the reviewer for noticing this. The correction has been made—the text now correctly reads “6 h.”

4. The duration of drug treatment varied in transcriptomic analysis (4.5 h) and metabolomic analysis (12 h). What’s the rationality of these conditions?

We thank the reviewer for this important question. In both assays, we consistently treated parasites at the trophozoite stage, which is the developmental phase most susceptible to SC83288 (Fig. 5a and Pegoraro et al., 2017). The differing exposure times reflect the distinct temporal dynamics of transcriptional and metabolic responses: transcriptional changes typically occur within a few hours after drug exposure, whereas metabolic perturbations require longer incubation to reach detectable steady-state alterations (Kim et al., 2020). The chosen time points therefore capture the respective peak response windows for each dataset, ensuring meaningful biological interpretation.

5. Why Pyrimethamine was shown in Fig. 1 and also used in the analysis of Fig 5B? I saw the structure of SC83288 likely from amicarbalide and sulfadoxine, but not Pyrimethamine?

We thank the reviewer for this observation. Pyrimethamine was included as a comparative control because it produces a phenotypically similar inhibition of DNA replication, despite acting through a distinct molecular mechanism (inhibition of dihydrofolate reductase). Its inclusion therefore served as a reference for the replication arrest phenotype observed under SC83288 treatment.

We initially attempted to include sulfadoxine as a second comparator, given its structural relatedness to amicarbalide derivatives; however, we were unable to establish stable culture conditions that allowed

reliable monitoring of sulfadoxine responses in our experimental setup, a recognized challenge in *Plasmodium falciparum* research (Wang et al., 1997; Ndounga et al. 2001).

6. The majority data of this manuscript likely focus on the pharmacological action or molecular targets instead of ER-enrichment-based resistance. The title may be modified to reflect the main part of this work.

We agree with the reviewer that the original title emphasized the resistance mechanism disproportionately. To better capture the scope of the work and its dual focus on drug action and resistance, we have revised the title to:

“Mechanisms of PfDNMT2 inhibition and PfATP6-mediated resistance to the antimalarial candidate SC83288 in *Plasmodium falciparum*” (page 1)

Reviewer #4 (Remarks to the Author):

Additional references

- Collins, C. R., F. Hackett, J. Atid, M. S. Y. Tan and M. J. Blackman (2017). "The *Plasmodium falciparum* pseudoprotease SERA5 regulates the kinetics and efficiency of malaria parasite egress from host erythrocytes." *PLoS Pathog* **13**(7): e1006453.
- Ke, H., I. A. Lewis, J. M. Morrissey, K. J. McLean, S. M. Ganesan, H. J. Painter, M. W. Mather, M. Jacobs-Lorena, M. Llinás and A. B. Vaidya (2015). "Genetic investigation of tricarboxylic acid metabolism during the *Plasmodium falciparum* life cycle." *Cell Rep* **11**(1): 164-174.
- Kim, S., Y. Kim, D. H. Suh, C. H. Lee, S. M. Yoo, S. Y. Lee and S. H. Yoon (2020). "Heat-responsive and time-resolved transcriptome and metabolome analyses of *Escherichia coli* uncover thermo-tolerant mechanisms." *Sci Rep* **10**(1): 17715.
- Ndounga, M., L. K. Basco and P. Ringwald (2001). "Evaluation of a new sulfadoxine sensitivity assay in vitro for field isolates of *Plasmodium falciparum*." *Trans R Soc Trop Med Hyg* **95**(1): 55-57.
- Pegoraro, S., M. Duffey, T. D. Otto, Y. Wang, R. Rosemann, R. Baumgartner, S. K. Fehler, L. Lucantoni, V. M. Avery, A. Moreno-Sabater, D. Mazier, H. J. Vial, S. Strobl, C. P. Sanchez and M. Lanzer (2017). "SC83288 is a clinical development candidate for the treatment of severe malaria." *Nat Commun* **8**: 14193.
- Rajaram, K., S. G. Tewari, A. Wallqvist and S. T. Prigge (2022). "Metabolic changes accompanying the loss of fumarate hydratase and malate-quinone oxidoreductase in the asexual blood stage of *Plasmodium falciparum*." *J Biol Chem* **298**(5): 101897.
- Wang, P., P. F. Sims and J. E. Hyde (1997). "A modified in vitro sulfadoxine susceptibility assay for *Plasmodium falciparum* suitable for investigating Fansidar resistance." *Parasitology* **115** (Pt 3): 223-230.

Nature Communications manuscript NCOMMS-25-23179A

We thank the reviewers for their thoughtful and constructive comments and sincerely appreciate the time and expertise they invested in evaluating our work. We are grateful for their insightful feedback and their support in considering our study for publication in Nature Communications.

Reviewer #2 (Remarks to the Author):

The authors made substantial changes to the manuscript based on reviewer comments, and in fact these changes considerably improved the manuscript. It now has a clearer message, and the argument that SC83288 most likely acts via inhibition of PfDNMT2 is well-supported by the exhaustive data provided. Although the modes of action and resistance of the compound are complex, and not fully explained by all available data, this large manuscript makes major advances in explaining drug action. The authors are praised for comprehensive responses to reviews and a very clear and comprehensive (albeit very long) manuscript. No additional changes are suggested.

We sincerely thank the reviewer for this very positive and encouraging assessment of our revised manuscript. We greatly appreciate the recognition that the revisions have strengthened the clarity of the study and the support for PfDNMT2 as a primary target of SC83288. We are particularly grateful for the acknowledgment of the comprehensive nature of the work and our efforts to address the reviewers' comments in depth.

Reviewer #3 (Remarks to the Author):

The authors have addressed part of the previous concerns, and the manuscript has been significantly improved. However, three major issues in my previous review comments remain unresolved in the revised version, which weaken the novelty and integrity of this work.

We thank the reviewer for acknowledging the substantial improvements made in the revised manuscript. We also appreciate the continued critical evaluation and understand the request for additional mechanistic depth. We agree that SC83288 likely exerts complex, multifactorial effects within the parasite. In the revised manuscript, however, we deliberately focused on validating PfDNMT2 as a primary molecular target, supported by multiple independent and convergent lines of evidence (biochemical inhibition, genetic modulation of drug response, methylation analyses, and phenotypic characterization). In parallel, we strengthened the mechanistic framework for PfATP6-mediated resistance.

We recognize that the potential involvement of additional factors, such as the RNA N⁶-methyladenosine methyltransferase complex, remains an important open question. While we consider this a promising avenue, fully dissecting such interactions would require substantial additional experimental work beyond the scope of the present study. Given the breadth of new data already incorporated and the overall length of the manuscript, we chose to prioritize consolidation of the strongest mechanistic axis (PfDNMT2 targeting and PfATP6-mediated sequestration) and to clearly delineate remaining questions for future investigation.

1. The transcriptome analysis remains incomplete without data from the PfATP6-mutant line (e.g., E4). This comparison is needed to confirm the specificity of the drug-responsive

signature and to investigate the fitness cost of the mutation. I urge the authors to provide this data.

We thank the reviewer for this thoughtful and well-justified suggestion. As outlined above, we had to carefully prioritize the additional experiments performed during revision. We therefore focused on repeating and refining the RNA-seq analysis in the wild-type 3D7 line under drug-treated and untreated conditions to ensure robustness and reproducibility of the transcriptional signature.

We did not extend the RNA-seq analysis to the PfATP6-mutant E4 line at this stage. In multiple independent phenotypic and biochemical assays, E4 parasites exposed to SC83288 behaved comparably to untreated wild-type parasites, consistent with effective drug sequestration and functional resistance. Based on these observations, we anticipated that transcriptomic profiling of drug-treated E4 parasites would largely resemble untreated controls and therefore provide limited additional mechanistic insight into either the drug's primary mode of action or the resistance mechanism.

Nonetheless, we agree that a comparative transcriptomic analysis of E4 under drug pressure could be informative, particularly with respect to fitness costs and compensatory adaptations. We plan to pursue this analysis in future studies.

2. The claim that SC83288 impacts the RNA m⁶A writer complex is currently weak and lacks direct experimental support. The authors have only discussed the involved enzymes (PfMT-A70/PfMT-A70.2) without providing any functional evidence to demonstrate how the drug alters m⁶A methylation.

We fully agree with the reviewer that the current data linking SC83288 to the RNA m⁶A writer complex remain indirect and do not yet establish a direct mechanistic interaction. Our conclusion in this regard is intentionally cautious and based on the observed reduction in global m⁶A RNA methylation levels following drug treatment, which raises the possibility of interference with the PfMT-A70/PfMT-A70.2 complex or associated regulatory pathways.

In the revised manuscript, we have therefore framed this aspect as a hypothesis rather than a demonstrated mechanism. Direct functional validation, such as enzymatic assays with the purified writer complex or genetic perturbation studies, will be required to clarify whether SC83288 directly targets this machinery or whether the observed m⁶A changes are secondary consequences of broader epigenetic and metabolic disruption.

In the present study, we deliberately focused on two conclusions that are supported by multiple independent lines of evidence: (i) PfDNMT2 as a primary molecular target contributing to parasite killing, and (ii) PfATP6 as a resistance determinant mediating drug sequestration. We appreciate the reviewer's recognition of the substantial additional work undertaken to strengthen these central findings.

3. The claim that SAHH is a direct target of SC83288 relies heavily on computational docking, which is not conclusive evidence. To validate this interaction, direct experimental proof is required. We urge the authors to perform in vitro binding assays (e.g., SPR, ABPP or ITC) with purified SAHH and the compound. Furthermore, an enzymatic activity assay demonstrating that SC83288 directly inhibits SAHH's function would provide the functional validation necessary to support this claim.

We thank the reviewer for this important comment. We would like to clarify that we do not consider PfSAHH a direct molecular target of SC83288. If our previous wording suggested otherwise, we apologize for the lack of clarity.

In the revised manuscript, we explicitly show that SC83288 does not inhibit the enzymatic activity of the NAD⁺-bound holo form of PfSAHH in vitro (Supplementary Fig. 12). This experimental result is consistent with our docking and MD simulations, which predict no meaningful interaction with the catalytically competent holo enzyme. Computational analyses suggested a possible interaction only with the apo form of PfSAHH at the NAD⁺-binding site; however, this state is unlikely to represent the dominant functional species in vivo.

We therefore conclude that PfSAHH is unlikely to be a direct enzymatic target of SC83288. Instead, we interpret the observed CETSA stabilization as an indirect effect, potentially resulting from elevated intracellular levels of adenine and MTA—both known SAHH product inhibitors—which accumulate upon drug treatment according to our metabolomics data.

To avoid further misunderstanding, we have revised the corresponding sections in the Results and Discussion to clearly state that PfSAHH is not considered a bona fide direct target of SC83288 and that the available evidence supports an indirect mechanism underlying the observed thermal shift.